# Generalized test utilities for long-tail performance in extreme multi-label classification

**Erik Schultheis**
Aalto University
Helsinki, Finland
`erik.schultheis@aalto.fi`

**Marek Wydmuch**
Poznan University of Technology
Poznan, Poland
`mwydmuch@cs.put.poznan.pl`

**Wojciech Kotłowski**
Poznan University of Technology
Poznan, Poland
`wkotlowski@cs.put.poznan.pl`

**Rohit Babbar**
University of Bath / Aalto University
Bath, UK / Helsinki, Finland
`rb2608@bath.ac.uk`

**Krzysztof Dembczyński**
Yahoo! Research / Poznan University of Technology
New York, USA / Poznan, Poland
`krzysztof.dembczynski@yahooinc.com`

## Abstract

Extreme multi-label classification (XMLC) is the task of selecting a small subset of relevant labels from a very large set of possible labels. As such, it is characterized by long-tail labels, i.e., most labels have very few positive instances. With standard performance measures such as precision@k, a classifier can ignore tail labels and still report good performance. However, it is often argued that correct predictions in the tail are more "interesting" or "rewarding," but the community has not yet settled on a metric capturing this intuitive concept. The existing propensity-scored metrics fall short on this goal by confounding the problems of long-tail and missing labels. In this paper, we analyze generalized metrics budgeted "at k" as an alternative solution. To tackle the challenging problem of optimizing these metrics, we formulate it in the *expected test utility* (ETU) framework, which aims to optimize the expected performance on a fixed test set. We derive optimal prediction rules and construct computationally efficient approximations with provable regret guarantees and robustness against model misspecification. Our algorithm, based on block coordinate ascent, scales effortlessly to XMLC problems and obtains promising results in terms of long-tail performance.

## 1   Introduction

Extreme multi-label classification (XMLC) is a challenging task with a wide spectrum of real-life applications, such as tagging of text documents [10], content annotation for multimedia search [14], or different type of recommendation [5, 1, 33, 44, 53, 28, 7]. Because of the nature of its applications, the typical approach in XMLC is to predict exactly $k$ labels (e.g., corresponding to $k$ slots in the user interface) which optimize a standard performance metric such as precision or (normalized) discounted cumulative gain. Given the enormous number of labels in XMLC tasks, which can reach millions or more, it is not surprising that many of them are very sparse, and hence make the label distribution strongly long-tailed [2]. It has been noticed that algorithms can achieve high performance on the standard metrics, but never predict any tail labels [42]. Therefore, there is a need to develop

37th Conference on Neural Information Processing Systems (NeurIPS 2023).

Table 1: Performance measures (%) on AmazonCat-13k of a classifier trained on the full set of labels and a classifier trained with only 1k head labels.

| Metric | full labels | | | head labels | | |
|---|---|---|---|---|---|---|
| | @1 | @3 | @5 | @1 (diff.) | @3 (diff.) | @5 (diff.) |
| Precision | 93.03 | 78.51 | 63.74 | 93.08 (+0.05%) | 76.42 (-2.66%) | 58.21 (-8.67%) |
| nDCG | 93.03 | 87.25 | 85.35 | 93.08 (+0.05%) | 85.75 (-1.71%) | 80.91 (-5.19%) |
| PS-Precision | 49.76 | 62.63 | 70.35 | 49.07 (-1.39%) | 57.71 (-7.84%) | 57.41 (-18.40%) |
| Macro-Precision | 13.28 | 32.65 | 44.16 | 4.31 (-67.54%) | 5.28 (-83.82%) | 4.32 (-90.21%) |
| Macro-Recall | 1.38 | 11.06 | 30.57 | 0.47 (-65.61%) | 2.69 (-75.71%) | 4.10 (-86.59%) |
| Macro-F1 | 2.26 | 14.67 | 32.84 | 0.74 (-67.37%) | 3.10 (-78.88%) | 3.77 (-88.51%) |
| Coverage | 15.19 | 40.53 | 60.88 | 5.11 (-66.32%) | 7.37 (-81.82%) | 7.52 (-87.65%) |

a metric that prefers "rewarding" [48], "diverse" [3], and "rare and informative" [34] labels over frequently-occurring head labels. Currently, the XMLC community attempts to capture this need using *propensity-scored* performance metrics [15]. These metrics give increased weight to tail labels, but have been derived from the perspective of missing labels, and as such they are not really solving the problem of tail labels [41].

In Table 1 we compare different metrics for budgeted at $k$ predictions. We train a PLT model [17] on the full AMAZONCAT-13K dataset [27] and a reduced version with the $1000$ most popular labels only. The test is performed for both models on the full set of labels. The standard metrics are only slightly perturbed by reducing the label space to the head labels. This holds even for propensity-scored precision, which decreases by just 1%-20% despite discarding over 90% of the label space. In contrast, macro measures and coverage decrease between 60% and 90% if tail labels are ignored. These results show that budgeted-at-$k$ macro measures might be very attractive in the context of long tails. Macro-averaging treats all the labels equally important, preventing the labels with a small number of positive examples to be ignored. Furthermore, the budget of $k$ labels "requires" the presence of long-tail labels in a compact set of predicted labels.

While we can easily use these measures to evaluate and compare different methods, we also would like to make predictions that directly optimize these metrics. The existing approaches to macro-averaged metrics consider the unconstrained case, in which label-wise optimization is possible [47, 11, 23, 16, 22, 26]. Each binary problem can be then solved under one of two frameworks for optimizing complex performance measures, namely *population utility* (PU) or *expected test utility* (ETU) [49, 12]. The former aims at optimizing the performance on the population level. The latter optimizes the performance directly on a given test set. Interestingly, in both frameworks the optimal solution is based on thresholding conditional label probabilities [12], but the resulting thresholds are different with the discrepancy diminishing with the size of the test set. The threshold tuning for PU is usually performed on a validation set [47, 26], while the exact optimization for ETU is performed on a test set. It requires cubic time in a general case and quadratic time in some special cases [49, 32]. Approximate solutions can be obtained in linear time [25, 12].

These approaches cannot be directly applied if prediction of exactly $k$ labels for each instance is required. In such case, the optimization problems for different labels are tightly coupled through this constraint, making the final problem much more difficult. Despite the fact that optimization of complex performance metrics is a well-established problem, considered not only in binary and multi-label classification as discussed above, but also in multi-class classification [30, 31], the results presented in this paper go beyond the state-of-the-art as budgeted-at-$k$ predictions have not yet been analyzed in this context. Let us underline that the requirement of $k$ predictions is natural for recommendation systems, in which exactly $k$ slots are available in the user interface to display recommendations. Even in situations where this does not apply, requiring the prediction to be "at $k$" can be advantageous, as it prevents trivial solutions such as predicting nothing (for precision) or everything (for recall).

In this paper, we investigate optimal solutions for the class of utility functions that can be linearly decomposed over labels into binary utilities, which includes both instance-wise weighted measures and macro-averages. We solve the problem in the ETU framework which is well-suited, for example, to recommendation tasks in which recommendations for all users or items are rebuilt in regular intervals. In this case, we can first obtain probability estimates of individual labels for each instance in the test set, and then provide optimal predictions for a given metric based on these estimates. We

derive optimal prediction rules and construct computationally efficient approximations with provable guarantees, formally quantifying the influence of the estimation error of the label probabilities on the suboptimality of the resulting classifier. This result is expressed in the form of a regret bound [4, 30, 22, 13]. It turns out that for most metrics of interest, a small estimation error results in at most a small drop of the performance, which confirms our method is viable for applications. Our general algorithm, based on block coordinate ascent, scales effortlessly to XMLC problems and obtains promising empirical results.

## 2 Setup and notation

Let $\boldsymbol{x} \in \mathcal{X}$ denote an input instance, and $\boldsymbol{y} \in \{0, 1\}^m =: \mathcal{Y}$ the vector indicating the relevant labels, distributed according to $\mathbb{P}(\boldsymbol{y}|\boldsymbol{x})$. We consider the prediction problem in the *expected test utility* (ETU) framework, that is, we assume that we are given a known set of $n$ instances $\boldsymbol{X} = [\boldsymbol{x}_1, \ldots, \boldsymbol{x}_n]^\mathsf{T} \in \mathcal{X}^n$ with unknown labels, on which we have to make predictions.[1] Our goal is to assign each instance $\boldsymbol{x}_i$ a set of exactly $k$ (out of $m$) labels represented as a $k$-hot vector $\hat{\boldsymbol{y}}_i \in \mathcal{Y}_k := \{\boldsymbol{y} \in \mathcal{Y} : \|\boldsymbol{y}\|_1 = k\}$, and we let $\hat{\boldsymbol{Y}} = [\hat{\boldsymbol{y}}_1, \ldots, \hat{\boldsymbol{y}}_n]^\mathsf{T}$ denote the entire $n \times m$ prediction matrix for a set of instances $\boldsymbol{X}$.

In the ETU framework, we treat $\boldsymbol{X}$ as given and only make an assumption about the labeling process for the test sample: the labels $\boldsymbol{y}_i \in \mathcal{Y}$ corresponding to $\boldsymbol{x}_i \in \mathcal{X}$ do not depend on any other instances, that is $\mathbb{P}(\boldsymbol{Y}|\boldsymbol{X}) = \prod_{i=1}^n \mathbb{P}(\boldsymbol{y}_i|\boldsymbol{x}_i)$, where we use $\boldsymbol{Y} = [\boldsymbol{y}_1, \ldots, \boldsymbol{y}_n]^\mathsf{T} \in \mathcal{Y}^n$ to denote the entire label matrix. We assume the quality of predictions $\hat{\boldsymbol{Y}}$ is jointly evaluated against the observed labels $\boldsymbol{Y}$ by a *task utility* $\Psi(\boldsymbol{Y}, \hat{\boldsymbol{Y}})$, and define the *optimal (Bayes) prediction* $\hat{\boldsymbol{Y}}^\star$ as the one maximizing the *expected task utility* $\Psi_{\mathrm{ETU}}$:

$$\hat{\boldsymbol{Y}}^\star = \operatorname*{argmax}_{\hat{\boldsymbol{Y}} \in \mathcal{Y}_k^n} \mathbb{E}_{\boldsymbol{Y}|\boldsymbol{X}}[\Psi(\boldsymbol{Y}, \hat{\boldsymbol{Y}})] =: \operatorname*{argmax}_{\hat{\boldsymbol{Y}} \in \mathcal{Y}_k^n} \Psi_{\mathrm{ETU}}(\hat{\boldsymbol{Y}}) . \tag{1}$$

We consider task utilities $\Psi(\boldsymbol{Y}, \hat{\boldsymbol{Y}})$ that linearly decompose over labels, i.e., there exists $\psi^j$ such that

$$\Psi(\boldsymbol{Y}, \hat{\boldsymbol{Y}}) = \sum_{j=1}^m \psi^j(\boldsymbol{y}_{:j}, \hat{\boldsymbol{y}}_{:j}) . \tag{2}$$

We allow the functions $\psi^j$ to be non-linear themselves and different for each label $j$. This is a large class of functions, which encompasses weighted instance-wise and macro-averaged utilities, the two groups of functions which we thoroughly analyze in the next sections.

Let us next define the binary confusion matrix $\boldsymbol{C}(\boldsymbol{y}, \hat{\boldsymbol{y}})$ for a vector $\boldsymbol{y}$ of $n$ ground truth labels and a corresponding vector $\hat{\boldsymbol{y}}$ of binary predictions:

$$\boldsymbol{C}(\boldsymbol{y}, \hat{\boldsymbol{y}}) := \begin{pmatrix} \frac{1}{n} \sum_{i=1}^n (1 - y_i)(1 - \hat{y}_i) & \frac{1}{n} \sum_{i=1}^n (1 - y_i)\hat{y}_i \\ \frac{1}{n} \sum_{i=1}^n y_i(1 - \hat{y}_i) & \frac{1}{n} \sum_{i=1}^n y_i\hat{y}_i \end{pmatrix} . \tag{3}$$

By indexing from 0, the entry $c_{00}$ corresponds to true negatives, $c_{01}$ to false positives, $c_{10}$ to false negatives, and $c_{11}$ to true positives. We define the *multi-label confusion tensor*[2] $\mathbf{C}(\boldsymbol{Y}, \hat{\boldsymbol{Y}}) := [\boldsymbol{C}(\boldsymbol{y}_{:1}, \hat{\boldsymbol{y}}_{:1}), \ldots, \boldsymbol{C}(\boldsymbol{y}_{:m}, \hat{\boldsymbol{y}}_{:m})]$ being the concatenation of binary confusion matrices of all $m$ labels.

Assuming the utility function (2) to be invariant under instance reordering, i.e., its value does not change if rows of both matrices are re-ordered using the same permutation, we can define $\Psi$ in terms of confusion matrices, instead of ground-truth labels and predictions (shown in Appendix A.1):

$$\Psi(\boldsymbol{Y}, \hat{\boldsymbol{Y}}) = \Psi(\mathbf{C}(\boldsymbol{Y}, \hat{\boldsymbol{Y}})) = \sum_{j=1}^m \psi^j(\boldsymbol{C}(\boldsymbol{y}_{:j}, \hat{\boldsymbol{y}}_{:j})) . \tag{4}$$

Finally, we assume that we have access to a *label probability estimator (LPE)* $\hat{\boldsymbol{\eta}}(\boldsymbol{x})$ that estimates the marginal probability of each label given the instance, $\boldsymbol{\eta}(\boldsymbol{x}) = (\eta_1(\boldsymbol{x}), \ldots, \eta_m(\boldsymbol{x})) := \mathbb{E}_{\boldsymbol{y}|\boldsymbol{x}}[\boldsymbol{y}]$. Such an LPE can be attained by fitting a model on an additional training set of $n'$ examples $(\boldsymbol{x}_i, \boldsymbol{y}_i)_{i=1}^{n'}$ using a proper composite loss function [37], which is a common approach in XMLC, e.g., [18].

---

[1] We use calligraphic letters for sets $\mathcal{S}$, bold font for vectors $\boldsymbol{v}$ with entries $v_i$, bold capital letters for matrices $\boldsymbol{Y}$ with entries $y_{ij}$, rows $\boldsymbol{y}_i$, and columns $\boldsymbol{y}_{:j}$. $\mathbb{1}[S]$ denotes the indicator of event $S$, and $[s] := \{1, \ldots, s\}$

[2] Notice that the confusion matrix can be computed either for multi-label predictions for a given instance $\boldsymbol{x}$ or binary predictions for label $j$ obtained on a set of instances $\boldsymbol{X}$. In the following, we focus on the latter.

# 3 Performance measures for tail labels

## 3.1 Instance-wise weighted utility functions

By assigning utility (or cost) to each correct/wrong prediction for each label, we can construct an *instance-wise weighted utility* $u_{\mathrm{w}} \colon \mathcal{Y} \times \mathcal{Y} \longrightarrow \mathbb{R}_{\geq 0}$ with labels $\boldsymbol{y} \in \mathcal{Y}$ and predictions $\hat{\boldsymbol{y}} \in \mathcal{Y}$ as

$$u_{\mathrm{w}}(\boldsymbol{y}, \hat{\boldsymbol{y}}) = \sum_{j=1}^{m} w_{00}^{j}(1 - y_j)(1 - \hat{y}_j) + w_{01}^{j}(1 - y_j)\hat{y}_j + w_{10}^{j} y_j (1 - \hat{y}_j) + w_{11}^{j} y_j \hat{y}_j . \quad (5)$$

We use $w_{00}^{j}$, $w_{01}^{j}$, $w_{10}^{j}$, and $w_{11}^{j}$ to express the utility of true negatives, false positives, false negatives, and true positives, respectively. The corresponding task loss results from summing over all instances. By interchanging the order of summation, we can see that it is of form (4):

$$\Psi(\boldsymbol{Y}, \hat{\boldsymbol{Y}}) = \sum_{i=1}^{n} u_{\mathrm{w}}(\boldsymbol{y}_i, \hat{\boldsymbol{y}}_i) = \sum_{j=1}^{m} w_{00}^{j} c_{00}^{j} + w_{01}^{j} c_{01}^{j} + w_{10}^{j} c_{10}^{j} + w_{11}^{j} c_{11}^{j} = \sum_{j=1}^{m} \psi^{j}(\boldsymbol{C}(\boldsymbol{y}_{:j}, \hat{\boldsymbol{y}}_{:j})) . \quad (6)$$

In other words, the instance-wise weighted utilities can be seen as *linear* confusion-based metrics. By choosing $w_{00}^{j} = w_{11}^{j} = m^{-1}$ and $w_{01}^{j} = w_{10}^{j} = 0$, the expression (5) reduces to the @$k$-variant of the *Hamming utility*. Similarly, $w_{11}^{j} = k^{-1}$ and $w_{00}^{j} = w_{01}^{j} = w_{10}^{j} = 0$ yield precision@$k$.

Another example is the popular *propensity-scoring* approach [15], commonly used as a tail-performance metric in XMLC. Here, the weights are computed based on training data through

$$w_{11}^{j,\mathrm{prop}} = k^{-1} \left( 1 + (\log n' - 1)(b + 1)^{a} (n' \hat{\pi}_j + b)^{-a} \right) , \quad (7)$$

where $n'$ is the number of training instances, $\hat{\pi}_j$ is the empirical prior of label $j$, and parameters $a$ and $b$ are (potentially) dataset-dependent. This form of weighting has been derived from a missing-labels perspective, so its application to tail labels is not fully justified [41]. Also, it introduces two more hyperparameters, which makes the interpretation and comparison of its values rather difficult. It is not less heuristical than other approaches like power-law or logarithmic weighting, given by:

$$w_{11}^{j,\mathrm{pl}} \propto \hat{\pi}_j^{-\beta}, \quad w_{11}^{j,\mathrm{log}} \propto -\log(\hat{\pi}_j) . \quad (8)$$

## 3.2 Macro-average of non-decomposable utilities

Macro-averaging usually concerns non-decomposable binary utilities such as the F-measure. In this case, we set up $\psi^{j}(\cdot, \cdot) = m^{-1}\psi(\cdot, \cdot)$ for all labels $j$ in (2), yielding:

$$\Psi(\boldsymbol{Y}, \hat{\boldsymbol{Y}}) = m^{-1} \sum_{j=1}^{m} \psi(\boldsymbol{y}_{:j}, \hat{\boldsymbol{y}}_{:j}) = m^{-1} \sum_{j=1}^{m} \psi(\boldsymbol{C}(\boldsymbol{y}_{:j}, \hat{\boldsymbol{y}}_{:j})) . \quad (9)$$

By using $\psi_{\mathrm{pr}}(\boldsymbol{C}) \coloneqq c_{11}/(c_{11} + c_{01})$, this becomes *macro-precision*, for $\psi_{\mathrm{rec}}(\boldsymbol{C}) \coloneqq c_{11}/(c_{11} + c_{10})$ we get *macro-recall*, and for $\psi_{\mathrm{F}_\beta}(\boldsymbol{C}) \coloneqq (\beta + 1)c_{11}/((1+\beta)c_{11} + \beta^2 c_{10} + c_{01})$ the *macro-F-measure*.

Another measure that is promising for the evaluation of long-tailed performance is *coverage*. It is sometimes used as an auxiliary measure in XMLC [15, 3, 42, 41]. This metric detects for how many different labels the classifier is able to make *at least one* correct prediction. In our framework, this is achieved by using an indicator function on the true positives, $\psi_{\mathrm{cov}}(\boldsymbol{C}) \coloneqq \mathbb{1}[c_{11} > 0]$.

# 4 Optimal predictions

Let us start with the observation that the label probabilities $\boldsymbol{\eta}$ are sufficient to make optimal predictions. With the assumption that $\mathbb{P}(\boldsymbol{Y}|\boldsymbol{X}) = \prod_{i=1}^{n} \mathbb{P}(\boldsymbol{y}_i|\boldsymbol{x}_i)$, we obtain (cf. Appendix A.2):

$$\mathbb{E}_{\boldsymbol{Y}|\boldsymbol{X}}[\Psi(\boldsymbol{Y}, \hat{\boldsymbol{Y}})] = \sum_{j=1}^{m} \sum_{\boldsymbol{y}' \in \{0,1\}^n} \left( \prod_{i=1}^{n} \eta_j(\boldsymbol{x}_i) y_i' + (1 - \eta_j(\boldsymbol{x}_i))(1 - y_i') \right) \psi^{j}(\boldsymbol{y}', \hat{\boldsymbol{y}}_{:j}) . \quad (10)$$

This equation lays out a daunting optimization task, as is requires summing over $2^n$ summands $\boldsymbol{y}'$. In case of binary classification, there exist methods to solve the problem exactly in $\mathcal{O}(n^3)$, or in $\mathcal{O}(n^2)$

in some special cases [32]. By using *semi-empirical* quantities (defined below), [12] provides an approximate algorithm that runs in $O(n)$. Following this approach, we construct a semi-empirical ETU approximation. If this approximation results in a *linear* function of the predictions, the problem decomposes over instances and can be solved easily. Otherwise, we use an algorithm that leads to locally optimal predictions. A minor modification of this algorithm can be used for coverage.

## 4.1 Semi-empirical ETU approximation

Since the entries of the confusion matrix are linearly dependent, it suffices to use three independent combinations. More precisely, we parameterize the confusion matrix by the true positives $t = c_{11}$, predicted positives $q = c_{11} + c_{01}$, and ground-truth positives $p = c_{11} + c_{10}$, and use $\boldsymbol{t} = (t_1, \ldots, t_m)$, $\boldsymbol{q} = (q_1, \ldots, q_m)$, and $\boldsymbol{p} = (p_1, \ldots, p_m)$ to reformulate the ETU objective:

$$\Psi_{\mathrm{ETU}}(\hat{\boldsymbol{Y}}) = \mathbb{E}_{\boldsymbol{Y}|\boldsymbol{X}}[\Psi(\mathbf{C}(\boldsymbol{Y}, \hat{\boldsymbol{Y}}))] = \mathbb{E}_{\boldsymbol{Y}|\boldsymbol{X}}[\Psi(\boldsymbol{t}, \boldsymbol{q}, \boldsymbol{p})] = \mathbb{E}_{\boldsymbol{Y}|\boldsymbol{X}}\left[\sum_{j=1}^m \psi^j(t_j, q_j, p_j)\right]. \quad (11)$$

In order to compute $\Psi_{\mathrm{ETU}}$, one needs to take into account every possible combination of confusion-matrix values, and calculate the corresponding value of $\Psi$, which is then averaged according to the respective probabilities. A computationally easier approach is to take the expectation over the labels first, leading to *semi-empirical* quantities:

$$\tilde{\boldsymbol{t}} := \mathbb{E}_{\boldsymbol{Y}|\boldsymbol{X}}[\boldsymbol{t}], \quad \tilde{\boldsymbol{q}} := \mathbb{E}_{\boldsymbol{Y}|\boldsymbol{X}}[\boldsymbol{q}] = \boldsymbol{q}, \quad \tilde{\boldsymbol{p}} := \mathbb{E}_{\boldsymbol{Y}|\boldsymbol{X}}[\boldsymbol{p}], \quad (12)$$

where $\tilde{\boldsymbol{q}} = \boldsymbol{q}$ follows because the number of predicted positives depends only on the predictions $\hat{\boldsymbol{Y}}$. This allows us to define the semi-empirical ETU risk

$$\tilde{\Psi}_{\mathrm{ETU}}(\hat{\boldsymbol{Y}}) := \Psi(\tilde{\boldsymbol{t}}, \boldsymbol{q}, \tilde{\boldsymbol{p}}) \approx \mathbb{E}_{\boldsymbol{Y}|\boldsymbol{X}}[\Psi(\boldsymbol{t}, \boldsymbol{q}, \boldsymbol{p})] = \Psi_{\mathrm{ETU}}(\hat{\boldsymbol{Y}}). \quad (13)$$

In particular, the third argument to $\Psi$, $\tilde{\boldsymbol{p}}$, is a constant that does not depend on predictions.

Note that, if $\Psi$ is *linear* in all arguments *depending on the random variable $\boldsymbol{Y}$*, then the approximation is exact, due to the linearity of expectations. Aside from instance-wise measures, which we showed to be linear above, the approximation is also exact for the more general class of functions of the form

$$\psi(t, q, p) = f_{\mathrm{t}}(q) \cdot t + f_{\mathrm{q}}(q) + f_{\mathrm{p}}(q) \cdot p. \quad (14)$$

An important example is macro-precision, with $f_{\mathrm{t}}(q) = q^{-1}$ and $f_{\mathrm{q}} = f_{\mathrm{p}} = 0$. In the general case, $\tilde{\Psi}_{\mathrm{ETU}}$ as a surrogate for $\Psi_{\mathrm{ETU}}$ leads only to $\mathcal{O}(1/\sqrt{n})$ error as will be shown in Theorem 5.2, while substantially simplifying the optimization process.

## 4.2 Linear confusion-matrix measures

We start the discussion on optimization of (13) with a special case in which $\tilde{\Psi}_{\mathrm{ETU}}$ is *linear in the prediction-dependent arguments* $t, q$, that is, if

$$\psi(t, q, p) = f_{\mathrm{t}}(p) \cdot t + f_{\mathrm{q}}(p) \cdot q, \quad (15)$$

i.e., both $f_{\mathrm{t}}(p)$ and $f_{\mathrm{q}}(p)$ depend on $p$ only, and $f_{\mathrm{p}}(p) \cdot p$ can be dropped as it is a constant.

Aside from instance-wise weighted utilities (cf. Appendix A.3), which are linear in all arguments, this form also holds for weights dependent on the (empirical) label priors, e.g., power law weights of the form $f_{\mathrm{t}}(p) = p^{-\alpha}$ and $f_{\mathrm{q}} = 0$, which reduce to macro-recall for $\alpha = 1$. If one defines the weights with respect to externally determined label priors, i.e., approximations to $\mathbb{E}[y_j]$, which are fixed and thus independent of the test sample $\boldsymbol{X}$, then the power-law metrics turn into instance-wise weighted utilities.

From (6) we know that we can reformulate the optimization problem using an instance-wise weighted utility $u_{\mathrm{w}}$ with weights:

$$w_{11}^j = f_{\mathrm{t}}(p_j) + f_{\mathrm{q}}(p_j), \quad w_{01}^j = f_{\mathrm{q}}(p_j), \quad w_{10}^j = 0, \quad w_{00}^j = 0. \quad (16)$$

Hence, the optimal predictions can be derived for each instance $\boldsymbol{x} \in \mathcal{X}$ separately, leading to

$$\hat{\boldsymbol{y}}^* = \mathrm{argmax}_{\hat{\boldsymbol{y}} \in \mathcal{Y}_k} \mathbb{E}_{\boldsymbol{y}|\boldsymbol{x}}[u_{\mathrm{w}}(\boldsymbol{y}, \hat{\boldsymbol{y}})]. \quad (17)$$

Plugging in the definition of $u_{\mathrm{w}}$ from (5), and collecting terms, the expected loss is of the form

$$\mathbb{E}_{\boldsymbol{y}|\boldsymbol{x}}[u_{\mathrm{w}}(\boldsymbol{y}, \hat{\boldsymbol{y}})] = \sum_{j=1}^m \mathbb{E}_{\boldsymbol{y}|\boldsymbol{x}}\left[w_{11}^j y_j \hat{y}_j + w_{01}^j (1 - y_j) \hat{y}_j\right] = \sum_{j=1}^m \hat{y}_j \left(\eta_j(\boldsymbol{x}) f_{\mathrm{t}}(p_j) + f_{\mathrm{q}}(p_j)\right), \quad (18)$$

where we call $g_j(\boldsymbol{x}) := \eta_j(\boldsymbol{x}) f_\mathrm{t}(p_j) + f_\mathrm{q}(p_j)$ the *gain* of predicting label $j$ for a given instance $\boldsymbol{x}$. The optimal prediction is to select the $k$ labels with the largest values $g_j(\boldsymbol{x})$,

$$\hat{\boldsymbol{y}}^* = \operatorname*{argmax}_{\hat{\boldsymbol{y}} \in \hat{\mathcal{Y}}_k} \sum_{j=1}^{m} g_j(\boldsymbol{x}) \hat{y}_j = \text{select-top-}k(\boldsymbol{g}(\boldsymbol{x})). \tag{19}$$

Here, select-top-$k(\boldsymbol{g})$ denotes an operation that maps a vector of scores to a binary vector that contains 1s only at the positions of the $k$ largest elements of $\boldsymbol{g}$.

### 4.3 General non-decomposable macro measures

Finally, we turn to the general problem for budgeted-at-$k$ macro-measures based on non-linear $\Psi$. As this can be a very hard discrete optimization problem in general, we use an iterative approach based on *block-coordinate ascent* that constructs a sequence of predictions, $\hat{\boldsymbol{Y}}^0, \hat{\boldsymbol{Y}}^1, \ldots$, with non-decreasing utility, so that we end up with a solution that is locally optimal.

Assume the predictions are fixed for all instances except $\boldsymbol{x}_s$, where they are given by $\boldsymbol{z}$. In that case, we can write the semi-empirical quantities from (12) as

$$\tilde{\boldsymbol{t}} = \tfrac{1}{n} \Big( \eta_j(\boldsymbol{x}_s) z_j + \sum_{i \in [n] \setminus \{s\}} \eta_j(\boldsymbol{x}_i) \hat{y}_{ij} \Big), \quad (20)$$

with analog expansions for $\boldsymbol{q}$ and $\tilde{\boldsymbol{p}}$. Plugging into (13) leads to the following optimization:

---

**Algorithm 1** BCA$(\boldsymbol{X}, \hat{\boldsymbol{\eta}}, k, \epsilon)$

1: $\hat{\boldsymbol{y}}_i \leftarrow$ select-random-$k(m)$ for all $i \in [n]$
2: $\tilde{\boldsymbol{t}} \leftarrow \frac{1}{n} \sum_{i=1}^{n} \hat{\boldsymbol{\eta}}(\boldsymbol{x}_i) \odot \hat{\boldsymbol{y}}_i$
3: $\boldsymbol{q} \leftarrow \frac{1}{n} \sum_{i=1}^{n} \hat{\boldsymbol{y}}_i$, $\tilde{\boldsymbol{p}} \leftarrow \frac{1}{n} \sum_{i=1}^{n} \hat{\boldsymbol{\eta}}(\boldsymbol{x}_i)$
4: $u_{\text{old}} \leftarrow -\infty$, $u_{\text{new}} \leftarrow \Psi(\tilde{\boldsymbol{t}}, \boldsymbol{q}, \tilde{\boldsymbol{p}})$
5: **while** $u_{\text{new}} > u_{\text{old}} + \epsilon$ **do**
6:     **for** $s \in$ shuffle$([n])$ **do**
7:         $\tilde{\boldsymbol{t}} \leftarrow \tilde{\boldsymbol{t}} - \frac{1}{n} \hat{\boldsymbol{\eta}}(\boldsymbol{x}_s) \odot \hat{\boldsymbol{y}}_s$, $\boldsymbol{q} \leftarrow \boldsymbol{q} - \frac{1}{n} \hat{\boldsymbol{y}}_s$
8:         **for** $j \in [m]$ **do**
9:             $\psi^j(1) \leftarrow \psi(\tilde{t}_j + \frac{1}{n} \hat{\eta}_j(\boldsymbol{x}_s), q_j + \frac{1}{n}, \tilde{p}_j)$
10:           $\psi^j(0) \leftarrow \psi(\tilde{t}_j, q_j, \tilde{p}_j)$
11:           $g_j \leftarrow \psi^j(1) - \psi^j(0)$
12:         $\hat{\boldsymbol{y}}_s \leftarrow$ select-top-$k(\boldsymbol{g})$
13:         $\tilde{\boldsymbol{t}} \leftarrow \tilde{\boldsymbol{t}} + \frac{1}{n} \hat{\boldsymbol{\eta}}(\boldsymbol{x}_s) \odot \hat{\boldsymbol{y}}_s$, $\boldsymbol{q} \leftarrow \boldsymbol{q} + \frac{1}{n} \hat{\boldsymbol{y}}_s$
14:     $u_{\text{old}} \leftarrow u_{\text{new}}$, $u_{\text{new}} \leftarrow \Psi(\tilde{\boldsymbol{t}}, \boldsymbol{q}, \tilde{\boldsymbol{p}})$
15: **return** $\hat{\boldsymbol{Y}}$

---

$$\max_{\boldsymbol{z} \in \mathcal{Y}_k} \sum_{j=1}^{m} \psi^j \Big( \tfrac{1}{n} \eta_j(\boldsymbol{x}_s) z_j + \tfrac{1}{n} \sum_{i \in [n] \setminus \{s\}} \eta_j(\boldsymbol{x}_i) \hat{y}_{ij}, \ \tfrac{1}{n} z_j + \tfrac{1}{n} \sum_{i \in [n] \setminus \{s\}} \hat{y}_{ij}, \ \tfrac{1}{n} \sum_{i=1}^{n} \eta_j(\boldsymbol{x}_i) \Big). \tag{21}$$

As everything except $z_j \in \{0, 1\}$ is given, we can interpret $\psi^j$ as a linear function of $z_j$, and define a gain vector with elements $g_j = \psi^j(1) - \psi^j(0)$. The optimal prediction $\boldsymbol{z}^*$ is then given by $\boldsymbol{z}^* = \text{select-top-}k(\boldsymbol{g})$, in a similar form as in case of linear confusion-matrix measures. We get $\hat{\boldsymbol{Y}}^{t+1}$ by replacing the $s^{\text{th}}$ row of $\hat{\boldsymbol{Y}}^t$ with $\boldsymbol{z}^*$, and know that $\tilde{\Psi}_{\text{ETU}}(\hat{\boldsymbol{Y}}^{t+1}) \geq \tilde{\Psi}_{\text{ETU}}(\hat{\boldsymbol{Y}}^t)$. Then we switch to the next instance $s \leftarrow s + 1$, and repeat this process until no more progress is made.

The algorithm starts by predicting $k$ random labels for each instance. To speed up the computations we cache two quantities, $\tilde{t}_j^t$ and $q_j^t$, for each label:

$$\tilde{t}_j^0 := \tfrac{1}{n} \sum_{i=1}^{n} \eta_j(\boldsymbol{x}_i) \hat{y}_{ij}^0, \qquad \tilde{t}_j^{t+1} := \tilde{t}_j^t + \tfrac{1}{n} \eta_j(\boldsymbol{x}_s)(\hat{y}_{sj}^{t+1} - \hat{y}_{sj}^t),$$
$$q_j^0 := \tfrac{1}{n} \sum_{i=1}^{n} \hat{y}_{ij}^0, \qquad q_j^{t+1} := q_j^t + \tfrac{1}{n} \big( \hat{y}_{sj}^{t+1} - \hat{y}_{sj}^t \big). \tag{22}$$

With this, we can compute (21) in $\mathcal{O}(m)$ time using the following formulas:

$$\psi^j(1) := \psi^j \big( \tilde{t}_j^t + \tfrac{1}{n} \eta_j(\boldsymbol{x}_s)(1 - \hat{y}_{sj}^t), q_j^t + \tfrac{1}{n}(1 - \hat{y}_{sj}^t), \tilde{p}_j \big),$$
$$\psi^j(0) := \psi^j \big( \tilde{t}_j^t - \tfrac{1}{n} \eta_j(\boldsymbol{x}_s) \hat{y}_{sj}^t, q_j^t - \tfrac{1}{n} \hat{y}_{sj}^t, \tilde{p}_j \big). \tag{23}$$

The block coordinate ascent (BCA) procedure is shown in more detail in Algorithm 1. Obviously, the actual implementation cannot use the unknown values $\boldsymbol{\eta}$, but instead has to rely on the LPE estimates $\hat{\boldsymbol{\eta}}(\boldsymbol{x}_i)$. The stopping criterion for the algorithm is whether, after going over a whole set of instances, the improvement in the objective value over the previous value is lower than $\epsilon$, which ensures that the algorithm terminates for every bounded utility in $\mathcal{O}(1/\epsilon)$. In practice, even with small $\epsilon$, the algorithm usually terminates after a few iterations. The time and space complexity of the single iteration are both $\mathcal{O}(nm)$. If $\psi$ is a linear function, corresponding to an instance-wise weighted utility (6) such as macro-recall, the algorithm recovers the optimal solution in the first iteration, stopping after the second.

In general, Algorithm 1 requires multiple iterations. This can be computationally expensive and requires all of the data to be available at once. We thus propose a greedy algorithm that takes into account only statistics of *previously seen* instances while performing a *single pass* over the dataset, which allows for semi-online optimization. The greedy algorithm is outlined in Appendix B.4.

## 4.4 Optimization of coverage

One measure for which the ETU approximation is not exact is coverage as $\psi_{\mathrm{cov}}(t, q, p) \coloneqq \mathbb{1}[t > 0]$ is *nonlinear*. In this case, we can do better than Algorithm 1, by reformulating $\Psi_{\mathrm{ETU}}$ for coverage as

$$\Psi_{\mathrm{ETU}}(\hat{Y}) = \mathbb{E}_{Y|X}\Big[m^{-1}\sum_{j=1}^{m}\mathbb{1}[t_j > 0]\Big] = 1 - m^{-1}\sum_{j=1}^{m}\prod_{i=1}^{n}(1 - \eta_j(\boldsymbol{x}_i)\hat{y}_{ij}), \qquad (24)$$

and performing block coordinate ascent directly on this expression, as detailed in Appendix B.3.

## 5 Regret bounds

Computing optimal predictions relies on an access to the conditional marginal probabilities $\boldsymbol{\eta}(\boldsymbol{x})$. In practice, however, $\boldsymbol{\eta}(\boldsymbol{x})$ are unknown, and are replaced by the LPE $\hat{\boldsymbol{\eta}}(\boldsymbol{x})$ to do inference (*plug-in* approach). Furthermore, we replaced the ETU objective $\Psi_{\mathrm{ETU}}$ with an approximation $\tilde{\Psi}_{\mathrm{ETU}}$. As generally $\hat{\boldsymbol{\eta}}(\boldsymbol{x}) \neq \boldsymbol{\eta}(\boldsymbol{x})$ and $\Psi_{\mathrm{ETU}} \neq \tilde{\Psi}_{\mathrm{ETU}}$, this procedure may result in sub-optimal predictions, the errors of which we would like to control.

If we are able to control the change of the utility under small changes in its arguments, we can show that the suboptimality of the plug-in predictor relative to the optimal predictor (called $\Psi$-*regret*) decreases with increasing test size $n$ and decreasing probability estimation error.

To this end we assume that each $\psi^j$ is a *p-Lipschitz* function, which allows us to bound the approximation error.

**Definition 5.1** (*p*-Lipschitz [13]). A binary classification metric $\psi(t, q, p)$ is said to be *p*-Lipschitz if

$$|\psi(t, q, p) - \psi(t', q', p')| \leq L_{\mathrm{t}}(p)|t - t'| + L_{\mathrm{q}}(p)|q - q'| + L_{\mathrm{p}}(p)|p - p'|, \qquad (25)$$

for any $q, q' \in [0, 1]$, $p, p' \in (0, 1)$, $0 \leq t \leq \min(p, q)$, and $0 \leq t' \leq \min(p', q')$. The constants $L_{\mathrm{t}}(p), L_{\mathrm{q}}(p), L_{\mathrm{p}}(p)$ are allowed to depend on $p$, in contrast to the standard Lipschitz functions.

As shown in Appendix C.3, most of metrics of interest satisfy *p*-Lipschitz assumption, including the linear confusion-matrix measures (6) with fixed weights (e.g., Hamming utility, precision), macro-recall, macro-F-measure, etc., with macro-precision and coverage being notable exceptions.

**Theorem 5.2.** *Let each $\psi^j$ be p-Lipschitz with constants $L_{\mathrm{t}}^j(p), L_{\mathrm{q}}^j(p), L_{\mathrm{p}}^j(p)$. For any $\hat{Y}$ it holds:*

$$|\Psi_{\mathrm{ETU}}(\hat{Y}; X) - \tilde{\Psi}_{\mathrm{ETU}}(\hat{Y}; X)| \leq \frac{1}{2\sqrt{n}}\Big(\sum_{j=1}^{m}(L_{\mathrm{t}}^j(\tilde{p}_j) + L_{\mathrm{p}}^j(\tilde{p}_j))\Big). \qquad (26)$$

Thus, using $\tilde{\Psi}_{\mathrm{ETU}}$ as a surrogate for $\Psi_{\mathrm{ETU}}$ leads only to $\mathcal{O}(1/\sqrt{n})$ error, diminishing with the test size, while substantially simplifying the optimization process.

Given a decomposable metric of the form (4), let $\tilde{Y}^\dagger$ be the plug-in prediction matrix optimizing the semi-empirical ETU with plugged-in probability estimates, $\Psi\big(\mathbb{E}_{\boldsymbol{y}\sim\hat{\boldsymbol{\eta}}(X)}[\boldsymbol{t}], \boldsymbol{q}, \mathbb{E}_{\boldsymbol{y}\sim\hat{\boldsymbol{\eta}}(X)}[\boldsymbol{p}]\big)$.

**Theorem 5.3.** *Let $\tilde{Y}^\dagger$ be defined as above. Under the assumptions of Theorem 5.2:*

$$\Psi_{\mathrm{ETU}}(\hat{Y}^\star; X) - \Psi_{\mathrm{ETU}}(\tilde{Y}^\dagger; X) \leq \frac{m}{\sqrt{n}}B + 2\frac{\sqrt{m}}{n}B\sum_{i=1}^{n}\|\boldsymbol{\eta}(\boldsymbol{x}_i) - \hat{\boldsymbol{\eta}}(\boldsymbol{x}_i)\|_2, \qquad (27)$$

*where $B \coloneqq \sqrt{m^{-1}\sum_{j=1}^{m}(L_{\mathrm{t}}^j(\tilde{p}_j) + L_{\mathrm{p}}^j(p_j))^2}$ is the quadratic mean of the Lipschitz constants.*

A similar statement, presented in Appendix C.4, can be made for the unapproximated ETU case.

Thus, the methods described in Section 4 can be used with probability estimates replacing the true marginals, and as long as the estimator is reliable, the resulting predictions will have small $\Psi$-regret. This also justifies the plug-in approach used in the experiments in Section 7.

## 6 Efficient inference

The optimization algorithms introduced so far need to obtain $\hat{\eta}_j(\boldsymbol{x}_i)$ for *all* labels and instances first. Then, the BCA inference is of $\mathcal{O}(nm)$ time and $\mathcal{O}(nm)$ space complexity for a single iteration. This

is problematic in the setting of XMLC, where many methods aim to predict probabilities only for top labels in time sublinear in $m$. Fortunately, this characteristic of XMLC algorithms can be combined with the introduced algorithms to efficiently obtain an approximate solution. Instead of predicting all $\boldsymbol{\eta}$, we can predict probabilities only for top-$k'$ labels with highest $\hat{\eta}_j$, where $k \ll k' \ll m$. For all other labels we then assume $\hat{\eta}_j = 0$. Under the natural assumption that $\psi$ is *non-decreasing* in true positives and *non-increasing* in predicted positives, we can leverage the sparsity and consider labels with non-zero $\hat{\eta}_j$ only (using sparse vectors to represent $\hat{\boldsymbol{\eta}}(\boldsymbol{x}_i)$) to reduce the time and space complexity to $\mathcal{O}(n(k' + k\log k))$ and $\mathcal{O}(n(k' + k))$, respectively. As in real-world datasets the number of relevant labels $\|\boldsymbol{y}\|_1$ is much lower than $m$, and most $\eta_j(\boldsymbol{x}_i)$ are close to 0, with reasonably selected $k'$, according to Theorem 5.3, we should only slightly increase the regret. A pseudocode for the sparse variant of Algorithm 1 can be found in Appendix D.

Alternatively, we can leverage probabilistic label trees (PLTs) [16], a popular approach in XMLC [35, 45, 19, 50, 7], to efficiently search for "interesting" labels. Originally, PLTs find top labels with the highest $\hat{\eta}_j$. In [46], an $A^*$-search algorithm has been introduced for finding $k$ labels with highest value of $g_j\eta_j(\boldsymbol{x})$, where $g_j \in [0, \infty)$ is a gain assigned to label $j$. In Appendix F, we present a more general version of this procedure that can be used to efficiently obtain an exact solution of presented BCA or Greedy algorithms for some of the metrics considered in this work.

## 7 Experiments

To empirically test the introduced framework, we use popular benchmarks from the XMLC repository [6]. We train the LIGHTXML [18] model (with suggested default hyper-parameters) on provided training sets to obtain $\hat{\boldsymbol{\eta}}$ for all test instances. We then plug these estimates into different inference strategies and report the results across the discussed measures. To run the optimization algorithm efficiently, we use $k' = 100$ or $k' = 1000$ to pre-select for each instance the top $k'$ labels with the highest $\hat{\eta}_j$ as described in Section 6.[3]

We use the following inference strategies:

- TOP-K– the optimal strategy for precision@$k$: selection of $k$ labels with the highest $\eta_j$ (default prediction strategy in many XMLC methods).
- PS-K– the optimal strategy for propensity-scored precision@$k$: selection of $k$ labels with the highest $w_{11}^{j,\text{prop}}\eta_j$, with $w_{11}^{j,\text{prop}}$ given by the empirical model of Jain et al. [15] (Equation 7) with values $a$ and $b$ recommended by the authors.
- POW-K, LOG-K– the optimal strategy for power-law and log weighted instance-wise utilities: selection of $k$ labels with the highest $w_{11}^{j,\text{pl}}\eta_j$ or $w_{11}^{j,\text{log}}\eta_j$. For power-law, we use $\beta = 0.5$.
- MACRO-P$_{\text{BCA}}$, MACRO-R$_{\text{BCA}}$, MACRO-F1$_{\text{BCA}}$, COV$_{\text{BCA}}$– the block coordinate ascent (Algorithm 1) for optimizing macro-precision, -recall, -F1, and coverage,

We expect a strategy suited for a given metric to obtain the best results on this metric. Nevertheless, this might not always be the case, as in the derivation of our algorithms, we needed to apply different types of approximation to scale them to XMLC problems. We are mainly interested in the performance on the general non-decomposable macro measures since they seem to be well-tailored to long tails, and their optimization is the most challenging. The results are presented in Table 2. Notice that in almost all cases, the specialized inference strategies are indeed the best on the measure they aim to optimize and achieve substantial gains on corre-

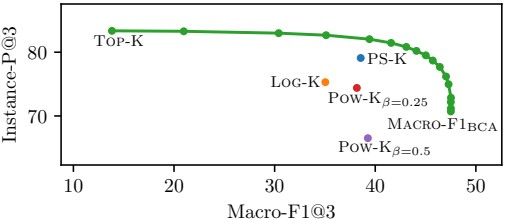

Figure 1: Results of an inference strategy with a mixed utility on AMAZONCAT-13K and $k = 3$. The green line shows the results for different interpolations between two measures.

sponding metrics compared to the basic TOP-K inference. The other weighted strategies, PS-K, POW-K, LOG-K, usually provide a much smaller improvement over TOP-K and never beat strategies designed for specific macro measures. As the reported performance depends on three things: the inherent difficulty of the data, the success of the inference algorithm, and the quality of the provided

---

[3]A code to reproduce all the experiments: `https://github.com/mwydmuch/xCOLUMNs`

Table 2: Results of different inference strategies on @$k$ measures with $k \in \{3, 5, 10\}$. Notation: P—precision, R—recall, F1—F1-measure, Cov—Coverage. The green color indicates cells in which the strategy matches the metric. The best results are in **bold** and the second best are in *italic*.

| Inference strategy | Instance @3 | | Macro @3 | | | | Instance @5 | | Macro @5 | | | | Instance @10 | | Macro @10 | | | |
|---|---|---|---|---|---|---|---|---|---|---|---|---|---|---|---|---|---|---|
| | P | R | P | R | F1 | Cov | P | R | P | R | F1 | Cov | P | R | P | R | F1 | Cov |
| *EURLEX-4K, $k'=100$* | | | | | | | | | | | | | | | | | | |
| Top-K | **74.20** | **44.21** | 24.85 | 16.45 | 18.63 | 31.27 | **62.31** | **60.59** | 27.43 | 25.99 | 25.50 | 39.76 | **39.29** | **75.11** | 22.15 | 36.59 | *26.01* | 47.55 |
| PS-K | 69.01 | 41.06 | 30.12 | 23.72 | 25.08 | 40.29 | 60.77 | 59.17 | 29.10 | 29.89 | 28.17 | 44.03 | 38.98 | 74.54 | 21.36 | *38.27* | 25.78 | 49.47 |
| Pow-K $_{\beta=0.5}$ | 65.35 | 38.95 | 30.49 | **25.26** | *25.93* | 42.11 | 58.53 | 57.10 | 28.88 | *31.25* | *28.53* | 45.48 | 38.48 | 73.66 | 20.77 | **38.85** | 25.43 | 50.13 |
| Log-K | *71.03* | *42.30* | 29.04 | 22.10 | 23.76 | 37.99 | *61.51* | *59.87* | 28.48 | 28.47 | 27.26 | 42.32 | *39.12* | *74.80* | 21.60 | 37.49 | 25.83 | 48.61 |
| Macro-P$_{BCA}$ | 41.13 | 24.29 | **38.78** | 16.77 | 20.63 | 40.60 | 31.05 | 30.33 | **38.70** | 18.46 | 21.59 | 41.66 | 18.45 | 35.65 | **37.94** | 20.28 | 21.82 | 42.89 |
| Macro-R$_{BCA}$ | 41.47 | 24.61 | 30.36 | *25.16* | 24.37 | 44.26 | 38.32 | 37.54 | 28.97 | **31.27** | 27.05 | *49.04* | 29.80 | 57.58 | 22.99 | 38.14 | 25.99 | *52.15* |
| Macro-F1$_{BCA}$ | 61.96 | 36.94 | *34.42* | 25.13 | **27.59** | *44.52* | 56.18 | 54.70 | *33.82* | 29.52 | **30.13** | 45.93 | 33.55 | 64.29 | *32.78* | 32.21 | **30.47** | 47.56 |
| Cov$_{BCA}$ | 24.00 | 14.03 | 31.46 | 21.30 | 19.05 | **47.19** | 16.40 | 15.95 | 29.86 | 24.53 | 18.81 | **50.09** | 9.31 | 18.05 | 26.99 | 28.28 | 17.41 | **52.84** |
| *AMAZONCAT-13K, $k'=100$* | | | | | | | | | | | | | | | | | | |
| Top-K | **83.35** | **63.01** | 25.71 | 11.37 | 13.83 | 33.05 | **68.01** | **79.01** | 33.92 | 31.34 | 29.28 | 52.60 | **41.72** | **89.41** | 24.63 | 51.76 | 28.84 | 68.61 |
| PS-K | *79.09* | *60.15* | 43.89 | 38.64 | 38.56 | 62.06 | *66.63* | *77.89* | 38.41 | 45.00 | 38.34 | 65.82 | *41.50* | *89.11* | 22.56 | 58.69 | 24.29 | 74.29 |
| Pow-K $_{\beta=0.5}$ | 66.52 | 50.75 | 38.18 | 46.42 | *39.26* | 67.76 | 56.94 | 68.14 | 31.65 | *54.06* | 36.90 | 71.95 | 37.86 | 83.91 | 19.26 | *64.05* | 26.29 | 77.27 |
| Log-K | 75.30 | 56.79 | 41.09 | 33.92 | 35.04 | 56.05 | 64.07 | 75.21 | 37.40 | 41.86 | 36.78 | 61.84 | 40.66 | 88.05 | 23.09 | 55.85 | 28.91 | 71.37 |
| Macro-P$_{BCA}$ | 54.97 | 41.61 | **64.27** | 29.22 | 35.76 | 76.18 | 41.53 | 49.66 | **63.81** | 30.43 | 35.99 | 76.75 | 25.32 | 57.33 | **61.84** | 33.06 | *35.08* | 78.17 |
| Macro-R$_{BCA}$ | 47.74 | 37.13 | 31.40 | **58.32** | 34.68 | *80.71* | 38.93 | 48.26 | 25.17 | **65.47** | 30.37 | *82.91* | 17.53 | 72.63 | 26.50 | 61.99 | 23.22 | *84.84* |
| Macro-F1$_{BCA}$ | 70.61 | 53.86 | *51.95* | *48.22* | **47.93** | 77.83 | 60.70 | 71.93 | *50.89* | 52.32 | **49.48** | 79.63 | 37.80 | 82.24 | *49.43* | 55.17 | **49.21** | 81.39 |
| Cov$_{BCA}$ | 4.53 | 2.29 | 34.93 | 35.16 | 15.91 | **82.67** | 3.20 | 2.63 | 29.40 | 39.05 | 14.23 | **84.39** | 2.13 | 3.36 | 19.02 | 44.28 | 10.74 | **85.40** |
| *WIKI-31K, $k'=100$* | | | | | | | | | | | | | | | | | | |
| Top-K | **77.17** | 13.48 | 2.01 | 0.50 | 0.72 | 2.54 | **68.31** | 19.59 | 2.66 | 0.92 | 1.24 | 3.72 | **52.00** | 29.05 | 3.55 | 2.07 | 2.34 | 6.14 |
| PS-K | *67.95* | *11.89* | 3.89 | 1.47 | 1.93 | 5.47 | *62.65* | *18.07* | 4.21 | 2.14 | 2.56 | 6.54 | *50.16* | *28.20* | 4.64 | 3.62 | 3.61 | 9.01 |
| Pow-K $_{\beta=0.5}$ | 55.06 | 9.59 | 4.49 | 2.14 | 2.59 | 6.98 | 50.12 | 14.40 | 4.83 | 3.13 | 3.37 | 8.53 | 40.15 | 22.57 | 5.07 | 5.10 | 4.47 | 11.37 |
| Log-K | 65.20 | 11.34 | 3.40 | 1.24 | 1.66 | 4.84 | 57.74 | 16.49 | 3.74 | 1.86 | 2.25 | 5.99 | 43.82 | 24.37 | 4.24 | 3.46 | 3.37 | 8.61 |
| Macro-P$_{BCA}$ | 32.86 | 5.66 | **9.13** | 2.55 | 3.38 | 9.21 | 30.52 | 8.56 | **9.68** | 2.86 | 3.74 | 9.82 | 24.81 | 13.62 | **9.79** | 2.98 | 3.85 | 10.04 |
| Macro-R$_{BCA}$ | 13.78 | 2.36 | 4.77 | **3.24** | 3.10 | 7.26 | 14.01 | 3.96 | 5.79 | **4.71** | 4.05 | 10.72 | 13.88 | 7.79 | 5.72 | **7.21** | *5.02* | *15.72* |
| Macro-F1$_{BCA}$ | 37.32 | 6.52 | *7.81* | 3.15 | **4.10** | *10.42* | 38.98 | 11.24 | *8.30* | 4.14 | **5.08** | *11.88* | 39.31 | 22.28 | *8.18* | 5.11 | **5.78** | 13.07 |
| Cov$_{BCA}$ | 27.00 | 4.63 | 6.24 | *3.18* | *3.41* | **11.04** | 20.80 | 5.88 | 6.42 | *4.60* | *4.18* | **13.23** | 13.95 | 7.78 | 6.23 | *6.88* | 4.81 | **16.39** |
| *WIKIPEDIALARGE-500K, $k'=1000$* | | | | | | | | | | | | | | | | | | |
| Top-K | **56.02** | 45.70 | 20.15 | 18.87 | 17.14 | 32.24 | **43.12** | *54.28* | 20.51 | 25.86 | 19.84 | 40.62 | *27.01* | 63.13 | 17.16 | 34.79 | 19.57 | 50.15 |
| PS-K | *54.91* | 45.61 | 23.33 | 22.68 | 20.41 | 37.85 | *42.86* | **54.51** | 21.84 | 29.15 | 21.77 | 45.03 | **27.05** | 63.43 | 16.70 | 37.58 | 19.87 | 53.67 |
| Pow-K $_{\beta=0.5}$ | 51.81 | 43.94 | 23.73 | 23.67 | 21.13 | 39.36 | 40.55 | 52.72 | 21.87 | 30.03 | 22.18 | 46.21 | 26.09 | 62.26 | 16.48 | 38.17 | 19.85 | 54.38 |
| Log-K | 54.82 | 45.21 | 21.43 | 20.18 | 18.38 | 34.28 | 42.66 | 54.09 | 20.97 | 26.81 | 20.49 | 41.86 | 26.97 | *63.19* | 17.08 | 35.51 | 19.69 | 51.04 |
| Macro-P$_{BCA}$ | 25.19 | 21.81 | **37.69** | 20.18 | 23.44 | 45.08 | 16.25 | 22.54 | **37.88** | 21.12 | *24.10* | 46.23 | 8.48 | 22.97 | **37.77** | 21.46 | *24.19* | 46.65 |
| Macro-R$_{BCA}$ | 43.40 | 39.60 | 25.35 | **27.55** | *23.72* | 46.30 | 33.58 | 47.34 | 22.08 | **33.56** | 23.63 | *51.91* | 21.81 | 56.27 | 15.92 | **40.97** | 20.07 | *58.42* |
| Macro-F1$_{BCA}$ | 43.82 | 36.40 | *35.42* | 23.69 | **26.01** | *46.36* | 32.96 | 41.21 | *35.79* | 26.19 | **27.64** | 49.20 | 19.32 | 44.17 | *35.55* | 27.43 | **28.15** | 50.67 |
| Cov$_{BCA}$ | 27.31 | 24.55 | 25.92 | *26.76* | 21.60 | **50.16** | 19.46 | 28.06 | 23.14 | *32.13* | 21.08 | **55.40** | 11.59 | 32.09 | 18.45 | *38.56* | 18.08 | **61.15** |
| *AMAZON-670K, $k'=1000$* | | | | | | | | | | | | | | | | | | |
| Top-K | **41.71** | **24.08** | 10.77 | 9.68 | 9.51 | 14.35 | **37.71** | **35.23** | 14.26 | 15.16 | 13.76 | 20.41 | **25.35** | *46.74* | 14.83 | 21.56 | 16.20 | 26.85 |
| PS-K | 41.05 | 23.77 | 11.86 | 10.54 | 10.42 | 15.50 | 37.54 | 35.13 | 14.86 | 15.73 | 14.31 | 21.16 | 25.28 | 46.65 | 14.89 | 21.85 | 16.34 | 27.22 |
| Pow-K $_{\beta=0.5}$ | 40.90 | 23.71 | 11.99 | 10.65 | 10.53 | 15.64 | 37.47 | 35.07 | 14.98 | 15.84 | 14.42 | 21.30 | 25.24 | 46.60 | 14.90 | *21.92* | 16.36 | 27.30 |
| Log-K | *41.54* | *24.00* | 11.34 | 10.10 | 9.98 | 14.94 | *37.69* | *35.22* | 14.48 | 15.37 | 13.97 | 20.68 | *25.35* | **46.75** | 14.85 | 21.64 | 16.24 | 26.94 |
| Macro-P$_{BCA}$ | 33.79 | 19.75 | **17.27** | 10.53 | *12.12* | 17.75 | 27.52 | 26.09 | **21.09** | 14.38 | *15.96* | 21.96 | 15.02 | 28.23 | **23.21** | 16.36 | *17.90* | 24.13 |
| Macro-R$_{BCA}$ | 39.42 | 22.92 | 13.71 | **11.17** | 11.47 | 17.12 | 36.43 | 34.19 | 16.48 | **16.35** | 15.40 | 22.61 | 24.72 | 45.70 | 16.30 | **22.23** | 17.46 | *28.08* |
| Macro-F1$_{BCA}$ | 37.34 | 21.70 | *16.49* | 10.75 | **12.17** | 17.65 | 31.97 | 30.04 | *20.39* | 15.15 | **16.37** | 22.25 | 18.48 | 34.28 | *22.79* | 17.95 | **18.89** | 25.12 |
| Cov$_{BCA}$ | 35.38 | 20.32 | 14.04 | *10.85* | 11.23 | *17.70* | 30.27 | 28.39 | 16.44 | *15.94* | 14.85 | **22.93** | 20.08 | 37.23 | 16.20 | 21.52 | 16.61 | **28.15** |

marginal probabilities, the results might diverge from expectations in some cases. In Appendix E, we provide more details and conduct further experiments to investigate the impact of randomness, stopping conditions, quality of probability estimates, and shortlisting on the results. We also present similar experiments with probabilistic label trees used as an LPE.

Unfortunately, our results show that optimization of macro-measures comes with the cost of a significant drop in performance on instance-wise measures. Ideally, we would like to improve the performance on tail labels without sacrificing too much of general performance. To achieve such a trade-off, we can use straight-forward interpolation of instance-wise precision-at-k, as it is covered by our framework, and a selected macro-measure, since the considered class of utility functions is closed under linear combinations. Such an objective can be optimized by the proposed block-coordinate algorithm without any modification. As an example, we plot in Table 7 the results of optimizing a linear combination of instance-wise precision and the macro-F1 measure: $\Psi(\boldsymbol{Y}, \hat{\boldsymbol{Y}}) = (1 - \alpha)\Psi_{\text{Instance-P}}(\boldsymbol{Y}, \hat{\boldsymbol{Y}}) + \alpha\Psi_{\text{Macro-F1}}(\boldsymbol{Y}, \hat{\boldsymbol{Y}})$, using different values of $\alpha \in [0, 1]$. In Appendix E.5, we provide more plots for different utilities, datasets, and values of $k$. All the plots show that the instance-vs-macro curve has a nice concave shape that dominates simple baselines. In particular, the initial significant improvement on macro-measures comes with a minor drop in instance-measures, and only if one wants to optimize more strongly for macro-measures, the drop on instance-wise measures becomes more severe.

# 8 Discussion

The advantage of the ETU framework is that one can use multiple inference strategies without re-training a model. This is especially useful in cases where it is not a-priori clear which performance measure should be optimized, or predictions for different purposes are needed at the same time. On the other hand, as this framework optimizes the performance directly on a given test set, it is not designed to make predictions for single instances independently. The ETU framework has mainly been studied in the case of binary classification problems. We are not aware of any work that focuses on optimizing the complex performance metrics in the ETU framework for multi-label classification. One could try to generalize the results from binary classification, but the existing algorithms might not scale well to the extreme number of labels and they do not take the budget of $k$ labels into account.

Our paper gives novel and non-trivial results regarding this challenging optimization problem. We have thoroughly analyzed the ETU framework for a wide class of performance metrics, derived optimal prediction rules and constructed their computationally efficient approximations with provable regret guarantees and being robust against model misspecification. Our algorithm, based on block coordinate descent, scales effortlessly to XMLC problems and obtains promising empirical results.

Overall, we identified four categories of utilities, that differ in the complexity of the optimization algorithm—whether to use instance-wise optimization as in Section 4.2, or the block coordinate-ascent (Algorithm 1)— and the guarantees for the result—whether semi-empirical quantities (Section 4.1) lead to an optimal solution, or a suboptimal with error bounded by Theorem 5.2. These are given as follows and summarized in Table 3:

Table 3: Four different classes of utilities

| Linear in | Approx. | Algorithm |
|---|---|---|
| $t, q, p$ | No | Instance-wise |
| $t, q$ | Yes | Instance-wise |
| $t, p$ | No | Block-Coordinate |
| — | Yes | Block-Coordinate |

- **Fully linear**: Optimal predictions for metrics that are linear in all entries of the confusion matrix, as (6), can be solved *exactly* in an *instance-wise* manner. Examples are classical metrics such as instance-wise precision@$k$, or propensity-scored precision@$k$.

- **Linear in predictions**: *Approximately optimal* predictions for metrics that are linear in the predictions as given in (15) can be obtained using *instance-wise* optimization, by switching from $\Psi_{\mathrm{ETU}}$ to $\bar{\Psi}_{\mathrm{ETU}}$. An example is macro recall@$k$.

- **Linear in labels**: If a metric is linear in the label variables as given in (14), then $\tilde{\Psi}_{\mathrm{ETU}} \equiv \Psi_{\mathrm{ETU}}$. However, the resulting combinatorial optimization problem for $\tilde{\Psi}_{\mathrm{ETU}}$ is still complex enough, and we can solve it *only locally*. An example is macro precision@$k$.

- **Nonlinear metrics**: If none of the above apply, we have $\tilde{\Psi}_{\mathrm{ETU}} \neq \Psi_{\mathrm{ETU}}$, and have to solve it *locally* using *block-coordinate ascent*. This is the case of macro $F$-measure@$k$, or coverage@$k$.

The macro-averaged metrics budgeted at $k$ are attractive for measuring the long-tail performance. Macro-averaging treats all the labels as equally important, while the budget of $k$ predictions requires the prediction algorithm to choose the labels "wisely." We believe that this approach is substantially better than the one based on propensity-scored metrics. It is important to make the distinction between a metric used as a surrogate for training or inference, and its use as a performance metric in itself. While the former might be well justified for many metrics, the latter not necessarily as a metric might not have a clear interpretation of the calculated numbers.

Finally, the proposed framework is a plug-in approach that works on estimates of marginal probabilities and can be seamlessly applied to many existing state-of-the-art XMLC algorithms that are able to output such predictions, including XR-Transformer [51], CascadeXML [20], and algorithmic approaches for dealing with missing labels [40, 36]. It can also be combined with recently proposed methods that try to improve predictive performance on the tail labels by, e.g., leveraging labels features to estimate labels-correlations [29, 39, 9, 52] or to do data augmentation and generate new training points for tail labels [43, 21, 8].

## Acknowledgments

A part of computational experiments for this paper had been performed in Poznan Supercomputing and Networking Center. We want to acknowledge the support of Academy of Finland via grants 347707 and 348215, and also thank Mohammadreza Qaraei for providing pre-generated LPEs, $\hat{\eta}$, obtained by LightXML using computational resources of CSC – IT Center for Science, Finland.

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

# A  Order-invariant linearly decomposable task metrics

In this section, we provide a closer look at the family $\mathcal{L}_{\mathrm{OI}}$ of order-invariant task utilities that can be decomposed into a sum of label-specific functions as laid out in Section 2 in the main paper.

We first formalize the notion of instance-order invariance, and then show that this implies the possibility of reformulating the utility function based on the confusion matrix. We further prove that it is sufficient to know the *marginal* label probabilities $\boldsymbol{\eta}(\boldsymbol{x})$ for each instance $\boldsymbol{x}$ in order to evaluate any utility $\psi \in \mathcal{L}_{\mathrm{OI}}$ in the ETU-framework.

## A.1  Order-invariant task utilities as confusion-matrix metrics

We show in Theorem A.3 that any utility function $\Psi(\boldsymbol{Y}, \hat{\boldsymbol{Y}})$ that is invariant under instance reordering can be defined in terms of confusion matrices. In this way, we justify our choice in the main paper to focus on losses of the form given in (4).

More precisely, let $\sigma \in \mathfrak{S}(n)$ be a permutation of rows, that is, for $\boldsymbol{Y} = [\boldsymbol{y}_1, \ldots, \boldsymbol{y}_n]^{\mathsf{T}}$, we define $\sigma \boldsymbol{Y} = [\boldsymbol{y}_{\sigma(1)}, \ldots, \boldsymbol{y}_{\sigma(n)}]^{\mathsf{T}}$. Then we can make the following definition:

**Definition A.1** (Invariant under instance reordering). Let $n, m \in \mathbb{N}$ and $\Psi \colon \{0,1\}^{n \times m} \times \{0,1\}^{n \times m} \longrightarrow \mathbb{R}$ be a function of discrete labels and predictions. If $\Psi$ remains unchanged for every permutation of rows $\sigma \in \mathfrak{S}(n)$, i.e.,

$$\Psi(\boldsymbol{Y}, \hat{\boldsymbol{Y}}) = \Psi(\sigma \boldsymbol{Y}, \sigma \hat{\boldsymbol{Y}}), \tag{28}$$

then we call $\Psi$ a function that is *invariant under instance reordering*. We denote the set of instance-order-invariant losses with $m$ labels as

$$\mathcal{I}_m := \left\{ \Psi \colon \{0,1\}^{n \times m} \times \{0,1\}^{n \times m} \longrightarrow \mathbb{R} \colon \ \Psi \text{ is invariant under instance reordering} \right\}. \tag{29}$$

We further define the set of all possible confusion matrices with $n$ instances as

$$\mathcal{C}(n) := \left\{ \boldsymbol{C} \colon n\boldsymbol{C} \in \mathbb{N}^{2 \times 2}, \|\boldsymbol{C}\|_{1,1} = 1 \right\}, \tag{30}$$

where $\mathbb{N}$ is the set of natural numbers including 0. Now we are ready to provide a lemma for the binary case:

**Lemma A.2.** *Let* $\psi \colon \{0,1\}^n \times \{0,1\}^n \longrightarrow \mathbb{R}$ *be a binary loss function that is invariant under instances reordering. Then there exists a function* $\phi \colon \mathcal{C}(n) \longrightarrow \mathbb{R}$ *such that* $\psi = \phi \circ \boldsymbol{C}$.

*Proof.* We provide an explicit construction of $\psi$. To that end, let $\tilde{\boldsymbol{C}} \in \mathcal{C}(n)$ be one such confusion matrix, then there exists $(\boldsymbol{y}, \hat{\boldsymbol{y}}) \in \{0,1\}^n \times \{0,1\}^n$ given by

$$(\boldsymbol{y}, \hat{\boldsymbol{y}}) = [\underbrace{(1,1), \ldots, (1,1)}_{\times n \cdot \tilde{c}_{11}}, \underbrace{(0,1), \ldots, (0,1)}_{\times n \cdot \tilde{c}_{01}}, \underbrace{(1,0), \ldots, (1,0)}_{\times n \cdot \tilde{c}_{10}}, \underbrace{(0,0), \ldots, (0,0)}_{\times n \cdot \tilde{c}_{00}}]^{\mathsf{T}}. \tag{31}$$

Define $\phi$ such that $\phi(\tilde{\boldsymbol{C}}) = \psi(\boldsymbol{y}, \hat{\boldsymbol{y}})$.

Now, let $(\boldsymbol{y}', \hat{\boldsymbol{y}}') \in \{0,1\}^n \times \{0,1\}^n$ be an arbitrary label-prediction combination, with $\boldsymbol{C}(\boldsymbol{y}', \hat{\boldsymbol{y}}') = \tilde{\boldsymbol{C}}$. Then there exists a permutation $\sigma$ such that $\sigma \boldsymbol{y}' = \boldsymbol{y}$ and $\sigma \hat{\boldsymbol{y}}' = \hat{\boldsymbol{y}}$. By the invariance assumption, it holds that

$$\psi(\boldsymbol{y}', \hat{\boldsymbol{y}}') = \psi(\sigma \boldsymbol{y}', \sigma \hat{\boldsymbol{y}}') = \psi(\boldsymbol{y}, \hat{\boldsymbol{y}}) = \phi(\tilde{\boldsymbol{C}}) = \phi(\boldsymbol{C}(\boldsymbol{y}', \hat{\boldsymbol{y}}')). \tag{32}$$

As the original $\tilde{\boldsymbol{C}} \in \mathcal{C}(n)$ was arbitrary, the statement is shown. $\square$

We can now extend this lemma to show the equivalence of two definitions of the task losses considered in this paper:

**Theorem A.3** (Equivalence of order-invariance and confusion-matrix losses). *Let* $n, m \in \mathbb{N}$, *and* $\mathcal{Y} = \{0,1\}^m$. *Define the set of instance-order invariant, label-averaged losses as*

$$\mathcal{L}_{\mathrm{OI}} := \left\{ \Psi \in \mathcal{I}_m : (\boldsymbol{Y}, \hat{\boldsymbol{Y}}) \mapsto \sum_{j=1}^{m} \psi^j(\boldsymbol{y}_{:j}, \hat{\boldsymbol{y}}_{:j}) \right\}, \tag{33}$$

*and the set of confusion-matrix based, label-averaged losses as*

$$\mathcal{L}_{\mathrm{CM}} := \left\{ \Phi \colon \mathcal{C}(n)^m \longrightarrow \mathbb{R} \text{ s.t. } \mathbf{C} \mapsto \sum_{j=1}^{m} \phi^j(\mathbf{C}^j) \right\}. \tag{34}$$

*Then these two descriptions are equivalent, in the sense that*

$$\mathcal{L}_{\mathrm{OI}} = \left\{ \Phi \circ \mathbf{C} \colon \Phi \in \mathcal{L}_{\mathrm{CM}} \right\}, \tag{35}$$

*that is, every instance-order invariant loss can be written as a confusion-matrix loss, and every confusion-matrix loss leads to an instance-order invariant loss.*

*Proof.* As calculating the confusion matrix is in itself an operation that is invariant under instance reordering, each $\Phi \circ \mathbf{C}$ clearly is instance-order invariant.

On the other hand, let $\Psi(\mathbf{Y}, \hat{\mathbf{Y}}) = \sum_{j=1}^{m} \psi^j(\mathbf{y}_{:j}, \hat{\mathbf{y}}_{:j})$, then by Lemma A.2 there exist $\phi^1, \dots, \phi^m$ such that

$$\psi^j(\mathbf{y}_{:j}, \hat{\mathbf{y}}_{:j}) = \phi^j(\mathbf{C}(\mathbf{y}_{:j}, \hat{\mathbf{y}}_{:j})). \tag{36}$$

Summing up the individual $\phi^j$ gives $\Phi$ of the correct form. $\qquad\square$

## A.2 Sufficiency of label probability estimates

If $\Psi \in \mathcal{L}_{\mathrm{OI}}$ is an instance-order invariant decomposable task utility, then its ETU risk is a function of the predictions $\hat{\mathbf{Y}}$ and the *marginal* label probabilities $\boldsymbol{\eta}(\boldsymbol{x})$ as given by (10). This means that it is not required to estimate the $m^2 - 1$ values of the full joint probability distribution $\mathbb{P}[\mathbf{Y} \mid \boldsymbol{x}]$.

**Lemma A.4.** *Let $\Psi \in \mathcal{L}_{\mathrm{OI}}$ be an instance-order invariant decomposable task utility, and assume that the labels of* different *instances are independent, i.e., that $\mathbb{P}(\mathbf{Y}|\mathbf{X}) = \prod_{i=1}^{n} \mathbb{P}(\mathbf{y}_i|\boldsymbol{x}_i)$. Then we have*

$$\mathbb{E}_{\mathbf{Y}|\mathbf{X}}[\Psi(\mathbf{Y}, \hat{\mathbf{Y}})] = \sum_{j=1}^{m} \sum_{\mathbf{y}' \in \{0,1\}^n} \left( \prod_{i=1}^{n} \eta_j(\boldsymbol{x}_i) y_i' + (1 - \eta_j(\boldsymbol{x}_i))(1 - y_i') \right) \psi^j(\mathbf{y}', \hat{\mathbf{y}}_{:j}). \tag{37}$$

*Proof.*

$$\begin{aligned}
\mathbb{E}_{\mathbf{Y}|\mathbf{X}}[\Psi(\mathbf{Y}, \hat{\mathbf{Y}})] &= \sum_{j=1}^{m} \mathbb{E}_{\mathbf{Y}|\mathbf{X}}\left[ \psi^j(\mathbf{y}_{:j}, \hat{\mathbf{y}}_{:j}) \right] = \sum_{j=1}^{m} \sum_{\mathbf{y}' \in \{0,1\}^n} \mathbb{P}[\mathbf{y}_{:j} = \mathbf{y}' \mid \mathbf{X}] \psi^j(\mathbf{y}', \hat{\mathbf{y}}_{:j}) \\
&= \sum_{j=1}^{m} \sum_{\mathbf{y}' \in \{0,1\}^n} \left( \prod_{i=1}^{n} \mathbb{P}[Y_{ij} = y_i' \mid \boldsymbol{x}_i] \right) \psi^j(\mathbf{y}', \hat{\mathbf{y}}_{:j}) \\
&= \sum_{j=1}^{m} \sum_{\mathbf{y}' \in \{0,1\}^n} \left( \prod_{i=1}^{n} \eta_j(\boldsymbol{x}_i) y_i' + (1 - \eta_j(\boldsymbol{x}_i))(1 - y_i') \right) \psi^j(\mathbf{y}', \hat{\mathbf{y}}_{:j}). \tag{38}
\end{aligned}$$

$\square$

## A.3 Representation change from the instance-wise form to the cost-matrix form

If we start with the linear metric given in the instance-wise form according to (5), the corresponding representation according to (15) may be shifted by a constant value, and can be derived through the following:

$$\begin{aligned}
c_{11}w_{11} + c_{01}w_{01} + c_{10}w_{10} + c_{00}w_{00} &= c_{11}w_{11} + c_{01}w_{01} + (p - t)w_{10} + (1 - p - c_{01})w_{00} \\
&= t\,(w_{11} - w_{10}) + c_{01}\,(w_{01} - w_{00}) + pw_{10} + (1 - p)w_{00} \\
&= t\,(w_{11} - w_{10} - w_{01} + w_{00}) + q\,(w_{01} - w_{00}) + \mathrm{const}. \tag{39}
\end{aligned}$$

Thus, we get $f_{\mathrm{t}}(p) = w_{11} - w_{10} - w_{01} + w_{00}$ and $f_{\mathrm{q}}(p) = w_{01} - w_{00}$.

# B Block Coordinate Ascent

In this section, we provide additional analysis and variations of the block coordinate ascent algorithm. We first give an intuition into the gain-based selection of labels performed by the algorithm in the case of macro-precision. Then, we will illustrate how, for metrics that are linear in the prediction argument as per (15), the algorithm will recover the closed-form solution given in (19). Afterwards, we present an alternative formulation specifically for coverage, and conclude with greedy versions of the BCA algorithm.

## B.1 Example: Macro-Precision

As an example, consider the optimization of macro-precision. In this case, $\psi^j(t, q, p) = t/q$. Plugging into the gain formula gives

$$g_j = \frac{t^j + \hat{\eta}_j(\boldsymbol{x}_s)}{q^j + 1} - \frac{t^j}{q^j} = \frac{p^j(t^j + \hat{\eta}_j(\boldsymbol{x}_s)) - t^j(q^j + 1)}{q^j(q^j + 1)} = \frac{q^j \hat{\eta}_j(\boldsymbol{x}_s) - t^j}{q^j(q^j + 1)}. \tag{40}$$

This term is positive only if $t^j/q^j < \hat{\eta}_j(\boldsymbol{x}_s)$, i.e., predicting the label has a positive impact on the overall utility if its chance of being relevant, $\hat{\eta}_j(\boldsymbol{x}_s)$, is larger than the current estimate of precision for label $j$. Of course, the "at-$k$" constraint means that, for a given instance, it may not be possible to select all labels with positive gain, or necessary to select some labels with negative gain to fulfill the constraint.

## B.2 Global optimality for linear metrics

If $\psi^j$ is a linear function, i.e., it corresponds to an instance-wise measure as in (6), the gain $\boldsymbol{g}$, defined through lines 9-11 in Algorithm 1, becomes

$$\psi^j(t + \hat{\eta}_j(\boldsymbol{x}_s), q + 1, p) - \psi^j(t, q, p) = (t + \hat{\eta}_j(\boldsymbol{x}_s) - t)f_t(p) + (q + 1 - q)f_q(p)$$
$$= \hat{\eta}_j(\boldsymbol{x}_s)f_t(p) + f_q(p), \tag{41}$$

because the influence of the other instances cancels out. This exactly reproduces the gain formula from Section 4.2, which means that for each instance, the block coordinate-ascent algorithm will select the globally (approximately) optimal decision on the first pass. If the metric is fully linear, this will be the Bayes-optimal decision, otherwise, it is approximately optimal in the sense of Theorem 5.2. In both cases, the algorithm will perform a second pass over the entire dataset in order to check the stopping criterion. This second pass will leave all predictions unchanged.

## B.3 The ETU approach for coverage

In this section, we derive the ETU risk presented initially in Section 4.4. Then, we present a block coordinate ascent algorithm for optimizing it. The step-by-step derivation of (24) is given below:

$$\Psi_{\text{ETU}}(\hat{\boldsymbol{Y}}; \boldsymbol{X}) = \mathbb{E}_{\boldsymbol{Y}|\boldsymbol{X}}[\Psi_{\text{cov}}(\boldsymbol{t}(\boldsymbol{Y}, \hat{\boldsymbol{Y}}), \boldsymbol{q}(\boldsymbol{Y}, \hat{\boldsymbol{Y}}), \boldsymbol{p}(\boldsymbol{Y}, \hat{\boldsymbol{Y}}))]$$

$$= \mathbb{E}_{\boldsymbol{Y}|\boldsymbol{X}}\left[\frac{1}{m}\sum_{j=1}^{m} \mathbb{1}[t_j > 0]\right] = \mathbb{E}_{\boldsymbol{Y}|\boldsymbol{X}}\left[\frac{1}{m}\sum_{j=1}^{m}(1 - \mathbb{1}[t_j = 0])\right]$$

$$= \mathbb{E}_{\boldsymbol{Y}|\boldsymbol{X}}\left[\frac{1}{m}\sum_{j=1}^{m}\left(1 - \prod_{i=1}^{j}(1 - y_{ij}\hat{y}_{ij})\right)\right] = \mathbb{E}_{\boldsymbol{Y}|\boldsymbol{X}}\left[1 - \frac{1}{m}\sum_{j=1}^{m}\prod_{i=1}^{j}(1 - y_{ij}\hat{y}_{ij})\right]. \tag{42}$$

Because we assume labels for one instance to be independent on all other instances, i.e., $\mathbb{P}(\boldsymbol{Y}|\boldsymbol{X}) = \prod_{i=1}^{n} \mathbb{P}(\boldsymbol{y}_i|\boldsymbol{x}_i)$, we obtain:

$$\Psi_{\text{ETU}}(\hat{\boldsymbol{Y}}; \boldsymbol{X}) = 1 - \frac{1}{m}\sum_{j=1}^{m}\prod_{i=1}^{n}\left(1 - \mathbb{E}_{\boldsymbol{Y}|\boldsymbol{X}}[y_{ij}\hat{y}_{ij}]\right) = 1 - \frac{1}{m}\sum_{j=1}^{m}\prod_{i=1}^{n}(1 - \mathbb{P}[y_{ij} = 1|\boldsymbol{x}_i])$$

$$= 1 - \frac{1}{m}\sum_{j=1}^{m}\prod_{i=1}^{n}(1 - \eta_j(\boldsymbol{x}_i)\hat{y}_{ij}). \tag{43}$$

Based on this result we can construct a block coordinate ascent procedure which, for predictions being fixed for all instances except $\boldsymbol{x}_s$, optimizes the following problem:

$$\max_{\boldsymbol{z} \in \mathcal{Y}_k} \sum_{j=1}^m \psi_{\text{cov}}^j(z_j) = \sum_{j=1}^m \left( 1 - (1 - \eta_j(\boldsymbol{x}_s) z_j) \prod_{i \in [n] \setminus \{s\}} (1 - \eta_j(\boldsymbol{x}_i) \hat{y}_{ij}) \right) . \tag{44}$$

Analogously to Section 4.3, everything except $z_j \in \{0, 1\}$ is given and we can define a gain vector with elements $g_j = \psi_{\text{cov}}^j(1) - \psi_{\text{cov}}^j(0)$. Again the optimal prediction $\boldsymbol{z}^*$ is then given by $\boldsymbol{z}^* = \text{select-top-}k(\boldsymbol{g})$, and we get $\hat{\boldsymbol{Y}}^{t+1}$ by replacing the $s^{\text{th}}$ row of $\hat{\boldsymbol{Y}}^t$ with $\boldsymbol{z}^*$. Then we switch to the next instance $s \leftarrow s+1$, and repeat this process until no more progress is made.

Notice that $\prod_{i=1}^n (1 - \eta_j(\boldsymbol{x}_i) \hat{y}_{ij})$ corresponds to the probability of label $j$ being irrelevant for all instances it was selected for. We denote this value as $f_j$ and use it to speed up computations in each iteration of the algorithm:

---

**Algorithm 2** BCA for coverage$(\boldsymbol{X}, \hat{\boldsymbol{\eta}}, k, \epsilon)$

1: $\hat{\boldsymbol{y}}_i \leftarrow \text{select-random-}k(m)$ for all $i \in [n]$
2: $\boldsymbol{f} \leftarrow \prod_{i=1}^n (1 - \hat{\boldsymbol{\eta}}(\boldsymbol{x}_i) \odot \hat{\boldsymbol{y}}_i)$
3: $u_{\text{old}} \leftarrow -\infty$, $u_{\text{new}} \leftarrow 1 - (m^{-1} \sum_{j=1}^m f_j)$
4: **while** $u_{\text{new}} > u_{\text{old}} + \epsilon$ **do**
5:     **for** $s \in \text{shuffle}([n])$ **do**
6:         $\boldsymbol{f} \leftarrow \boldsymbol{f} \odot (1 - \hat{\boldsymbol{\eta}}(\boldsymbol{x}_s) \odot \hat{\boldsymbol{y}}_s)^{-1}$
7:         $\boldsymbol{g} \leftarrow \boldsymbol{f} \odot \hat{\boldsymbol{\eta}}(\boldsymbol{x}_s)$
8:         $\hat{\boldsymbol{y}}_s \leftarrow \text{select-top-}k(\boldsymbol{g})$
9:         $\boldsymbol{f} \leftarrow \boldsymbol{f} \odot (1 - \hat{\boldsymbol{\eta}}(\boldsymbol{x}_s) \odot \hat{\boldsymbol{y}}_s)$
10:    $u_{\text{old}} \leftarrow u_{\text{new}}$, $u_{\text{new}} \leftarrow 1 - (m^{-1} \sum_{j=1}^m f_j)$
11: **return** $\hat{\boldsymbol{Y}}$

---

$$f_j^0 := \prod_{i=1}^n \left( 1 - \eta_j(\boldsymbol{x}_i) \hat{y}_{ij}^0 \right) , \quad f_j^{t+1} := f_j^t \frac{1 - \eta_j(\boldsymbol{x}_s) \hat{y}_{sj}^{t+1}}{1 - \eta_j(\boldsymbol{x}_s) \hat{y}_{sj}^t} . \tag{45}$$

We can then compute the gain vector using the following formula:

$$g_j = \left( 1 - f_j^t \frac{1 - \eta_j(\boldsymbol{x}_s)}{1 - \eta_j(\boldsymbol{x}_s) \hat{y}_{sj}^t} \right) - \left( 1 - f_j^t \frac{1}{1 - \eta_j(\boldsymbol{x}_s) \hat{y}_{sj}^t} \right) = \frac{\eta_j(\boldsymbol{x}_s) f_j^t}{1 - \eta_j(\boldsymbol{x}_s) \hat{y}_{sj}^t} . \tag{46}$$

This block coordinate ascent procedure for coverage is presented as Algorithm 2.

### B.4 Greedy version of block coordinate-ascent

The inference strategy presented in Algorithm 1 requires multiple passes over the entire data. This might be computationally expensive and requires the whole dataset to be available at once. We, thus, propose a greedy version of the algorithm, that performs just a *single pass* over the dataset and makes decisions for each instance by taking into account only statistics of *previously seen* instances instead of all other instances in the dataset. This allows to apply this algorithm to the semi-online setting, where the algorithm observes and predicts for instances one by one, but does not receive immediate feedback for its decisions, in contrast to the usual online learning setting. The procedure is outlined as Algorithm 3, and its greedy variant as Algorithm 4. The results of this algorithm in comparison to the BCA approach are given later in Appendix E.

---

**Algorithm 3** Greedy $(\boldsymbol{X}, \hat{\boldsymbol{\eta}}, k)$

1: $\tilde{\boldsymbol{t}} \leftarrow \boldsymbol{0}, \boldsymbol{q} \leftarrow \boldsymbol{0}, \tilde{\boldsymbol{p}} \leftarrow \boldsymbol{0}$
2: **for** $i \in [n]$ **do**
3:    $\tilde{\boldsymbol{p}} \leftarrow \tilde{\boldsymbol{p}} + \frac{1}{n} \hat{\boldsymbol{\eta}}(\boldsymbol{x}_i)$
4:    **for** $j \in [m]$ **do**
5:       $\psi^j(1) \leftarrow \psi(\tilde{t}_j + \frac{1}{n} \hat{\eta}_j(\boldsymbol{x}_s), q_j + \frac{1}{n}, \tilde{p}_j)$
6:       $\psi^j(0) \leftarrow \psi(\tilde{t}_j, q_j, \tilde{p}_j)$
7:       $g_j \leftarrow \psi^j(1) - \psi^j(0)$
8:    $\hat{\boldsymbol{y}}_i \leftarrow \text{select top-}k(\boldsymbol{g})$
9:    $\tilde{\boldsymbol{t}} \leftarrow \tilde{\boldsymbol{t}} + \frac{1}{n} \hat{\boldsymbol{\eta}}(\boldsymbol{x}_i) \odot \hat{\boldsymbol{y}}_i$
10:   $\boldsymbol{q} \leftarrow \boldsymbol{q} + \frac{1}{n} \hat{\boldsymbol{y}}_i$
11: **return** $\hat{\boldsymbol{Y}}$

---

**Algorithm 4** Greedy for coverage $(\boldsymbol{X}, \hat{\boldsymbol{\eta}}, k)$

1: $\boldsymbol{f} \leftarrow \boldsymbol{1}$
2: **for** $i \in [n]$ **do**
3:    $\boldsymbol{g} \leftarrow \boldsymbol{f} \odot \hat{\boldsymbol{\eta}}(\boldsymbol{x}_i)$
4:    $\hat{\boldsymbol{y}}_i \leftarrow \text{select top-}k(\boldsymbol{g})$
5:    $\boldsymbol{f} \leftarrow \boldsymbol{f} \odot (1 - \hat{\boldsymbol{\eta}}(\boldsymbol{x}_i) \odot \hat{\boldsymbol{y}}_i)$
6: **return** $\hat{\boldsymbol{Y}}$

---

## C   Detailed Regret Analysis

In this section, we discuss the $p$-Lipschitzness condition of the metrics of interest, and prove Theorem 5.2 and Theorem 5.3.

### C.1   $p$-Lipschitz utility functions

First, recall the definition from the main paper:

**Definition 5.1** ($p$-Lipschitz [13]). *A binary classification metric $\psi(t, q, p)$ is said to be $p$-Lipschitz if*

$$|\psi(t, q, p) - \psi(t', q', p')| \leq L_{\mathrm{t}}(p)|t - t'| + L_{\mathrm{q}}(p)|q - q'| + L_{\mathrm{p}}(p)|p - p'|, \tag{25}$$

*for any $q, q' \in [0, 1]$, $p, p' \in (0, 1)$, $0 \leq t \leq \min(p, q)$, and $0 \leq t' \leq \min(p', q')$. The constants $L_{\mathrm{t}}(p), L_{\mathrm{q}}(p), L_{\mathrm{p}}(p)$ are allowed to depend on $p$, in contrast to the standard Lipschitz functions.*

The rationale behind this definition is that while we need to control the change in value of the metric under small changes in its arguments, a standard definition of Lipschitzness (with global constant) would not be satisfied by many popular metrics. For the same reason, we only require stability for *non-trivial* problems, that is, in cases where the rate of positives $p$ is neither zero nor one.

Relaxing the definition to allow the constants vary as a function of $p$ suffices to prove our stability results as well as regret bounds, while it is satisfied by most of the metrics of interest, as shown below. The notable exception not present in Table 4 is the *precision* metric, which is not $p$-Lipschitz due to its behavior for $q \to 0$.

Table 4: Examples of $p$-Lipschitz metrics. We use tp, fp, fn, and tn to denote true positives, false positives, false negatives, and true negatives.

| Metric | Definition | $\psi(t, q, p)$ |
|--------|-----------|-----------------|
| Accuracy | $\mathrm{tp} + \mathrm{tn}$ | $1 + 2t - q - p$ |
| Recall | $\frac{\mathrm{tp}}{\mathrm{tp+fn}}$ | $\frac{t}{p}$ |
| Bal. Acc. | $\frac{\mathrm{tp}/2}{\mathrm{tp+fn}} + \frac{\mathrm{tn}/2}{\mathrm{tn+fp}}$ | $\frac{t + p(1-q-p)}{2p(1-p)}$ |
| $F_\beta$ | $\frac{(1+\beta^2)\mathrm{tp}}{(1+\beta^2)\mathrm{tp}+\beta^2\mathrm{fn+fp}}$ | $\frac{(1+\beta^2)t}{\beta^2 p + q}$ |
| Jaccard | $\frac{\mathrm{tp}}{\mathrm{tp+fp+fn}}$ | $\frac{p+q-2t}{p+q-t}$ |
| G-Mean | $\sqrt{\frac{\mathrm{tp\cdot tn}}{(\mathrm{tp+fn})(\mathrm{tn+fp})}}$ | $\frac{t(1-q-p+t)}{p(1-p)}$ |
| AUC | $\frac{\mathrm{fp\cdot fn}}{(\mathrm{tp+fn})(\mathrm{fp+tn})}$ | $\frac{(q-t)(p-t)}{p(1-p)}$ |

**Lemma C.1.** *The linear confusion-matrix measures defined by (6):*

$$\Psi(\boldsymbol{Y}, \hat{\boldsymbol{Y}}) = \sum_{j=1}^{m} \left( w_{00}^j c_{00}^j + w_{01}^j c_{01}^j + w_{10}^j c_{10}^j + w_{11}^j c_{11}^j \right) \tag{47}$$

*with fixed coefficient matrices $\{\boldsymbol{W}^j\}_{j=1}^m$ are decomposable functions with p-Lipschitz components.*

*Proof.* The metric can be rewritten in a decomposable form $\Psi = \sum_j \psi^j$, where each $\psi^j$ in the $(t, q, p)$-parameterization has the following form:

$$\psi^j(t, q, p) = T_j \cdot t + Q_j \cdot q + P_j \cdot p + C_j \tag{48}$$

where $T_j, Q_j, P_j, C_j$ are some combinations of the coefficient matrices $\{\boldsymbol{W}^j\}_{j=1}^m$. Being a linear function of $t, q, p$, $\psi^j$ is Lipschitz. $\qquad\square$

**Lemma C.2** ($p$-Lipschitzness of common utilities [13]). *All metrics in Table 4 are p-Lipschitz.*

*Proof.* Here, we only prove the lemma for recall, which has not been covered by Proposition 1 of Dembczyński et al. [13].

$$|\psi(t, q, p) - \psi(t', q', p')| = \left| \frac{t}{p} - \frac{t'}{p'} \right| = \left| \frac{t - t'}{p} + \frac{t'}{p'} \frac{p' - p}{p} \right|$$

$$\leq \underbrace{\frac{1}{p}}_{=L_{\mathrm{t}}(p)} |t - t'| + \underbrace{\frac{t'}{p'}}_{\leq 1} \cdot \underbrace{\frac{1}{p}}_{=L_{\mathrm{p}}(p)} |p - p'|. \tag{49}$$

$$\square$$

## C.2 Stability of the semi-ETU approximation

We are now ready to prove that when the metric of interest has $p$-Lipschitz components, the semi-ETU approximation $\tilde{\Psi}_{\text{ETU}}$ presented in Section 4.1 remains close to the true objective $\Psi_{\text{ETU}}$. For the sake of convenience, let us recall the definition of the ETU objective:

$$\Psi_{\text{ETU}}(\hat{\boldsymbol{Y}}; \boldsymbol{X}) = \mathbb{E}_{\boldsymbol{Y}|\boldsymbol{X}}[\Psi(\boldsymbol{Y}, \hat{\boldsymbol{Y}})] = \sum_{j=1}^{m} \mathbb{E}_{\boldsymbol{Y}|\boldsymbol{X}}\left[\psi^j(t(\boldsymbol{y}_{:j}, \hat{\boldsymbol{y}}_{:j}), q(\hat{\boldsymbol{y}}_{:j}), p(\boldsymbol{y}_{:j}))\right], \quad (50)$$

as well as its approximation:

$$\tilde{\Psi}_{\text{ETU}}(\hat{\boldsymbol{Y}}; \boldsymbol{X}) = \sum_{j=1}^{m} \psi^j\left(\mathbb{E}_{\boldsymbol{Y}|\boldsymbol{X}}[t(\boldsymbol{y}_{:j}, \hat{\boldsymbol{y}}_{:j})], q(\hat{\boldsymbol{y}}_{:j}), \mathbb{E}_{\boldsymbol{Y}|\boldsymbol{X}}[p(\boldsymbol{y}_{:j})]\right). \quad (51)$$

We prove the following result:

**Theorem 5.2.** *Let each $\psi^j$ be $p$-Lipschitz with constants $L_t^j(p), L_q^j(p), L_p^j(p)$. For any $\hat{\boldsymbol{Y}}$ it holds:*

$$|\Psi_{\text{ETU}}(\hat{\boldsymbol{Y}}; \boldsymbol{X}) - \tilde{\Psi}_{\text{ETU}}(\hat{\boldsymbol{Y}}; \boldsymbol{X})| \leq \frac{1}{2\sqrt{n}}\left(\sum_{j=1}^{m}(L_t^j(\tilde{p}_j) + L_p^j(\tilde{p}_j))\right). \quad (26)$$

*Proof.* For the sake of the analysis, denote the Lipschitz constants as $L_t^j := L_t^j(\tilde{p}_j)$ and $L_p^j := L_p^j(\tilde{p}_j)$. Using definitions (50) and (51) and applying Jensen's inequality, we have

$$|\Psi_{\text{ETU}}(\hat{\boldsymbol{Y}}; \boldsymbol{X}) - \tilde{\Psi}_{\text{ETU}}(\hat{\boldsymbol{Y}}; \boldsymbol{X})| = \left|\sum_{j=1}^{m}\left(\mathbb{E}_{\boldsymbol{Y}|\boldsymbol{X}}\left[\psi^j(t_j, q_j, p_j)\right] - \psi^j(\tilde{t}_j, q_j, \tilde{p}_j)\right)\right|$$

$$\leq \sum_{j=1}^{m}\mathbb{E}_{\boldsymbol{Y}|\boldsymbol{X}}\left[\left|\psi^j(t_j, q_j, p_j) - \psi^j(\tilde{t}_j, q_j, \tilde{p}_j)\right|\right], \quad (52)$$

We now bound each term in the sum by $(L_t^j + L_p^j)/(2\sqrt{n})$, which will prove the theorem.

For each $j \in [m]$, using $p$-Lipschitzness of $\psi_j$ we have:

$$\left|\psi^j(t_j, q_j, p_j) - \psi^j(\tilde{t}_j, q_j, \tilde{p}_j)\right| = \left|\psi^j(\tilde{t}_j, q_j, \tilde{p}_j) - \psi^j(t_j, q_j, p_j)\right|$$

$$\leq L_t^j|t_j - \tilde{t}_j| + L_p^j|p_j - \tilde{p}_j|. \quad (53)$$

Taking expectation on both sides gives:

$$\mathbb{E}_{\boldsymbol{Y}|\boldsymbol{X}}\left[\left|\psi^j(t_j, q_j, p_j) - \psi^j(\tilde{t}_j, q_j, \tilde{p}_j)\right|\right] \leq L_t^j\,\mathbb{E}_{\boldsymbol{Y}|\boldsymbol{X}}[|t_j - \tilde{t}_j|] + L_p^j\,\mathbb{E}_{\boldsymbol{Y}|\boldsymbol{X}}[|p_j - \tilde{p}_j|]$$

$$= L_t^j\,\mathbb{E}_{\boldsymbol{Y}|\boldsymbol{X}}\left[\sqrt{(t_j - \tilde{t}_j)^2}\right] + L_p^j\,\mathbb{E}_{\boldsymbol{Y}|\boldsymbol{X}}\left[\sqrt{(p_j - \tilde{p}_j)^2}\right]$$

$$\leq L_t^j\sqrt{\mathbb{E}_{\boldsymbol{Y}|\boldsymbol{X}}[(t_j - \tilde{t}_j)^2]} + L_p^j\sqrt{\mathbb{E}_{\boldsymbol{Y}|\boldsymbol{X}}[(p_j - \tilde{p}_j)^2]}, \quad (54)$$

where the last inequality follows from Jensen's inequality applied to a concave function $x \mapsto \sqrt{x}$. Using the fact that $\tilde{t}_j = \mathbb{E}_{\boldsymbol{Y}|\boldsymbol{X}}[t_j]$ and $\tilde{p}_j = \mathbb{E}_{\boldsymbol{Y}|\boldsymbol{X}}[p_j]$, we have

$$\mathbb{E}_{\boldsymbol{Y}|\boldsymbol{X}}\left[(t_j - \tilde{t}_j)^2\right] = \text{Var}_{\boldsymbol{Y}|\boldsymbol{X}}(t_j) \leq \frac{1}{4n}, \quad (55)$$

as $t_j = n^{-1}\sum_{i=1}^{n} y_{ij}\hat{y}_{ij}$ is an average of $n$ Bernoulli i.i.d. random variables $y_{ij}\hat{y}_{ij}$, each having variance at most $\frac{1}{4}$; and using the same argument, $\mathbb{E}_{\boldsymbol{Y}|\boldsymbol{X}}\left[(p_j - \tilde{p}_j)^2\right] \leq \frac{1}{4n}$. This gives

$$\mathbb{E}_{\boldsymbol{Y}|\boldsymbol{X}}\left[\left|\psi^j(\tilde{t}_j, q_j, \tilde{p}_j) - \psi^j(t_j, q_j, p_j)\right|\right] \leq \frac{L_t^j + L_p^j}{2\sqrt{n}}, \quad (56)$$

and finishes the proof. $\qquad\square$

## C.3 Regret of semi-ETU under model misspecification

In this section, we quantify the influence of the estimation error of marginal probabilities, proving Theorem 5.3.

**Notation** To emphasize the dependence of $\Psi_{\text{ETU}}$ on the label probability estimates, in this section we will write

$$\Psi_{\text{ETU}}(\hat{\boldsymbol{Y}}; \boldsymbol{X}) = \mathbb{E}_{\boldsymbol{Y} \sim \boldsymbol{\eta}(\boldsymbol{X})}[\Psi(\boldsymbol{Y}, \hat{\boldsymbol{Y}})] =: \Psi_{\text{ETU}}(\hat{\boldsymbol{Y}}; \boldsymbol{\eta}) \,, \tag{57}$$

This notation is well-defined, as we have shown in Appendix A.2 that in fact, the dependence on $\boldsymbol{X}$ is mediated *only* through the marginal label probabilities $\boldsymbol{\eta}(\boldsymbol{X}) = (\boldsymbol{\eta}(\boldsymbol{x}_1), \dots, \boldsymbol{\eta}(\boldsymbol{x}_n))$, and we abbreviate $\boldsymbol{\eta}(\boldsymbol{X})$ as $\boldsymbol{\eta}$ . Similarly, we will write

$$\tilde{t}_j(\boldsymbol{\eta}) = \mathbb{E}_{\boldsymbol{y}_{:j} \sim \eta_j(\boldsymbol{X})}[t(\boldsymbol{y}_{:j}, \hat{\boldsymbol{y}}_{:j})] = n^{-1} \sum_{i=1}^{n} \eta_j(\boldsymbol{x}_i) \hat{y}_{ij} \,,$$

$$\tilde{p}_j(\boldsymbol{\eta}) = \mathbb{E}_{\boldsymbol{y}_{:j} \sim \eta_j(\boldsymbol{X})}[p(\boldsymbol{y}_{:j})] = n^{-1} \sum_{i=1}^{n} \eta_j(\boldsymbol{x}_i) \,. \tag{58}$$

Note that $q$ is independent of $\boldsymbol{\eta}$. This allows us to write the semi-ETU objective as

$$\tilde{\Psi}_{\text{ETU}}(\hat{\boldsymbol{Y}}; \boldsymbol{\eta}) := \Psi(\tilde{\boldsymbol{t}}, \boldsymbol{q}, \tilde{\boldsymbol{p}}) \,, \tag{59}$$

where we dropped the dependence of $\tilde{\boldsymbol{t}}$ and $\tilde{\boldsymbol{p}}$ on $\boldsymbol{\eta}$ as clear from the context. The optimal (Bayes) predictor $\hat{\boldsymbol{Y}}^\star$ is the one which maximizes the expected utility with regard to the *true* label probabilities. In the notation of this chapter, (1) reads

$$\hat{\boldsymbol{Y}}^\star = \operatorname*{argmax}_{\hat{\boldsymbol{Y}} \in \hat{\mathcal{Y}}_k^n} \Psi_{\text{ETU}}(\hat{\boldsymbol{Y}}; \boldsymbol{\eta}) \,. \tag{60}$$

Unfortunately, the learning algorithm does not know the true label marginals $\boldsymbol{\eta}(\boldsymbol{X})$, but has only access to the estimates $\hat{\boldsymbol{\eta}}(\boldsymbol{X}) = (\hat{\boldsymbol{\eta}}(\boldsymbol{x}_1), \dots, \hat{\boldsymbol{\eta}}(\boldsymbol{x}_n))$ for the considered set of instances. The algorithm computes its predictions $\hat{\boldsymbol{Y}}^\dagger$ by using the estimates in place of the true marginals. Thus, it can only generate

$$\hat{\boldsymbol{Y}}^\dagger = \operatorname*{argmax}_{\hat{\boldsymbol{Y}} \in \hat{\mathcal{Y}}_k^n} \Psi_{\text{ETU}}(\hat{\boldsymbol{Y}}; \hat{\boldsymbol{\eta}}) \,. \tag{61}$$

We can make the same definitions also for the semi-empirical ETU optimization, leading to

$$\tilde{\boldsymbol{Y}}^\star = \operatorname*{argmax}_{\hat{\boldsymbol{Y}} \in \hat{\mathcal{Y}}_k^n} \tilde{\Psi}_{\text{ETU}}(\hat{\boldsymbol{Y}}; \boldsymbol{\eta}) \quad \text{and} \quad \tilde{\boldsymbol{Y}}^\dagger = \operatorname*{argmax}_{\hat{\boldsymbol{Y}} \in \hat{\mathcal{Y}}_k^n} \tilde{\Psi}_{\text{ETU}}(\hat{\boldsymbol{Y}}; \hat{\boldsymbol{\eta}}) \,. \tag{62}$$

**Regret for semi-ETU approximation** First, we show that for metrics with $p$-Lipschitz components, the $\Psi$-*regret* of the resulting semi-ETU predictor (62), which is the suboptimality of $\tilde{\boldsymbol{Y}}^\dagger$ (with respect to $\hat{\boldsymbol{Y}}^\star$) in terms of $\Psi_{\text{ETU}}$, is well-controlled and upper-bounded by the estimation error of the marginals. As the resulting expression is somewhat unwieldy, we then apply some further bounding to arrive at the much simpler result stated in the main paper in Theorem 5.3.

**Lemma C.3** (Misspecification for semi-ETU approximation)**.** *Let* $\Psi \in \mathcal{L}_{\text{OI}}$ *be an instance-order invariant linearly decomposable loss function that is $p$-Lipschitz, and* $\hat{\boldsymbol{Y}}$ *an arbitrary set of predictions. Then, the difference of the semi-empirical ETU risk when using two different versions of the marginals, $\boldsymbol{\eta}$ and $\boldsymbol{\eta}'$, is bounded by the difference $\boldsymbol{\eta} - \boldsymbol{\eta}'$ through*

$$|\tilde{\Psi}_{\text{ETU}}(\hat{\boldsymbol{Y}}; \boldsymbol{\eta}) - \tilde{\Psi}_{\text{ETU}}(\hat{\boldsymbol{Y}}; \boldsymbol{\eta}')| \le n^{-1} \sum_{j=1}^{m} \left( L_{\text{t}}^j(\hat{t}^j(\boldsymbol{\eta})) + L_{\text{p}}^j(\hat{p}^j(\boldsymbol{\eta})) \right) \sum_{i=1}^{n} |\eta_j(\boldsymbol{x}_i) - {\eta_j}'(\boldsymbol{x}_i)| \,. \tag{63}$$

*Proof.* Plugging in the definitions, we have

$$|\tilde{\Psi}_{\text{ETU}}(\hat{\boldsymbol{Y}}; \boldsymbol{\eta}) - \tilde{\Psi}_{\text{ETU}}(\hat{\boldsymbol{Y}}; \boldsymbol{\eta})| = \left| \sum_{j=1}^{m} \psi^j\left(\tilde{t}^j(\boldsymbol{\eta}), q^j, \tilde{p}^j(\boldsymbol{\eta})\right) - \sum_{j=1}^{m} \psi^j\left(\tilde{t}^j(\boldsymbol{\eta}'), q^j, \tilde{p}^j(\boldsymbol{\eta}')\right) \right|$$

$$\le \sum_{j=1}^{m} \left| \psi^j\left(\tilde{t}^j(\boldsymbol{\eta}), q^j, \tilde{p}^j(\boldsymbol{\eta})\right) - \psi^j\left(\tilde{t}^j(\boldsymbol{\eta}'), q^j, \tilde{p}^j(\boldsymbol{\eta}')\right) \right|$$

$$\le \sum_{j=1}^{m} L_{\text{t}}^j(\tilde{t}^j(\boldsymbol{\eta})) \left| \tilde{t}^j(\boldsymbol{\eta}) - \tilde{t}^j(\boldsymbol{\eta}') \right| + L_{\text{p}}^j(\tilde{p}^j(\boldsymbol{\eta})) \left| \tilde{p}^j(\boldsymbol{\eta}) - \tilde{p}^j(\boldsymbol{\eta}') \right| \,, \tag{64}$$

where the last line used the definition of $p$-Lipschitzness.

The terms under the sum can be bounded by the estimation error of $\boldsymbol{\eta}'$:

$$\left|\tilde{t}_j(\boldsymbol{\eta}) - \tilde{t}_j(\boldsymbol{\eta}')\right| = \left|n^{-1}\sum_{i=1}^{n}\hat{y}_{ij}(\eta_j(\boldsymbol{x}_i) - \eta_j'(\boldsymbol{x}_i))\right| \leq n^{-1}\sum_{i=1}^{n}|\eta_j(\boldsymbol{x}_i) - \eta_j'(\boldsymbol{x}_i)|\,,$$

$$\left|\tilde{p}_j(\boldsymbol{\eta}) - \tilde{p}_j(\boldsymbol{\eta}')\right| = \left|n^{-1}\sum_{i=1}^{n}(\eta_j(\boldsymbol{x}_i) - \eta_j'(\boldsymbol{x}_i))\right| \leq n^{-1}\sum_{i=1}^{n}|\eta_j(\boldsymbol{x}_i) - \eta_j'(\boldsymbol{x}_i)|\,, \qquad (65)$$

where in the first line we used $\hat{y}_{ij} \in \{0, 1\}$. $\qquad\square$

**Lemma C.4** (Regret bound for semi-ETU approximation). *Let $\Psi \in \mathcal{L}_{\mathrm{OI}}$ be an instance-order invariant linearly decomposable loss function that is p-Lipschitz. Then we have*

$$\Psi_{\mathrm{ETU}}(\hat{\boldsymbol{Y}}^\star; \boldsymbol{X}) - \Psi_{\mathrm{ETU}}(\tilde{\boldsymbol{Y}}^\dagger; \boldsymbol{X}) \leq \frac{1}{\sqrt{n}}\sum_{j=1}^{m}\left(L_{\mathfrak{t}}^j(\tilde{p}_j(\boldsymbol{\eta})) + L_{\mathfrak{p}}^j(\tilde{p}_j(\boldsymbol{\eta}))\right) +$$

$$2\sum_{j=1}^{m}\frac{L_{\mathfrak{t}}^j(\tilde{p}_j(\boldsymbol{\eta})) + L_{\mathfrak{p}}^j(p_j(\boldsymbol{\eta}))}{n}\sum_{i=1}^{n}|\eta_j(\boldsymbol{x}_i) - \hat{\eta}_j(\boldsymbol{x}_i)|\,. \quad (66)$$

*Proof.* Using the optimality of $\tilde{\boldsymbol{Y}}^\dagger$ and a supremum bound, we get

$$\begin{aligned}\Psi_{\mathrm{ETU}}(\hat{\boldsymbol{Y}}^\star; \boldsymbol{\eta}) - \Psi_{\mathrm{ETU}}(\tilde{\boldsymbol{Y}}^\dagger; \boldsymbol{\eta}) &= \Psi_{\mathrm{ETU}}(\hat{\boldsymbol{Y}}^\star; \boldsymbol{\eta}) - \tilde{\Psi}_{\mathrm{ETU}}(\hat{\boldsymbol{Y}}^\star; \hat{\boldsymbol{\eta}}) \\ &\quad + \tilde{\Psi}_{\mathrm{ETU}}(\hat{\boldsymbol{Y}}^\star; \hat{\boldsymbol{\eta}}) - \tilde{\Psi}_{\mathrm{ETU}}(\tilde{\boldsymbol{Y}}^\dagger; \hat{\boldsymbol{\eta}}) \\ &\quad + \tilde{\Psi}_{\mathrm{ETU}}(\tilde{\boldsymbol{Y}}^\dagger; \hat{\boldsymbol{\eta}}) - \Psi_{\mathrm{ETU}}(\tilde{\boldsymbol{Y}}^\dagger; \boldsymbol{\eta}) \\ &\leq \Psi_{\mathrm{ETU}}(\hat{\boldsymbol{Y}}^\star; \boldsymbol{\eta}) - \tilde{\Psi}_{\mathrm{ETU}}(\hat{\boldsymbol{Y}}^\star; \hat{\boldsymbol{\eta}}) \qquad (67) \\ &\quad + \tilde{\Psi}_{\mathrm{ETU}}(\tilde{\boldsymbol{Y}}^\dagger; \hat{\boldsymbol{\eta}}) - \Psi_{\mathrm{ETU}}(\tilde{\boldsymbol{Y}}^\dagger; \boldsymbol{\eta}) \\ &\leq 2\sup_{\hat{\boldsymbol{Y}}}|\Psi_{\mathrm{ETU}}(\hat{\boldsymbol{Y}}; \boldsymbol{\eta}) - \tilde{\Psi}_{\mathrm{ETU}}(\hat{\boldsymbol{Y}}; \hat{\boldsymbol{\eta}})|\,. \qquad (68)\end{aligned}$$

We then use Theorem 5.2 and Lemma C.3 to bound

$$\begin{aligned}&\left|\Psi_{\mathrm{ETU}}(\hat{\boldsymbol{Y}}; \boldsymbol{\eta}) - \tilde{\Psi}_{\mathrm{ETU}}(\hat{\boldsymbol{Y}}; \hat{\boldsymbol{\eta}})\right| \\ &= |\Psi_{\mathrm{ETU}}(\hat{\boldsymbol{Y}}; \boldsymbol{\eta}) - \tilde{\Psi}_{\mathrm{ETU}}(\hat{\boldsymbol{Y}}; \boldsymbol{\eta}) + \tilde{\Psi}_{\mathrm{ETU}}(\hat{\boldsymbol{Y}}; \boldsymbol{\eta}) - \tilde{\Psi}_{\mathrm{ETU}}(\hat{\boldsymbol{Y}}; \hat{\boldsymbol{\eta}})| \\ &\leq |\Psi_{\mathrm{ETU}}(\hat{\boldsymbol{Y}}; \boldsymbol{\eta}) - \tilde{\Psi}_{\mathrm{ETU}}(\hat{\boldsymbol{Y}}; \boldsymbol{\eta})| + |\tilde{\Psi}_{\mathrm{ETU}}(\hat{\boldsymbol{Y}}; \boldsymbol{\eta}) - \tilde{\Psi}_{\mathrm{ETU}}(\hat{\boldsymbol{Y}}; \hat{\boldsymbol{\eta}})| \\ &\leq \frac{1}{2\sqrt{n}}\left(\sum_{j=1}^{m}(L_{\mathfrak{t}}^j(\tilde{p}_j) + L_{\mathfrak{p}}^j(\tilde{p}_j))\right) + \sum_{j=1}^{m}\frac{L_{\mathfrak{t}}^j(\tilde{p}_j) + L_{\mathfrak{p}}^j(\tilde{p}_j)}{n}\sum_{i=1}^{n}|\eta_j(\boldsymbol{x}_i) - \hat{\eta}_j(\boldsymbol{x}_i)|\,, \quad (69)\end{aligned}$$

where we use the shorthand $\tilde{p}_j = \tilde{p}_j(\boldsymbol{\eta})$. $\qquad\square$

At the cost of having a less strict bound, we can simplify this to

**Theorem 5.3.** *Let $\tilde{\boldsymbol{Y}}^\dagger$ be defined as above. Under the assumptions of Theorem 5.2:*

$$\Psi_{\mathrm{ETU}}(\hat{\boldsymbol{Y}}^\star; \boldsymbol{X}) - \Psi_{\mathrm{ETU}}(\tilde{\boldsymbol{Y}}^\dagger; \boldsymbol{X}) \leq \frac{m}{\sqrt{n}}B + 2\frac{\sqrt{m}}{n}B\sum_{i=1}^{n}\|\boldsymbol{\eta}(\boldsymbol{x}_i) - \hat{\boldsymbol{\eta}}(\boldsymbol{x}_i)\|_2\,, \qquad (27)$$

*where $B := \sqrt{m^{-1}\sum_{j=1}^{m}(L_{\mathfrak{t}}^j(\tilde{p}_j) + L_{\mathfrak{p}}^j(p_j))^2}$ is the quadratic mean of the Lipschitz constants.*

*Proof.* Using Cauchy-Schwartz, we can bound

$$\sum_{j=1}^{m} \frac{L_{\mathrm{t}}^{j}(\tilde{p}_j(\boldsymbol{\eta})) + L_{\mathrm{p}}^{j}(p_j(\boldsymbol{\eta}))}{n} \sum_{i=1}^{n} |\eta_j(\boldsymbol{x}_i) - \hat{\eta}_j(\boldsymbol{x}_i)|$$

$$\leq n^{-1} \sum_{i=1}^{n} \sqrt{\sum_{j=1}^{m} \big(L_{\mathrm{t}}^{j}(\tilde{p}_j(\boldsymbol{\eta})) + L_{\mathrm{p}}^{j}(p_j(\boldsymbol{\eta}))\big)^2} \cdot \sqrt{\sum_{j=1}^{m} (\eta_j(\boldsymbol{x}_i) - \hat{\eta}_j(\boldsymbol{x}_i))^2}$$

$$= n^{-1} \sqrt{\sum_{j=1}^{m} \big(L_{\mathrm{t}}^{j}(\tilde{p}_j(\boldsymbol{\eta})) + L_{\mathrm{p}}^{j}(p_j(\boldsymbol{\eta}))\big)^2} \cdot \sum_{i=1}^{n} \sqrt{\|\boldsymbol{\eta}(\boldsymbol{x}_i) - \hat{\boldsymbol{\eta}}(\boldsymbol{x}_i)\|_2^2}$$

$$= \frac{\sqrt{m}}{n} B \sum_{i=1}^{n} \|\boldsymbol{\eta}(\boldsymbol{x}_i) - \hat{\boldsymbol{\eta}}(\boldsymbol{x}_i)\|_2 \,. \tag{70}$$

For the other term, we can use the inequality between arithmetic and quadratic mean, so that

$$\sum_{j=1}^{m} \big(L_{\mathrm{t}}^{j}(\tilde{p}_j(\boldsymbol{\eta})) + L_{\mathrm{p}}^{j}(\tilde{p}_j(\boldsymbol{\eta}))\big) \leq m \sqrt{m^{-1} \sum_{j=1}^{m} \big(L_{\mathrm{t}}^{j}(\tilde{p}_j(\boldsymbol{\eta})) + L_{\mathrm{p}}^{j}(\tilde{p}_j(\boldsymbol{\eta}))\big)^2} = mB \,. \tag{71}$$

$\square$

## C.4 Regret for non-approximated ETU

We can formulate the equivalent of Lemma C.4 also for the maximizer of the true empirical ETU risk, $\hat{\boldsymbol{Y}}^{\dagger}$:

**Lemma C.5** (Regret bound for ETU maximization). *Let $\Psi \in \mathcal{L}_{\mathrm{OI}}$ be an instance-order invariant linearly decomposable loss function that is $p$-Lipschitz. Then we have*

$$\Psi_{\mathrm{ETU}}(\hat{\boldsymbol{Y}}^{\star}; \boldsymbol{X}) - \Psi_{\mathrm{ETU}}\big(\hat{\boldsymbol{Y}}^{\dagger}; \boldsymbol{X}\big) \leq \frac{1}{\sqrt{n}} \sum_{j=1}^{m} \big(L_{\mathrm{t}}^{j}(\tilde{p}_j(\boldsymbol{\eta})) + L_{\mathrm{t}}^{j}(\tilde{p}_j(\hat{\boldsymbol{\eta}})) + L_{\mathrm{p}}^{j}(\tilde{p}_j(\boldsymbol{\eta})) + L_{\mathrm{p}}^{j}(\tilde{p}_j(\hat{\boldsymbol{\eta}}))\big) +$$

$$2 \sum_{j=1}^{m} \frac{L_{\mathrm{t}}^{j}(\tilde{p}_j(\boldsymbol{\eta})) + L_{\mathrm{p}}^{j}(p_j(\boldsymbol{\eta}))}{n} \sum_{i=1}^{n} |\eta_j(\boldsymbol{x}_i) - \hat{\eta}_j(\boldsymbol{x}_i)| \,. \tag{72}$$

*Proof.* Following the same line of argument as for Lemma C.4, we get

$$\begin{aligned}
\Psi_{\mathrm{ETU}}(\hat{\boldsymbol{Y}}^{\star}; \boldsymbol{\eta}) - \Psi_{\mathrm{ETU}}\big(\hat{\boldsymbol{Y}}^{\dagger}; \boldsymbol{\eta}\big) &= \Psi_{\mathrm{ETU}}(\hat{\boldsymbol{Y}}^{\star}; \boldsymbol{\eta}) - \Psi_{\mathrm{ETU}}(\hat{\boldsymbol{Y}}^{\star}; \hat{\boldsymbol{\eta}}) \\
&\quad + \Psi_{\mathrm{ETU}}(\hat{\boldsymbol{Y}}^{\star}; \hat{\boldsymbol{\eta}}) - \Psi_{\mathrm{ETU}}\big(\hat{\boldsymbol{Y}}^{\dagger}; \hat{\boldsymbol{\eta}}\big) \\
&\quad + \Psi_{\mathrm{ETU}}\big(\hat{\boldsymbol{Y}}^{\dagger}; \hat{\boldsymbol{\eta}}\big) - \Psi_{\mathrm{ETU}}\big(\hat{\boldsymbol{Y}}^{\dagger}; \boldsymbol{\eta}\big) \\
&\leq \Psi_{\mathrm{ETU}}(\hat{\boldsymbol{Y}}^{\star}; \boldsymbol{\eta}) - \Psi_{\mathrm{ETU}}(\hat{\boldsymbol{Y}}^{\star}; \hat{\boldsymbol{\eta}}) \\
&\quad + \Psi_{\mathrm{ETU}}\big(\hat{\boldsymbol{Y}}^{\dagger}; \hat{\boldsymbol{\eta}}\big) - \Psi_{\mathrm{ETU}}\big(\hat{\boldsymbol{Y}}^{\dagger}; \boldsymbol{\eta}\big) \\
&\leq 2 \sup_{\hat{\boldsymbol{Y}}} |\Psi_{\mathrm{ETU}}(\hat{\boldsymbol{Y}}; \boldsymbol{\eta}) - \Psi_{\mathrm{ETU}}(\hat{\boldsymbol{Y}}; \hat{\boldsymbol{\eta}})| \,. \tag{73}
\end{aligned}$$

Next, we make use of the semi-empirical ETU risk to bound

$$\begin{aligned}
|\Psi_{\mathrm{ETU}}(\hat{\boldsymbol{Y}}; \boldsymbol{\eta}) - \Psi_{\mathrm{ETU}}(\hat{\boldsymbol{Y}}; \hat{\boldsymbol{\eta}})| &\leq |\Psi_{\mathrm{ETU}}(\hat{\boldsymbol{Y}}; \boldsymbol{\eta}) - \tilde{\Psi}_{\mathrm{ETU}}(\hat{\boldsymbol{Y}}; \boldsymbol{\eta})| \\
&\quad + |\tilde{\Psi}_{\mathrm{ETU}}(\hat{\boldsymbol{Y}}; \boldsymbol{\eta}) - \tilde{\Psi}_{\mathrm{ETU}}(\hat{\boldsymbol{Y}}; \hat{\boldsymbol{\eta}})| \\
&\quad + |\tilde{\Psi}_{\mathrm{ETU}}(\hat{\boldsymbol{Y}}; \hat{\boldsymbol{\eta}}) - \Psi_{\mathrm{ETU}}(\hat{\boldsymbol{Y}}; \hat{\boldsymbol{\eta}})| \tag{74}
\end{aligned}$$

The individual terms can now again be bounded by Theorem 5.2 and Lemma C.3. $\square$

**Algorithm 5** Sparse BCA($\boldsymbol{X}, \hat{\boldsymbol{\eta}}^{\mathrm{csr}}, k, \epsilon$)

1: $\hat{\boldsymbol{y}}_i^{\mathrm{csr}} \leftarrow$ select-random-$k(m)$ for all $i \in [n]$
2: $\tilde{\boldsymbol{t}} \leftarrow \frac{1}{n} \sum_{i=1}^n \hat{\boldsymbol{\eta}}^{\mathrm{csr}}(\boldsymbol{x}_i) \odot \hat{\boldsymbol{y}}_i^{\mathrm{csr}}$
3: $\boldsymbol{q} \leftarrow \frac{1}{n} \sum_{i=1}^n \hat{\boldsymbol{y}}_i^{\mathrm{csr}}$
4: $\tilde{\boldsymbol{p}} \leftarrow \frac{1}{n} \sum_{i=1}^n \hat{\boldsymbol{\eta}}^{\mathrm{csr}}(\boldsymbol{x}_i)$
5: $u_{\mathrm{old}} \leftarrow -\infty, \; u_{\mathrm{new}} \leftarrow \Psi(\tilde{\boldsymbol{t}}, \boldsymbol{q}, \tilde{\boldsymbol{p}})$
6: **while** $u_{\mathrm{new}} > u_{\mathrm{old}} + \epsilon$ **do**
7:    **for** $s \in \mathrm{shuffle}([n])$ **do**
8:       $\tilde{\boldsymbol{t}} \leftarrow \tilde{\boldsymbol{t}} - \frac{1}{n} \hat{\boldsymbol{\eta}}^{\mathrm{csr}}(\boldsymbol{x}_s) \odot \hat{\boldsymbol{y}}_s^{\mathrm{csr}}$
9:       $\boldsymbol{q} \leftarrow \boldsymbol{q} - \frac{1}{n} \hat{\boldsymbol{y}}_s^{\mathrm{csr}}$
10:       $\boldsymbol{g}^{\mathrm{csr}} \leftarrow \emptyset$
11:       **for** $(j, \hat{\eta}_j(\boldsymbol{x}_s)) \in \hat{\boldsymbol{\eta}}^{\mathrm{csr}}(\boldsymbol{x}_i)$ **do**
12:          $\psi^j(1) \leftarrow \psi(\tilde{t}_j + \frac{1}{n}\hat{\eta}_j(\boldsymbol{x}_s), q_j + \frac{1}{n}, \tilde{p}_j)$
13:          $\psi^j(0) \leftarrow \psi(\tilde{t}_j, q_j, \tilde{p}_j)$
14:          $\boldsymbol{g}^{\mathrm{csr}} \leftarrow \boldsymbol{g}^{\mathrm{csr}} \cup \{(j, \psi^j(1) - \psi^j(0))\}$
15:       $\hat{\boldsymbol{y}}_s^{\mathrm{csr}} \leftarrow$ select-top-$k(\boldsymbol{g}^{\mathrm{csr}})$
16:       $\tilde{\boldsymbol{t}} \leftarrow \tilde{\boldsymbol{t}} + \frac{1}{n} \hat{\boldsymbol{\eta}}^{\mathrm{csr}}(\boldsymbol{x}_s) \odot \hat{\boldsymbol{y}}_s^{\mathrm{csr}}$
17:       $\boldsymbol{q} \leftarrow \boldsymbol{q} + \frac{1}{n} \hat{\boldsymbol{y}}_s^{\mathrm{csr}}$
18:    $u_{\mathrm{old}} \leftarrow u_{\mathrm{new}}, \; u_{\mathrm{new}} \leftarrow \Psi(\tilde{\boldsymbol{t}}, \boldsymbol{q}, \tilde{\boldsymbol{p}})$
19: **return** $\hat{\boldsymbol{Y}}^{\mathrm{csr}}$

---

**Algorithm 6** Sparse BCA for coverage ($\boldsymbol{X}, \hat{\boldsymbol{\eta}}^{\mathrm{csr}}, k, \epsilon$)

1: $\hat{\boldsymbol{y}}_i^{\mathrm{csr}} \leftarrow$ select-random-$k(m)$ for all $i \in [n]$
2: $\boldsymbol{f} \leftarrow \boldsymbol{1}$
3: **for** $i \in [n]$ **do**
4:    **for** $(j, \hat{\eta}_j(\boldsymbol{x}_i)) \in \hat{\boldsymbol{\eta}}^{\mathrm{csr}}(\boldsymbol{x}_i) \odot \hat{\boldsymbol{y}}_i$ **do**
5:       $f_j \leftarrow f_j(1 - \hat{\eta}_j(\boldsymbol{x}_i))$
6: $u_{\mathrm{old}} \leftarrow -\infty, \; u_{\mathrm{new}} \leftarrow 1 - (m^{-1} \sum_{j=1}^m f_j)$
7: **while** $u_{\mathrm{new}} > u_{\mathrm{old}} + \epsilon$ **do**
8:    **for** $s \in \mathrm{shuffle}([n])$ **do**
9:       **for** $(j, \hat{\eta}_j(\boldsymbol{x}_s)) \in \hat{\boldsymbol{\eta}}^{\mathrm{csr}}(\boldsymbol{x}_s) \odot \hat{\boldsymbol{y}}_s^{\mathrm{csr}}$ **do**
10:          $f_j \leftarrow f_j/(1 - \hat{\eta}_j^{\mathrm{csr}}(\boldsymbol{x}_s))$
11:       $\boldsymbol{g}^{\mathrm{csr}} \leftarrow \emptyset$
12:       **for** $(j, \hat{\eta}_j(\boldsymbol{x}_s)) \in \hat{\boldsymbol{\eta}}^{\mathrm{csr}}(\boldsymbol{x}_s)$ **do**
13:          $\boldsymbol{g}^{\mathrm{csr}} \leftarrow \boldsymbol{g}^{\mathrm{csr}} \cup \{(j, \hat{\eta}_j(\boldsymbol{x}_s) f_j)\}$
14:       $\hat{\boldsymbol{y}}_s^{\mathrm{csr}} \leftarrow$ select-top-$k(\boldsymbol{g}^{\mathrm{csr}})$
15:       **for** $(j, \hat{\eta}_j(\boldsymbol{x}_s)) \in \hat{\boldsymbol{\eta}}^{\mathrm{csr}}(\boldsymbol{x}_s) \odot \hat{\boldsymbol{y}}_s^{\mathrm{csr}}$ **do**
16:          $f_j \leftarrow f_j(1 - \hat{\eta}_j(\boldsymbol{x}_s))$
17:    $u_{\mathrm{old}} \leftarrow u_{\mathrm{new}}, \; u_{\mathrm{new}} \leftarrow 1 - (m^{-1} \sum_{j=1}^m f_j)$
18: **return** $\hat{\boldsymbol{Y}}^{\mathrm{csr}}$

---

**Algorithm 7** Sparse greedy ($\boldsymbol{X}, \hat{\boldsymbol{\eta}}^{\mathrm{csr}}, k$)

1: $\tilde{\boldsymbol{t}} \leftarrow \boldsymbol{0}, \; \boldsymbol{q} \leftarrow \boldsymbol{0}, \; \tilde{\boldsymbol{p}} \leftarrow \boldsymbol{0}$
2: **for** $i \in [n]$ **do**
3:    $\tilde{\boldsymbol{p}} \leftarrow \boldsymbol{p} + \frac{1}{n} \hat{\boldsymbol{\eta}}^{\mathrm{csr}}(\boldsymbol{x}_i)$
4:    $\boldsymbol{g}^{\mathrm{csr}} \leftarrow \emptyset$
5:    **for** $(j, \hat{\eta}_j(\boldsymbol{x}_i)) \in \hat{\boldsymbol{\eta}}^{\mathrm{csr}}(\boldsymbol{x}_i)$ **do**
6:       $\psi^j(1) \leftarrow \psi(\tilde{t}_j + \frac{1}{n}\hat{\eta}_j(\boldsymbol{x}_i), q_j + \frac{1}{n}, \tilde{p}_j)$
7:       $\psi^j(0) \leftarrow \psi(\tilde{t}_j, q_j, \tilde{p}_j)$
8:       $\boldsymbol{g}^{\mathrm{csr}} \leftarrow \boldsymbol{g}^{\mathrm{csr}} \cup \{(j, \psi^j(1) - \psi^j(0))\}$
9:    $\hat{\boldsymbol{y}}_i^{\mathrm{csr}} \leftarrow$ select top-k($\boldsymbol{g}^{\mathrm{csr}}$)
10:    $\tilde{\boldsymbol{t}} \leftarrow \tilde{\boldsymbol{t}} + \frac{1}{n} \hat{\boldsymbol{\eta}}^{\mathrm{csr}}(\boldsymbol{x}_i) \odot \hat{\boldsymbol{y}}_i^{\mathrm{csr}}$
11:    $\boldsymbol{q} \leftarrow \boldsymbol{q} + \frac{1}{n} \hat{\boldsymbol{y}}_i^{\mathrm{csr}}$
12: **return** $\hat{\boldsymbol{Y}}^{\mathrm{csr}}$

---

**Algorithm 8** Sparse greedy for coverage ($\boldsymbol{X}, \hat{\boldsymbol{\eta}}^{\mathrm{csr}}, k$)

1: $\boldsymbol{f} \leftarrow \boldsymbol{1}$
2: **for** $i \in [n]$ **do**
3:    $\boldsymbol{g}^{\mathrm{csr}} \leftarrow \emptyset$
4:    **for** $(j, \hat{\eta}_j(\boldsymbol{x}_s)) \in \hat{\boldsymbol{\eta}}^{\mathrm{csr}}(\boldsymbol{x}_i)$ **do**
5:       $\boldsymbol{g}^{\mathrm{csr}} \leftarrow^{\mathrm{csr}} \boldsymbol{g} \cup \{(j, \hat{\eta}_j(\boldsymbol{x}_s) f_j)\}$
6:    $\hat{\boldsymbol{y}}_i^{\mathrm{csr}} \leftarrow$ select top-k($\boldsymbol{g}^{\mathrm{csr}}$)
7:    **for** $(j, \hat{\eta}_j(\boldsymbol{x}_i)) \in \hat{\boldsymbol{\eta}}^{\mathrm{csr}}(\boldsymbol{x}_i) \odot \hat{\boldsymbol{y}}_i^{\mathrm{csr}}$ **do**
8:       $f_j \leftarrow f_j(1 - \hat{\eta}_j(\boldsymbol{x}_i))$
9: **return** $\hat{\boldsymbol{Y}}^{\mathrm{csr}}$

# D   Sparse and greedy algorithms

In this section, we present pseudocodes for sparse variants of the introduced algorithms, which we used in the main experiment to compute efficiently the results for large datasets. These variants use *compressed sparse row* (CSR) representation for row vectors. CSR vectors are represented as a list of tuples $\boldsymbol{a}_i^{\mathrm{csr}} := \{(\mathrm{index}, \mathrm{value}) : \mathrm{value} \neq 0\}$. In these algorithms, we replace $\hat{\boldsymbol{\eta}}(\boldsymbol{x}_i)$ and $\hat{\boldsymbol{y}}_i$ with their sparse variants, $\hat{\boldsymbol{\eta}}^{\mathrm{csr}}(\boldsymbol{x}_i) := \{(j, \hat{\eta}_j(\boldsymbol{x}_i)) : \hat{\eta}_j(\boldsymbol{x}_i) \neq 0\}$ and $\hat{\boldsymbol{y}}_i^{\mathrm{csr}} := \{(j, \hat{y}_{ij}) : \hat{y}_{ij} \neq 0\}$, respectively. We need to ensure that of both $\hat{\boldsymbol{\eta}}^{\mathrm{csr}}(\boldsymbol{x}_i)$ and $\hat{\boldsymbol{y}}_i^{\mathrm{csr}}$ are ordered by their indices for efficient element-wise multiplication (Hadamard product) between them. If we assume that the size of $|\hat{\boldsymbol{y}}_i^{\mathrm{csr}}| = k$ and $|\hat{\boldsymbol{\eta}}^{\mathrm{csr}}(\boldsymbol{x}_i)| = k'$, the resulting time and space complexity of the sparse algorithms are $\mathcal{O}(n(k' + k \log k))$ and $\mathcal{O}(n(k' + k))$, respectively. Additionally, we present the greedy variants of sparse BCA algorithms introduced so far that do only one pass over the dataset.

We present the sparse variant of BCA (Algorithm 1) in Algorithm 5 and the sparse variant of Greedy (Algorithm 3) in Algorithm 7. The sparse versions of their specialized counterparts for coverage are presented in Algorithm 6 and Algorithm 8.

# E   Extended results, experiments and details of empirical comparison

## E.1   Datasets characteristics

In Table 5 we include an overview of the main characteristics of the datasets used in this paper.

Table 5: Multi-label datasets from XMLC repository [6]. APpL and ALpP represent average points per label and average labels per point, respectively.

| Dataset | #Labels | #Training | #Testing | #Features | APpL | ALpP |
|---|---|---|---|---|---|---|
| EURLEX-4K | 3,993 | 15,539 | 3,809 | 5,000 | 25.7 | 5.3 |
| WIKI-31K | 30,938 | 14,146 | 6,616 | 101,938 | 8.5 | 18.6 |
| AMAZONCAT-13K | 13,330 | 1,186,239 | 306,782 | 203,882 | 448.6 | 5.0 |
| WIKIPEDIALARGE-500K | 501,070 | 1,813,391 | 783,743 | 2,381,304 | 24.8 | 4.8 |
| AMAZON-670K | 670,091 | 490,449 | 153,025 | 135,909 | 4.0 | 5.5 |

## E.2   Extended results of the main experiment

In this section, we present the extended results of our main experiment from Section 7. Here in Table 6 and Table 7, we present additional results for $k = 1$, as well as results for additional methods:

- MACRO-P$_{\text{GREED}}$, MACRO-F1$_{\text{GREED}}$, COV$_{\text{GREED}}$– the greedy algorithms (Algorithm 3) for optimizing macro-precision, -F1, and coverage (Algorithm 4),

- MACRO-R$_{\text{PRIOR}}$– the optimal strategy for macro-recall: selection of $k$ labels with the highest $p_j^{-1}\eta_j$, with $p$ estimated on the training set.

Because both greedy and block coordinate-ascent algorithms depend on the order in which examples are presented, we check if this has a significant influence on the results. We ran the procedures 5 times with different seeds and presented both mean and standard deviation, which is mostly less than 0.25 for the macro measures.

## E.3   Results on true labels

In the main experiment, the reported performance depends on three things: the inherent difficulty of the data, the success of the inference algorithm and the quality of the provided marginal probabilities. To be able to judge these terms, we also ran the inference algorithms on probabilities generated from the test set labels: $\eta_j(\boldsymbol{x}_i) = 1$ iff $Y_{ij} = 1$ to rest only our inference strategy. We present the result of this experiment in Table 8. Notice that in this experiment, the specialized inference strategies are almost always the best on measures they aim to optimize. Also, the obtained results for macro measures are significantly below 100%. This is because, in this datasets, not all labels are represented with even one positive training example in the test set. Still, there remains a significant gap between these results and the results of the main experiments, with probability estimates coming from the LIGHTXML model.

## E.4   Impact of stopping criterion and using only top-$k'$ predictions on predictive and computational performance.

In this section, we investigate the impact of stopping criterion and using precalculated top-$k'$ predictions on the predictive performance as well as computational performance.

To materialize all marginal probability estimates for datasets like WIKIPEDIALARGE-500K and AMAZON-670K, one requires an enormous amount of memory and time. Because of that, we are not able to calculate the exact results for these datasets, and instead, we used pre-select top-$k'$ labels with the highest $\hat{\eta}_j$ as described in Section 6. In the main experiment in Section 7, we used $k' = 100$ for smaller datasets (EURLEX-4K, WIKI-31K, AMAZONCAT-13K) and $k' = 1000$ for larger datasets (WIKIPEDIALARGE-500K and AMAZON-670K). Here, we additionally compare the results for $k' = 100$ and $k' = 1000$ for WIKIPEDIALARGE-500K and AMAZON-670K investigating the impact of predictive and computational performance.

Table 6: Mean results with standard deviation of different inference strategies on @$k$ measures calculated with $k \in \{1,3\}$, using probability estimates from LIGHTXML model. Notation: P—precision, R—recall, F1—F1-measure, Cov—Coverage. The green color indicates cells in which the strategy matches the metric. The best results are in **bold** and the second best are in *italic*.

| Inference strategy | Instance @1 P ±std | Instance @1 R ±std | Macro @1 P ±std | Macro @1 R ±std | Macro @1 F1 ±std | Macro @1 Cov ±std | Instance @3 P ±std | Instance @3 R ±std | Macro @3 P ±std | Macro @3 R ±std | Macro @3 F1 ±std | Macro @3 Cov ±std |
|---|---|---|---|---|---|---|---|---|---|---|---|---|
| **EURLEX-4K** | | | | | | | | | | | | |
| TOP-K | **85.77** ±0.00 | 17.41 ±0.00 | 15.02 ±0.00 | 5.10 ±0.00 | 6.98 ±0.00 | 16.63 ±0.00 | **74.20** ±0.00 | 44.21 ±0.00 | 24.85 ±0.00 | 16.45 ±0.00 | 18.63 ±0.00 | 31.27 ±0.00 |
| PS-K | 74.39 ±0.00 | 14.99 ±0.00 | 24.76 ±0.00 | 13.00 ±0.00 | 15.22 ±0.00 | 28.82 ±0.00 | 69.01 ±0.00 | 41.06 ±0.00 | 30.12 ±0.00 | 23.72 ±0.00 | 25.08 ±0.00 | 40.29 ±0.00 |
| POW-K $\beta=0.25$ | 76.40 ±0.00 | 15.42 ±0.00 | 24.58 ±0.00 | 12.50 ±0.00 | 14.86 ±0.00 | 28.29 ±0.00 | 70.08 ±0.00 | 41.73 ±0.00 | 29.43 ±0.00 | 22.87 ±0.00 | 24.34 ±0.00 | 38.93 ±0.00 |
| POW-K $\beta=0.5$ | 70.19 ±0.00 | 14.07 ±0.00 | 24.72 ±0.00 | 13.48 ±0.00 | 15.55 ±0.00 | 29.07 ±0.00 | 65.35 ±0.00 | 38.95 ±0.00 | 30.49 ±0.00 | 25.26 ±0.00 | 25.93 ±0.00 | 42.11 ±0.00 |
| LOG-K | 78.60 ±0.00 | 15.88 ±0.00 | 23.67 ±0.00 | 11.61 ±0.00 | 14.02 ±0.00 | 27.20 ±0.00 | 71.03 ±0.00 | 42.30 ±0.00 | 29.04 ±0.00 | 22.10 ±0.00 | 23.76 ±0.00 | 37.99 ±0.00 |
| MACRO-P$_{GREED}$ | 50.82 ±0.47 | 10.04 ±0.06 | 29.02 ±0.20 | 10.82 ±0.10 | 13.48 ±0.11 | 33.85 ±0.28 | 49.15 ±0.32 | 29.16 ±0.16 | 31.18 ±0.32 | 22.63 ±0.27 | 23.18 ±0.29 | 44.30 ±0.20 |
| MACRO-P$_{BCA}$ | 47.16 ±0.34 | 9.38 ±0.08 | **33.76** ±0.21 | 10.85 ±0.18 | 14.24 ±0.19 | 34.18 ±0.21 | 41.13 ±0.41 | 24.29 ±0.22 | **38.78** ±0.09 | 16.77 ±0.05 | 20.63 ±0.05 | 40.60 ±0.08 |
| MACRO-R$_{PRIOR}$ | 57.59 ±0.00 | 11.49 ±0.00 | 23.52 ±0.00 | **14.08** ±0.00 | 15.44 ±0.00 | 28.64 ±0.00 | 53.09 ±0.00 | 31.55 ±0.00 | 29.85 ±0.00 | **26.93** ±0.00 | 25.59 ±0.00 | 44.87 ±0.00 |
| MACRO-R$_{BCA}$ | 44.55 ±0.00 | 8.87 ±0.00 | 22.49 ±0.00 | 12.91 ±0.00 | 14.59 ±0.00 | 26.14 ±0.00 | 41.47 ±0.00 | 24.61 ±0.00 | 30.36 ±0.00 | 25.16 ±0.00 | 24.37 ±0.00 | 44.26 ±0.00 |
| MACRO-F1$_{GREED}$ | 63.90 ±0.11 | 12.78 ±0.02 | 28.35 ±0.13 | 12.77 ±0.14 | *15.77* ±0.14 | 33.29 ±0.20 | 60.95 ±0.24 | 36.30 ±0.17 | 31.60 ±0.11 | 24.93 ±0.11 | *26.26* ±0.12 | 43.98 ±0.20 |
| MACRO-F1$_{BCA}$ | 62.24 ±0.27 | 12.49 ±0.04 | 29.67 ±0.11 | *13.55* ±0.05 | **16.78** ±0.07 | 33.87 ±0.11 | 61.96 ±0.06 | 36.94 ±0.05 | *34.42* ±0.06 | 25.13 ±0.05 | **27.59** ±0.05 | 44.52 ±0.06 |
| COV$_{GREED}$ | 45.44 ±0.64 | 8.96 ±0.13 | 27.92 ±0.26 | 10.79 ±0.13 | 13.11 ±0.11 | *34.55* ±0.28 | 26.64 ±0.12 | 15.54 ±0.08 | 26.78 ±0.19 | 21.23 ±0.06 | 18.38 ±0.07 | *46.27* ±0.07 |
| COV$_{BCA}$ | 43.77 ±0.31 | 8.61 ±0.06 | *30.89* ±0.21 | 11.65 ±0.17 | 14.06 ±0.14 | **36.14** ±0.24 | 24.00 ±0.17 | 14.03 ±0.09 | 31.46 ±0.18 | 21.30 ±0.05 | 19.05 ±0.06 | **47.19** ±0.12 |
| **AMAZONCAT-13K** | | | | | | | | | | | | |
| TOP-K | **96.38** ±0.00 | 27.49 ±0.00 | 11.29 ±0.00 | 1.81 ±0.00 | 2.79 ±0.00 | 13.08 ±0.00 | **83.35** ±0.00 | 63.01 ±0.00 | 25.71 ±0.00 | 11.37 ±0.00 | 13.83 ±0.00 | 33.05 ±0.00 |
| PS-K | 86.41 ±0.00 | 24.26 ±0.00 | 42.58 ±0.00 | 25.12 ±0.00 | 28.17 ±0.00 | 55.90 ±0.00 | 79.09 ±0.00 | 60.15 ±0.00 | 43.89 ±0.00 | 38.64 ±0.00 | 38.56 ±0.00 | 62.06 ±0.00 |
| POW-K $\beta=0.25$ | 79.34 ±0.00 | 21.47 ±0.00 | 40.09 ±0.00 | 23.99 ±0.00 | 27.09 ±0.00 | 53.74 ±0.00 | 74.40 ±0.00 | 56.59 ±0.00 | 41.87 ±0.00 | 39.07 ±0.00 | 38.17 ±0.00 | 61.39 ±0.00 |
| POW-K $\beta=0.5$ | 71.88 ±0.00 | 19.08 ±0.00 | 40.09 ±0.00 | 28.57 ±0.00 | 30.02 ±0.00 | 59.59 ±0.00 | 66.52 ±0.00 | 50.75 ±0.00 | 38.18 ±0.00 | 46.42 ±0.00 | 39.26 ±0.00 | 67.76 ±0.00 |
| LOG-K | 80.12 ±0.00 | 21.30 ±0.00 | 37.42 ±0.00 | 19.75 ±0.00 | 23.32 ±0.00 | 47.55 ±0.00 | 75.30 ±0.00 | 56.79 ±0.00 | 41.09 ±0.00 | 33.92 ±0.00 | 35.04 ±0.00 | 56.05 ±0.00 |
| MACRO-P$_{GREED}$ | 84.10 ±0.15 | 23.96 ±0.02 | 50.08 ±0.04 | 27.65 ±0.08 | 30.90 ±0.05 | 73.70 ±0.09 | 65.58 ±0.94 | 49.76 ±0.72 | 44.69 ±0.12 | 48.79 ±0.06 | 41.63 ±0.07 | 79.96 ±0.06 |
| MACRO-P$_{BCA}$ | 92.29 ±0.03 | 26.30 ±0.01 | **63.33** ±0.07 | 22.58 ±0.05 | 29.98 ±0.03 | 73.35 ±0.10 | 54.97 ±0.35 | 41.61 ±0.24 | **64.27** ±0.03 | 29.22 ±0.03 | 35.76 ±0.02 | 76.18 ±0.05 |
| MACRO-R$_{PRIOR}$ | 61.09 ±0.00 | 16.00 ±0.00 | 36.45 ±0.00 | *34.11* ±0.00 | 31.01 ±0.00 | 64.32 ±0.00 | 49.82 ±0.00 | 38.32 ±0.00 | 30.64 ±0.00 | *54.38* ±0.00 | 34.06 ±0.00 | 75.91 ±0.00 |
| MACRO-R$_{BCA}$ | 59.70 ±0.00 | 15.80 ±0.00 | 37.98 ±0.00 | **37.71** ±0.00 | 32.90 ±0.00 | 69.52 ±0.00 | 47.74 ±0.00 | 37.13 ±0.00 | 31.40 ±0.00 | **58.32** ±0.00 | 34.68 ±0.00 | 80.71 ±0.00 |
| MACRO-F1$_{GREED}$ | 71.87 ±0.02 | 19.14 ±0.06 | 46.48 ±0.08 | 30.92 ±0.04 | *34.45* ±0.05 | 70.44 ±0.09 | 69.04 ±0.02 | 52.71 ±0.01 | 42.55 ±0.06 | 49.15 ±0.07 | *43.27* ±0.05 | 75.52 ±0.08 |
| MACRO-F1$_{BCA}$ | 72.50 ±0.00 | 19.30 ±0.00 | *50.42* ±0.04 | 31.67 ±0.02 | **36.33** ±0.02 | 71.90 ±0.04 | 70.61 ±0.00 | 53.86 ±0.00 | *51.95* ±0.01 | 48.22 ±0.02 | **47.93** ±0.01 | 77.83 ±0.02 |
| COV$_{GREED}$ | 10.14 ±0.02 | 1.78 ±0.00 | 32.17 ±0.09 | 22.70 ±0.05 | 15.13 ±0.04 | *74.83* ±0.15 | 4.83 ±0.01 | 2.44 ±0.00 | 23.11 ±0.12 | 33.89 ±0.04 | 12.83 ±0.02 | *80.86* ±0.07 |
| COV$_{BCA}$ | 9.79 ±0.01 | 1.73 ±0.00 | 41.23 ±0.03 | 24.94 ±0.04 | 17.08 ±0.03 | **77.28** ±0.03 | 4.53 ±0.00 | 2.29 ±0.00 | 34.93 ±0.08 | 35.16 ±0.02 | 15.91 ±0.03 | **82.67** ±0.03 |
| **WIKI-31K** | | | | | | | | | | | | |
| TOP-K | **87.85** ±0.00 | 5.27 ±0.00 | 0.99 ±0.00 | 0.14 ±0.00 | 0.23 ±0.00 | 1.11 ±0.00 | **77.17** ±0.00 | 13.48 ±0.00 | 2.01 ±0.00 | 0.50 ±0.00 | 0.72 ±0.00 | 2.54 ±0.00 |
| PS-K | 73.00 ±0.00 | *4.33* ±0.00 | 3.01 ±0.00 | 0.68 ±0.00 | 1.00 ±0.00 | 3.78 ±0.00 | *67.95* ±0.00 | *11.89* ±0.00 | 3.89 ±0.00 | 1.47 ±0.00 | 1.93 ±0.00 | 5.47 ±0.00 |
| POW-K $\beta=0.25$ | 72.93 ±0.00 | 4.29 ±0.00 | 2.80 ±0.00 | 0.61 ±0.00 | 0.91 ±0.00 | 3.50 ±0.00 | 66.00 ±0.00 | 11.50 ±0.00 | 3.62 ±0.00 | 1.35 ±0.00 | 1.79 ±0.00 | 5.15 ±0.00 |
| POW-K $\beta=0.5$ | 62.12 ±0.00 | 3.65 ±0.00 | 3.44 ±0.00 | 0.90 ±0.00 | 1.27 ±0.00 | 4.57 ±0.00 | 55.06 ±0.00 | 9.59 ±0.00 | 4.49 ±0.00 | 2.14 ±0.00 | 2.59 ±0.00 | 6.98 ±0.00 |
| LOG-K | *73.17* ±0.00 | 4.32 ±0.00 | 2.52 ±0.00 | 0.53 ±0.00 | 0.81 ±0.00 | 3.16 ±0.00 | 65.20 ±0.00 | 11.34 ±0.00 | 3.40 ±0.00 | 1.24 ±0.00 | 1.66 ±0.00 | 4.84 ±0.00 |
| MACRO-P$_{GREED}$ | 37.14 ±0.49 | 2.15 ±0.04 | 5.17 ±0.09 | 1.04 ±0.02 | 1.53 ±0.03 | 6.17 ±0.07 | 32.65 ±0.13 | 5.62 ±0.02 | 6.89 ±0.06 | 2.78 ±0.05 | 3.44 ±0.05 | 10.02 ±0.06 |
| MACRO-P$_{BCA}$ | 31.46 ±0.24 | 1.80 ±0.01 | **6.48** ±0.05 | 1.24 ±0.02 | 1.80 ±0.02 | *6.48* ±0.05 | 32.86 ±0.66 | 5.66 ±0.12 | **9.13** ±0.01 | 2.55 ±0.03 | 3.38 ±0.03 | 9.21 ±0.02 |
| MACRO-R$_{PRIOR}$ | 45.30 ±0.00 | 2.63 ±0.00 | 3.91 ±0.00 | **1.51** ±0.00 | *1.85* ±0.00 | 5.37 ±0.00 | 36.67 ±0.00 | 6.35 ±0.00 | 5.34 ±0.00 | **3.55** ±0.00 | 3.60 ±0.00 | 9.38 ±0.00 |
| MACRO-R$_{BCA}$ | 12.95 ±0.00 | 0.74 ±0.00 | 2.29 ±0.00 | 1.27 ±0.00 | 1.41 ±0.00 | 2.61 ±0.00 | 13.78 ±0.00 | 2.36 ±0.00 | 4.77 ±0.00 | *3.24* ±0.00 | 3.10 ±0.00 | 7.26 ±0.00 |
| MACRO-F1$_{GREED}$ | 43.15 ±0.46 | 2.53 ±0.02 | 5.05 ±0.08 | 1.23 ±0.02 | *1.78* ±0.03 | 6.22 ±0.06 | 38.12 ±0.15 | 6.65 ±0.02 | 6.56 ±0.03 | 2.91 ±0.03 | *3.63* ±0.03 | 9.91 ±0.02 |
| MACRO-F1$_{BCA}$ | 39.82 ±0.20 | 2.33 ±0.01 | *5.67* ±0.04 | *1.41* ±0.01 | **2.03** ±0.02 | 6.43 ±0.04 | 37.32 ±0.07 | 6.52 ±0.01 | *7.81* ±0.04 | 3.15 ±0.02 | **4.10** ±0.02 | *10.42* ±0.02 |
| COV$_{GREED}$ | 40.24 ±0.44 | 2.33 ±0.03 | 4.34 ±0.05 | 0.93 ±0.01 | 1.36 ±0.02 | 6.15 ±0.06 | 27.39 ±0.14 | 4.66 ±0.03 | 5.21 ±0.03 | 2.72 ±0.01 | 2.97 ±0.01 | 10.07 ±0.02 |
| COV$_{BCA}$ | 40.52 ±0.38 | 2.35 ±0.02 | 5.10 ±0.02 | 1.11 ±0.00 | 1.60 ±0.00 | **6.72** ±0.03 | 27.00 ±0.12 | 4.63 ±0.01 | 6.24 ±0.03 | 3.18 ±0.01 | 3.41 ±0.01 | **11.04** ±0.04 |
| **WIKIPEDIALARGE-500K** | | | | | | | | | | | | |
| TOP-K | **74.94** ±0.00 | 24.26 ±0.00 | 12.21 ±0.00 | 6.74 ±0.00 | 7.73 ±0.00 | 15.36 ±0.00 | **56.02** ±0.00 | 45.70 ±0.00 | 20.15 ±0.00 | 18.87 ±0.00 | 17.14 ±0.00 | 32.24 ±0.00 |
| PS-K | 72.00 ±0.00 | *23.90* ±0.00 | 18.99 ±0.00 | 10.96 ±0.00 | 12.47 ±0.00 | 23.58 ±0.00 | *54.91* ±0.00 | *45.61* ±0.00 | 23.33 ±0.00 | 22.68 ±0.00 | 20.41 ±0.00 | 37.85 ±0.00 |
| POW-K $\beta=0.25$ | 71.75 ±0.00 | 23.73 ±0.00 | 17.59 ±0.00 | 9.90 ±0.00 | 11.37 ±0.00 | 21.89 ±0.00 | 54.24 ±0.00 | 45.09 ±0.00 | 22.48 ±0.00 | 21.57 ±0.00 | 19.54 ±0.00 | 36.29 ±0.00 |
| POW-K $\beta=0.5$ | 69.11 ±0.00 | 23.26 ±0.00 | 19.80 ±0.00 | 11.59 ±0.00 | 13.14 ±0.00 | 24.96 ±0.00 | 51.81 ±0.00 | 43.94 ±0.00 | 23.73 ±0.00 | 23.67 ±0.00 | 21.13 ±0.00 | 39.36 ±0.00 |
| LOG-K | *72.89* ±0.00 | 23.86 ±0.00 | 15.56 ±0.00 | 8.57 ±0.00 | 9.88 ±0.00 | 19.32 ±0.00 | 54.82 ±0.00 | 45.21 ±0.00 | 21.43 ±0.00 | 20.18 ±0.00 | 18.38 ±0.00 | 34.28 ±0.00 |
| MACRO-P$_{GREED}$ | 54.81 ±0.03 | 18.49 ±0.01 | 26.59 ±0.02 | 12.99 ±0.01 | 15.55 ±0.01 | 34.05 ±0.03 | 44.19 ±0.01 | 37.85 ±0.02 | 26.33 ±0.02 | 25.49 ±0.02 | 23.29 ±0.01 | 46.04 ±0.02 |
| MACRO-P$_{BCA}$ | 47.68 ±0.01 | 15.82 ±0.00 | **32.70** ±0.02 | 13.38 ±0.01 | **16.94** ±0.01 | 36.25 ±0.02 | 25.19 ±0.05 | 21.81 ±0.03 | **37.69** ±0.00 | 20.18 ±0.01 | 23.44 ±0.01 | 45.08 ±0.01 |
| MACRO-R$_{PRIOR}$ | 63.83 ±0.00 | 22.18 ±0.00 | 22.59 ±0.00 | 14.26 ±0.00 | 15.65 ±0.00 | 29.25 ±0.00 | 46.16 ±0.00 | 40.99 ±0.00 | 24.82 ±0.00 | *26.93* ±0.00 | 22.92 ±0.00 | 44.12 ±0.00 |
| MACRO-R$_{BCA}$ | 60.61 ±0.00 | 21.47 ±0.00 | 24.76 ±0.00 | **14.91** ±0.00 | 16.71 ±0.00 | 32.61 ±0.00 | 43.40 ±0.00 | 39.60 ±0.00 | 25.35 ±0.00 | **27.55** ±0.00 | 23.72 ±0.00 | 46.30 ±0.00 |
| MACRO-F1$_{GREED}$ | 62.48 ±0.02 | 21.42 ±0.01 | 25.28 ±0.02 | 12.84 ±0.01 | *15.39* ±0.01 | 31.99 ±0.03 | 48.66 ±0.01 | 41.43 ±0.01 | 25.52 ±0.01 | 24.85 ±0.01 | 22.76 ±0.01 | 43.68 ±0.01 |
| MACRO-F1$_{BCA}$ | 61.41 ±0.00 | 20.98 ±0.00 | *29.23* ±0.01 | 13.61 ±0.00 | *16.84* ±0.00 | 34.32 ±0.01 | 43.82 ±0.00 | 36.40 ±0.00 | *35.42* ±0.01 | 23.69 ±0.00 | **26.01** ±0.00 | 46.36 ±0.00 |
| COV$_{GREED}$ | 47.41 ±0.03 | 15.55 ±0.02 | 23.70 ±0.02 | 13.07 ±0.01 | 14.76 ±0.01 | 33.36 ±0.02 | 28.67 ±0.00 | 25.48 ±0.00 | 23.56 ±0.00 | 25.44 ±0.00 | 20.65 ±0.00 | *47.46* ±0.02 |
| COV$_{BCA}$ | 46.84 ±0.02 | 15.47 ±0.01 | 26.16 ±0.01 | *14.30* ±0.01 | 16.05 ±0.01 | **36.06** ±0.01 | 27.31 ±0.00 | 24.55 ±0.00 | 25.92 ±0.00 | 26.76 ±0.01 | 21.60 ±0.00 | **50.16** ±0.01 |
| **AMAZON-670K** | | | | | | | | | | | | |
| TOP-K | **46.86** ±0.00 | 9.68 ±0.00 | 5.05 ±0.00 | 3.34 ±0.00 | 3.68 ±0.00 | 6.12 ±0.00 | **41.71** ±0.00 | 24.08 ±0.00 | 10.77 ±0.00 | 9.68 ±0.00 | 9.51 ±0.00 | 14.35 ±0.00 |
| PS-K | 44.95 ±0.00 | 9.37 ±0.00 | 5.87 ±0.00 | 3.97 ±0.00 | 4.35 ±0.00 | 6.88 ±0.00 | 41.05 ±0.00 | 23.77 ±0.00 | 11.86 ±0.00 | 10.54 ±0.00 | 10.42 ±0.00 | 15.50 ±0.00 |
| POW-K $\beta=0.25$ | 45.49 ±0.00 | 9.45 ±0.00 | 5.76 ±0.00 | 3.86 ±0.00 | 4.25 ±0.00 | 6.79 ±0.00 | 41.30 ±0.00 | 23.89 ±0.00 | 11.65 ±0.00 | 10.35 ±0.00 | 10.24 ±0.00 | 15.27 ±0.00 |
| POW-K $\beta=0.5$ | 44.68 ±0.00 | 9.33 ±0.00 | 5.94 ±0.00 | 4.04 ±0.00 | 4.41 ±0.00 | 6.93 ±0.00 | 40.90 ±0.00 | 23.71 ±0.00 | 11.99 ±0.00 | 10.65 ±0.00 | 10.53 ±0.00 | 15.64 ±0.00 |
| LOG-K | 46.09 ±0.00 | 9.54 ±0.00 | 5.55 ±0.00 | 3.68 ±0.00 | 4.06 ±0.00 | 6.61 ±0.00 | *41.54* ±0.00 | 24.00 ±0.00 | 11.34 ±0.00 | 10.10 ±0.00 | 9.98 ±0.00 | 14.94 ±0.00 |
| MACRO-P$_{GREED}$ | 43.47 ±0.04 | 9.03 ±0.01 | 7.94 ±0.04 | 3.86 ±0.00 | 4.69 ±0.01 | 8.51 ±0.01 | 38.80 ±0.02 | 22.48 ±0.01 | 14.18 ±0.01 | 10.63 ±0.01 | 11.34 ±0.01 | 17.20 ±0.01 |
| MACRO-P$_{BCA}$ | 42.37 ±0.06 | 8.83 ±0.01 | **8.73** ±0.01 | 3.99 ±0.01 | *4.91* ±0.00 | **8.77** ±0.01 | 33.79 ±1.46 | 19.75 ±0.80 | **17.27** ±0.37 | 10.53 ±0.11 | *12.12* ±0.02 | **17.75** ±0.01 |
| MACRO-R$_{PRIOR}$ | 43.08 ±0.00 | 9.08 ±0.00 | 6.09 ±0.00 | *4.26* ±0.00 | 4.61 ±0.00 | 6.98 ±0.00 | 39.85 ±0.00 | 23.20 ±0.00 | 12.48 ±0.00 | *11.07* ±0.00 | 10.91 ±0.00 | 16.12 ±0.00 |
| MACRO-R$_{BCA}$ | 42.33 ±0.00 | 8.92 ±0.00 | 6.82 ±0.00 | **4.33** ±0.00 | 4.86 ±0.00 | 7.46 ±0.00 | 39.42 ±0.70 | 22.92 ±0.36 | 13.71 ±0.61 | **11.17** ±0.21 | 11.47 ±0.34 | 17.12 ±0.41 |
| MACRO-F1$_{GREED}$ | 44.04 ±0.03 | 9.17 ±0.01 | 7.74 ±0.01 | 3.96 ±0.01 | *4.75* ±0.01 | 8.37 ±0.01 | 39.76 ±0.02 | 23.02 ±0.01 | 13.79 ±0.01 | 10.61 ±0.01 | 11.22 ±0.01 | 16.93 ±0.01 |
| MACRO-F1$_{BCA}$ | 43.36 ±0.03 | 9.05 ±0.01 | *8.32* ±0.01 | 4.14 ±0.01 | **5.01** ±0.00 | 8.56 ±0.01 | 37.34 ±0.63 | 21.70 ±0.35 | *16.49* ±0.36 | 10.75 ±0.03 | **12.17** ±0.08 | 17.65 ±0.07 |
| COV$_{GREED}$ | 42.47 ±0.04 | 8.58 ±0.01 | 7.21 ±0.01 | 3.75 ±0.00 | 4.43 ±0.00 | 8.25 ±0.00 | 35.88 ±0.03 | 20.57 ±0.02 | 13.05 ±0.00 | 10.53 ±0.01 | 10.75 ±0.01 | 17.08 ±0.01 |
| COV$_{BCA}$ | 42.31 ±0.04 | 8.56 ±0.01 | 7.69 ±0.00 | 3.86 ±0.00 | 4.61 ±0.00 | 8.52 ±0.00 | 35.38 ±0.01 | 20.32 ±0.01 | 14.04 ±0.00 | 10.85 ±0.00 | 11.23 ±0.00 | *17.70* ±0.00 |

Table 7: Mean results with standard deviation of different inference strategies on @$k$ measures calculated with $k \in \{5, 10\}$, using probability estimates from LIGHTXML model. Notation: P—precision, R—recall, F1—F1-measure, Cov—Coverage. The green color indicates cells in which the strategy matches the metric. The best results are in **bold** and the second best are in *italic*.

| Inference strategy | Instance @5 P ±std | R ±std | Macro @5 P ±std | R ±std | F1 ±std | Cov ±std | Instance @10 P ±std | R ±std | Macro @10 P ±std | R ±std | F1 ±std | Cov ±std |
|---|---|---|---|---|---|---|---|---|---|---|---|---|
| **EURLEX-4K** | | | | | | | | | | | | |
| Top-K | **62.31** ±0.00 | **60.59** ±0.00 | 27.43 ±0.00 | 25.99 ±0.00 | 25.50 ±0.00 | 39.76 ±0.00 | **39.29** ±0.00 | **75.11** ±0.00 | 22.15 ±0.00 | 36.59 ±0.00 | 26.01 ±0.00 | 47.55 ±0.00 |
| PS-K | 60.77 ±0.00 | 59.17 ±0.00 | 29.10 ±0.00 | 29.89 ±0.00 | 28.17 ±0.00 | 44.03 ±0.00 | 38.98 ±0.00 | 74.54 ±0.00 | 21.36 ±0.00 | 38.27 ±0.00 | 25.78 ±0.00 | 49.47 ±0.00 |
| Pow-K $_{\beta=0.25}$ | 61.08 ±0.00 | 59.46 ±0.00 | 28.75 ±0.00 | 29.23 ±0.00 | 27.74 ±0.00 | 43.23 ±0.00 | 39.09 ±0.00 | 74.72 ±0.00 | 21.50 ±0.00 | 38.04 ±0.00 | 25.88 ±0.00 | 49.14 ±0.00 |
| Pow-K $_{\beta=0.5}$ | 58.53 ±0.00 | 57.10 ±0.00 | 28.88 ±0.00 | 31.25 ±0.00 | *28.53* ±0.00 | 45.48 ±0.00 | 38.48 ±0.00 | 73.66 ±0.00 | 20.77 ±0.00 | *38.85* ±0.00 | 25.43 ±0.00 | 50.13 ±0.00 |
| Log-K | *61.51* ±0.00 | *59.87* ±0.00 | 28.48 ±0.00 | 28.47 ±0.00 | 27.26 ±0.00 | 42.32 ±0.00 | *39.12* ±0.00 | *74.80* ±0.00 | 21.60 ±0.00 | 37.49 ±0.00 | 25.83 ±0.00 | 48.61 ±0.00 |
| Macro-P$_{GREED}$ | 41.75 ±0.28 | 40.82 ±0.27 | 28.97 ±0.14 | 27.42 ±0.34 | 24.89 ±0.24 | 46.94 ±0.18 | 25.52 ±0.14 | 49.39 ±0.25 | 24.62 ±0.10 | 31.80 ±0.24 | 23.44 ±0.14 | 49.64 ±0.21 |
| Macro-P$_{BCA}$ | 31.05 ±0.19 | 30.33 ±0.19 | **38.70** ±0.06 | 18.46 ±0.04 | 21.59 ±0.07 | 41.66 ±0.04 | 18.45 ±0.06 | 35.65 ±0.11 | **37.94** ±0.02 | 20.28 ±0.08 | 21.82 ±0.05 | 42.89 ±0.13 |
| Macro-R$_{PRIOR}$ | 48.36 ±0.00 | 47.42 ±0.00 | 27.08 ±0.00 | **32.82** ±0.00 | 27.36 ±0.00 | 48.58 ±0.00 | 35.00 ±0.00 | 67.34 ±0.00 | 19.74 ±0.00 | **39.64** ±0.00 | 24.28 ±0.00 | 51.64 ±0.00 |
| Macro-R$_{BCA}$ | 38.32 ±0.00 | 37.54 ±0.00 | 28.97 ±0.00 | *31.27* ±0.00 | 27.05 ±0.00 | 49.04 ±0.00 | 29.80 ±0.00 | 57.58 ±0.00 | 22.99 ±0.00 | 38.14 ±0.00 | 25.99 ±0.00 | 52.15 ±0.00 |
| Macro-F1$_{GREED}$ | 54.88 ±0.15 | 53.53 ±0.17 | 29.31 ±0.10 | 30.46 ±0.12 | 28.24 ±0.10 | 46.79 ±0.04 | 33.91 ±0.06 | 65.06 ±0.13 | 23.88 ±0.05 | 35.17 ±0.12 | *26.04* ±0.07 | 49.81 ±0.14 |
| Macro-F1$_{BCA}$ | 56.18 ±0.06 | 54.70 ±0.05 | 33.82 ±0.05 | 29.52 ±0.07 | **30.13** ±0.05 | 45.93 ±0.05 | 33.55 ±0.05 | 64.29 ±0.07 | 32.78 ±0.02 | 32.21 ±0.04 | **30.47** ±0.03 | 47.56 ±0.05 |
| Cov$_{GREED}$ | 18.81 ±0.11 | 18.19 ±0.10 | 23.48 ±0.17 | 25.24 ±0.16 | 17.84 ±0.11 | *49.54* ±0.19 | 11.02 ±0.03 | 21.23 ±0.09 | 18.14 ±0.06 | 29.62 ±0.13 | 15.25 ±0.07 | *52.65* ±0.19 |
| Cov$_{BCA}$ | 16.40 ±0.09 | 15.95 ±0.05 | 29.86 ±0.10 | 24.53 ±0.05 | 18.81 ±0.07 | **50.09** ±0.11 | 9.31 ±0.04 | 18.05 ±0.10 | 26.99 ±0.14 | 28.28 ±0.05 | 17.41 ±0.06 | **52.84** ±0.05 |
| **AMAZONCAT-13K** | | | | | | | | | | | | |
| Top-K | 68.01 ±0.00 | **79.01** ±0.00 | 33.92 ±0.00 | 31.34 ±0.00 | 29.28 ±0.00 | 52.60 ±0.00 | **41.72** ±0.00 | **89.41** ±0.00 | 24.63 ±0.00 | 51.76 ±0.00 | 28.84 ±0.00 | 68.61 ±0.00 |
| PS-K | 66.63 ±0.00 | 77.89 ±0.00 | 38.41 ±0.00 | 45.00 ±0.00 | 38.34 ±0.00 | 65.82 ±0.00 | *41.50* ±0.00 | *89.11* ±0.00 | 22.56 ±0.00 | 58.69 ±0.00 | 28.59 ±0.00 | 74.29 ±0.00 |
| Pow-K $_{\beta=0.25}$ | 63.59 ±0.00 | 75.08 ±0.00 | 36.74 ±0.00 | 45.96 ±0.00 | 38.15 ±0.00 | 65.55 ±0.00 | 40.68 ±0.00 | 88.17 ±0.00 | 21.96 ±0.00 | 58.50 ±0.00 | 28.41 ±0.00 | 73.36 ±0.00 |
| Pow-K $_{\beta=0.5}$ | 56.94 ±0.00 | 68.14 ±0.00 | 31.65 ±0.00 | 54.06 ±0.00 | 36.90 ±0.00 | 71.95 ±0.00 | 37.86 ±0.00 | 83.91 ±0.00 | 19.26 ±0.00 | 64.05 ±0.00 | 26.29 ±0.00 | 77.27 ±0.00 |
| Log-K | 64.07 ±0.00 | 75.21 ±0.00 | 37.40 ±0.00 | 41.86 ±0.00 | 36.78 ±0.00 | 61.84 ±0.00 | 40.66 ±0.00 | 88.05 ±0.00 | 23.09 ±0.00 | 55.85 ±0.00 | 28.91 ±0.00 | 71.37 ±0.00 |
| Macro-P$_{GREED}$ | 52.18 ±0.25 | 62.08 ±0.35 | 41.03 ±0.08 | 54.76 ±0.11 | 40.70 ±0.07 | 82.88 ±0.09 | 29.50 ±0.09 | 66.03 ±0.35 | 39.69 ±0.12 | 54.54 ±0.12 | 37.46 ±0.08 | 83.86 ±0.06 |
| Macro-P$_{BCA}$ | 41.53 ±0.12 | 49.66 ±0.08 | **63.81** ±0.02 | 30.43 ±0.03 | 35.99 ±0.01 | 76.75 ±0.01 | 25.32 ±0.08 | 57.33 ±0.11 | **61.84** ±0.00 | 33.06 ±0.04 | 35.08 ±0.04 | 78.17 ±0.05 |
| Macro-R$_{PRIOR}$ | 40.67 ±0.00 | 49.85 ±0.00 | 24.16 ±0.00 | *62.27* ±0.00 | 29.63 ±0.00 | 79.08 ±0.00 | 27.59 ±0.00 | 63.96 ±0.00 | 16.25 ±0.00 | *70.77* ±0.00 | 21.98 ±0.00 | 82.57 ±0.00 |
| Macro-R$_{BCA}$ | 38.93 ±0.00 | 48.26 ±0.00 | 25.17 ±0.00 | **65.47** ±0.00 | 30.37 ±0.00 | *82.91* ±0.00 | 26.50 ±0.00 | 61.99 ±0.00 | 17.53 ±0.00 | **72.63** ±0.00 | 23.22 ±0.00 | *84.84* ±0.00 |
| Macro-F1$_{GREED}$ | 59.22 ±0.02 | 70.41 ±0.04 | 38.27 ±0.08 | 56.80 ±0.08 | *42.54* ±0.07 | 80.23 ±0.15 | 37.67 ±0.03 | 82.12 ±0.00 | 36.35 ±0.10 | 60.84 ±0.04 | *41.34* ±0.10 | 83.50 ±0.04 |
| Macro-F1$_{BCA}$ | 60.70 ±0.01 | 71.93 ±0.01 | *50.89* ±0.02 | 52.32 ±0.03 | **49.48** ±0.02 | 79.63 ±0.03 | 37.80 ±0.00 | 82.24 ±0.01 | *49.43* ±0.01 | 55.17 ±0.02 | **49.21** ±0.01 | 81.39 ±0.03 |
| Cov$_{GREED}$ | 3.44 ±0.00 | 2.82 ±0.00 | 17.87 ±0.09 | 38.35 ±0.06 | 11.01 ±0.02 | 82.81 ±0.08 | 2.29 ±0.00 | 3.61 ±0.00 | 11.10 ±0.04 | 44.57 ±0.04 | 8.48 ±0.04 | 84.58 ±0.02 |
| Cov$_{BCA}$ | 3.20 ±0.00 | 2.63 ±0.00 | 29.40 ±0.03 | 39.05 ±0.02 | 14.23 ±0.02 | **84.39** ±0.01 | 2.13 ±0.00 | 3.36 ±0.00 | 19.02 ±0.06 | 44.28 ±0.04 | 10.74 ±0.01 | **85.40** ±0.02 |
| **WIKI-31K** | | | | | | | | | | | | |
| Top-K | **68.31** ±0.00 | **19.59** ±0.00 | 2.66 ±0.00 | 0.92 ±0.00 | 1.24 ±0.00 | 3.72 ±0.00 | **52.00** ±0.00 | **29.05** ±0.00 | 3.55 ±0.00 | 2.07 ±0.00 | 2.34 ±0.00 | 6.14 ±0.00 |
| PS-K | 62.65 ±0.00 | *18.07* ±0.00 | 4.21 ±0.00 | 2.14 ±0.00 | 2.56 ±0.00 | 6.54 ±0.00 | *50.16* ±0.00 | *28.20* ±0.00 | 4.64 ±0.00 | 3.62 ±0.00 | 3.61 ±0.00 | 9.01 ±0.00 |
| Pow-K $_{\beta=0.25}$ | 59.99 ±0.00 | 17.26 ±0.00 | 4.00 ±0.00 | 2.01 ±0.00 | 2.42 ±0.00 | 6.29 ±0.00 | 48.52 ±0.00 | 27.29 ±0.00 | 4.49 ±0.00 | 3.50 ±0.00 | 3.49 ±0.00 | 8.77 ±0.00 |
| Pow-K $_{\beta=0.5}$ | 50.12 ±0.00 | 14.40 ±0.00 | 4.83 ±0.00 | 3.13 ±0.00 | 3.37 ±0.00 | 8.53 ±0.00 | 40.15 ±0.00 | 22.57 ±0.00 | 5.07 ±0.00 | 5.10 ±0.00 | 4.47 ±0.00 | 11.37 ±0.00 |
| Log-K | 57.74 ±0.00 | 16.49 ±0.00 | 3.74 ±0.00 | 1.86 ±0.00 | 2.25 ±0.00 | 5.99 ±0.00 | 43.82 ±0.00 | 24.37 ±0.00 | 4.24 ±0.00 | 3.46 ±0.00 | 3.37 ±0.00 | 8.61 ±0.00 |
| Macro-P$_{GREED}$ | 31.36 ±0.27 | 8.96 ±0.07 | 7.26 ±0.07 | 3.93 ±0.05 | 4.39 ±0.05 | 11.93 ±0.08 | 27.42 ±0.78 | 15.45 ±0.43 | 7.07 ±0.06 | 5.34 ±0.05 | 5.14 ±0.03 | 13.99 ±0.09 |
| Macro-P$_{BCA}$ | 30.52 ±0.65 | 8.56 ±0.18 | **9.68** ±0.02 | 2.86 ±0.00 | 3.74 ±0.00 | 9.82 ±0.03 | 24.81 ±0.24 | 13.62 ±0.12 | **9.79** ±0.01 | 2.98 ±0.00 | 3.85 ±0.01 | 10.04 ±0.01 |
| Macro-R$_{PRIOR}$ | 31.32 ±0.00 | 8.97 ±0.00 | 5.78 ±0.00 | **5.02** ±0.00 | 4.53 ±0.00 | 11.63 ±0.00 | 24.23 ±0.00 | 13.66 ±0.00 | 5.55 ±0.00 | **7.38** ±0.00 | 5.31 ±0.00 | 14.78 ±0.00 |
| Macro-R$_{BCA}$ | 14.01 ±0.00 | 3.96 ±0.00 | 5.79 ±0.00 | *4.71* ±0.00 | 4.05 ±0.00 | 10.72 ±0.00 | 13.88 ±0.00 | 7.79 ±0.00 | 5.72 ±0.00 | *7.21* ±0.00 | 5.02 ±0.00 | *15.72* ±0.00 |
| Macro-F1$_{GREED}$ | 36.50 ±0.15 | 10.50 ±0.05 | 6.91 ±0.06 | 4.06 ±0.04 | *4.64* ±0.05 | 11.69 ±0.06 | 34.80 ±0.09 | 19.72 ±0.05 | 6.80 ±0.05 | 5.68 ±0.04 | *5.61* ±0.03 | 13.82 ±0.08 |
| Macro-F1$_{BCA}$ | 38.98 ±0.03 | 11.24 ±0.01 | *8.30* ±0.03 | 4.14 ±0.02 | **5.08** ±0.02 | 11.88 ±0.04 | 39.31 ±0.01 | 22.28 ±0.01 | *8.18* ±0.02 | 5.11 ±0.00 | **5.78** ±0.01 | 13.07 ±0.02 |
| Cov$_{GREED}$ | 21.49 ±0.10 | 6.04 ±0.02 | 5.28 ±0.04 | 4.00 ±0.03 | 3.71 ±0.02 | *12.23* ±0.02 | 14.77 ±0.04 | 8.20 ±0.02 | 5.03 ±0.05 | 6.22 ±0.03 | 4.40 ±0.04 | 15.32 ±0.04 |
| Cov$_{BCA}$ | 20.80 ±0.07 | 5.88 ±0.02 | 6.42 ±0.02 | 4.60 ±0.02 | 4.18 ±0.02 | **13.23** ±0.02 | 13.95 ±0.04 | 7.78 ±0.02 | 6.23 ±0.02 | 6.88 ±0.01 | 4.81 ±0.01 | **16.39** ±0.02 |
| **WIKIPEDIALARGE-500K** | | | | | | | | | | | | |
| Top-K | **43.12** ±0.00 | *54.28* ±0.00 | 20.51 ±0.00 | 25.86 ±0.00 | 19.84 ±0.00 | 40.62 ±0.00 | *27.01* ±0.00 | 63.13 ±0.00 | 17.16 ±0.00 | 34.79 ±0.00 | 19.57 ±0.00 | 50.15 ±0.00 |
| PS-K | *42.86* ±0.00 | **54.51** ±0.00 | 21.84 ±0.00 | 29.15 ±0.00 | 21.77 ±0.00 | 45.03 ±0.00 | **27.05** ±0.00 | **63.43** ±0.00 | 16.70 ±0.00 | 37.58 ±0.00 | 19.87 ±0.00 | 53.67 ±0.00 |
| Pow-K $_{\beta=0.25}$ | 42.40 ±0.00 | 54.07 ±0.00 | 21.43 ±0.00 | 28.04 ±0.00 | 21.20 ±0.00 | 43.54 ±0.00 | 26.94 ±0.00 | *63.27* ±0.00 | 16.89 ±0.00 | 36.56 ±0.00 | 19.81 ±0.00 | 52.36 ±0.00 |
| Pow-K $_{\beta=0.5}$ | 40.55 ±0.00 | 52.72 ±0.00 | 21.87 ±0.00 | 30.03 ±0.00 | 22.18 ±0.00 | 46.21 ±0.00 | 26.09 ±0.00 | 62.26 ±0.00 | 16.48 ±0.00 | 38.17 ±0.00 | 19.85 ±0.00 | 54.38 ±0.00 |
| Log-K | 42.66 ±0.00 | 54.09 ±0.00 | 20.97 ±0.00 | 26.81 ±0.00 | 20.49 ±0.00 | 41.86 ±0.00 | 26.97 ±0.00 | 63.19 ±0.00 | 17.08 ±0.00 | 35.51 ±0.00 | 19.69 ±0.00 | 51.04 ±0.00 |
| Macro-P$_{GREED}$ | 32.30 ±0.01 | 42.79 ±0.01 | 24.02 ±0.01 | 31.48 ±0.01 | 24.10 ±0.01 | 52.94 ±0.01 | 14.34 ±0.01 | 37.04 ±0.03 | 22.11 ±0.01 | 32.04 ±0.02 | 21.95 ±0.01 | 56.11 ±0.02 |
| Macro-P$_{BCA}$ | 16.25 ±0.06 | 22.54 ±0.05 | **37.88** ±0.00 | 21.12 ±0.00 | 24.10 ±0.00 | 46.23 ±0.00 | 8.48 ±0.01 | 22.97 ±0.02 | **37.77** ±0.00 | 21.46 ±0.00 | *24.19* ±0.00 | 46.65 ±0.00 |
| Macro-R$_{PRIOR}$ | 35.80 ±0.00 | 48.94 ±0.00 | 21.81 ±0.00 | *33.03* ±0.00 | 22.95 ±0.00 | 50.38 ±0.00 | 23.09 ±0.00 | 57.82 ±0.00 | 15.65 ±0.00 | **40.65** ±0.00 | 19.51 ±0.00 | 57.65 ±0.00 |
| Macro-R$_{BCA}$ | 33.58 ±0.00 | 47.34 ±0.00 | 22.08 ±0.00 | **33.56** ±0.00 | 23.63 ±0.00 | 51.91 ±0.00 | 21.81 ±0.00 | 56.27 ±0.00 | 15.92 ±0.00 | **40.97** ±0.00 | 20.07 ±0.00 | 58.42 ±0.00 |
| Macro-F1$_{GREED}$ | 37.25 ±0.01 | 48.36 ±0.01 | 23.49 ±0.02 | 31.47 ±0.02 | 24.01 ±0.01 | 51.32 ±0.02 | 20.96 ±0.00 | 49.32 ±0.01 | 21.61 ±0.00 | 34.15 ±0.03 | 22.94 ±0.01 | 56.67 ±0.03 |
| Macro-F1$_{BCA}$ | 32.96 ±0.01 | 41.21 ±0.00 | *35.79* ±0.00 | 26.19 ±0.00 | **27.64** ±0.00 | 49.20 ±0.00 | 19.32 ±0.00 | 44.17 ±0.00 | *35.55* ±0.00 | 27.43 ±0.00 | **28.15** ±0.00 | 50.79 ±0.00 |
| Cov$_{GREED}$ | 20.75 ±0.01 | 29.49 ±0.01 | 20.54 ±0.01 | 31.08 ±0.02 | 20.33 ±0.01 | *52.98* ±0.01 | 12.57 ±0.00 | 34.15 ±0.01 | 15.11 ±0.01 | 37.95 ±0.01 | 17.19 ±0.01 | *59.17* ±0.02 |
| Cov$_{BCA}$ | 19.46 ±0.00 | 28.06 ±0.00 | 23.14 ±0.01 | 32.13 ±0.00 | 21.08 ±0.00 | **55.40** ±0.01 | 11.59 ±0.00 | 32.09 ±0.00 | 18.45 ±0.00 | 38.56 ±0.00 | 18.08 ±0.00 | **61.15** ±0.00 |
| **AMAZON-670K** | | | | | | | | | | | | |
| Top-K | 37.71 ±0.00 | 35.23 ±0.00 | 14.26 ±0.00 | 15.16 ±0.00 | 13.76 ±0.00 | 20.41 ±0.00 | **25.35** ±0.00 | 46.74 ±0.00 | 14.83 ±0.00 | 21.56 ±0.00 | 16.20 ±0.00 | 26.85 ±0.00 |
| PS-K | 37.54 ±0.00 | 35.13 ±0.00 | 14.86 ±0.00 | 15.73 ±0.00 | 14.31 ±0.00 | 21.16 ±0.00 | 25.28 ±0.00 | 46.65 ±0.00 | 14.89 ±0.00 | 21.85 ±0.00 | 16.34 ±0.00 | 27.22 ±0.00 |
| Pow-K $_{\beta=0.25}$ | 37.64 ±0.00 | 35.21 ±0.00 | 14.71 ±0.00 | 15.58 ±0.00 | 14.17 ±0.00 | 20.95 ±0.00 | 25.33 ±0.00 | 46.72 ±0.00 | 14.88 ±0.00 | 21.76 ±0.00 | 16.30 ±0.00 | 27.09 ±0.00 |
| Pow-K $_{\beta=0.5}$ | 37.47 ±0.00 | 35.07 ±0.00 | 14.98 ±0.00 | 15.84 ±0.00 | 14.42 ±0.00 | 21.30 ±0.00 | 25.24 ±0.00 | 46.60 ±0.00 | 14.90 ±0.00 | 21.92 ±0.00 | 16.36 ±0.00 | 27.30 ±0.00 |
| Log-K | *37.69* ±0.00 | *35.22* ±0.00 | 14.48 ±0.00 | 15.37 ±0.00 | 13.97 ±0.00 | 20.68 ±0.00 | *25.35* ±0.00 | **46.75** ±0.00 | 14.85 ±0.00 | 21.64 ±0.00 | 16.24 ±0.00 | 26.94 ±0.00 |
| Macro-P$_{GREED}$ | 34.06 ±0.03 | 31.94 ±0.02 | 17.00 ±0.01 | 15.67 ±0.01 | 15.27 ±0.01 | 22.39 ±0.01 | 19.91 ±0.02 | 37.00 ±0.05 | 16.65 ±0.01 | 19.64 ±0.01 | 16.47 ±0.01 | 26.76 ±0.01 |
| Macro-P$_{BCA}$ | 27.52 ±1.78 | 26.09 ±1.58 | **21.09** ±0.33 | 14.38 ±0.44 | *15.96* ±0.29 | 21.96 ±0.29 | 15.02 ±0.00 | 28.23 ±0.00 | **23.21** ±0.00 | 16.36 ±0.00 | *17.90* ±0.00 | 24.13 ±0.00 |
| Macro-R$_{PRIOR}$ | 36.78 ±0.00 | 34.50 ±0.00 | 15.44 ±0.00 | *16.23* ±0.00 | 14.76 ±0.00 | 21.84 ±0.00 | 24.93 ±0.00 | 46.08 ±0.00 | 15.02 ±0.00 | *22.12* ±0.00 | 16.46 ±0.00 | 27.60 ±0.00 |
| Macro-R$_{BCA}$ | 36.43 ±0.49 | 34.19 ±0.41 | 16.48 ±0.62 | **16.35** ±0.21 | 15.40 ±0.39 | 22.61 ±0.47 | 24.72 ±0.00 | 45.70 ±0.00 | 16.30 ±0.00 | **22.23** ±0.00 | 17.46 ±0.00 | *28.08* ±0.00 |
| Macro-F1$_{GREED}$ | 35.27 ±0.01 | 33.04 ±0.01 | 16.72 ±0.01 | 15.76 ±0.01 | 15.24 ±0.01 | 22.22 ±0.01 | 21.24 ±0.02 | 39.33 ±0.03 | 16.70 ±0.01 | 20.17 ±0.01 | 16.85 ±0.01 | 26.96 ±0.01 |
| Macro-F1$_{BCA}$ | 31.97 ±0.78 | 30.04 ±0.71 | *20.39* ±0.37 | 15.15 ±0.23 | **16.37** ±0.02 | 22.25 ±0.15 | 18.48 ±0.00 | 34.28 ±0.00 | 22.79 ±0.00 | 17.95 ±0.00 | **18.89** ±0.00 | 25.12 ±0.00 |
| Cov$_{GREED}$ | 31.06 ±0.01 | 29.05 ±0.02 | 15.40 ±0.01 | 15.67 ±0.00 | 14.35 ±0.00 | 22.31 ±0.01 | 20.87 ±0.00 | 38.61 ±0.00 | 15.10 ±0.01 | 21.48 ±0.01 | 16.12 ±0.01 | *27.77* ±0.00 |
| Cov$_{BCA}$ | 30.27 ±0.01 | 28.39 ±0.00 | 16.44 ±0.00 | 15.94 ±0.00 | 14.85 ±0.00 | **22.93** ±0.00 | 20.08 ±0.00 | 37.23 ±0.00 | 16.20 ±0.00 | 21.52 ±0.00 | 16.61 ±0.00 | **28.15** ±0.00 |

Table 8: Results of different inference strategies on @$k$ measures calculated with $k \in \{1, 3, 5, 10\}$, using true labels as predictions. Notation: P—precision, R—recall, F1—F1-measure. The green color indicates cells in which the strategy matches the metric. The best results are in **bold** and the second best are in *italic*.

| Inference strategy | Instance @1 P | R | Macro @1 P | R | F1 | Instance @3 P | R | Macro @3 P | R | F1 | Instance @5 P | R | Macro @5 P | R | F1 | Instance @10 P | R | Macro @10 P | R | F1 |
|---|---|---|---|---|---|---|---|---|---|---|---|---|---|---|---|---|---|---|---|---|
| | | | | | | | | | EURLEX-4K | | | | | | | | | | | |
| WEIGHTED-TOP-K | **100.00** | 20.53 | 50.05 | 33.19 | 36.83 | **99.12** | **60.06** | 65.95 | 58.09 | 60.43 | **92.58** | **89.95** | **66.43** | 65.11 | 65.66 | **52.93** | **99.96** | **66.43** | **66.42** | **66.42** |
| MACRO-P$_{\text{GREED}}$ | **100.00** | 20.53 | 55.47 | 28.26 | 33.76 | **99.12** | **60.06** | 65.54 | 56.65 | 59.69 | **92.58** | **89.95** | 66.36 | 64.93 | 65.54 | **52.93** | **99.96** | **66.43** | **66.42** | **66.42** |
| MACRO-P$_{\text{BCA}}$ | **100.00** | 20.53 | **60.13** | 31.89 | 37.57 | **99.12** | **60.06** | **66.35** | 58.33 | *61.03* | **92.58** | **89.95** | **66.43** | 65.18 | *65.71* | **52.93** | **99.96** | **66.43** | **66.42** | **66.42** |
| MACRO-R$_{\text{PRIOR}}$ | **100.00** | 20.53 | 50.05 | *33.19* | 36.83 | **99.12** | **60.06** | 65.95 | 58.09 | 60.43 | **92.58** | **89.95** | **66.43** | 65.11 | 65.66 | **52.93** | **99.96** | **66.43** | **66.42** | **66.42** |
| MACRO-R$_{\text{BCA}}$ | **100.00** | 20.53 | 53.01 | **35.24** | *39.11* | **99.12** | **60.06** | 66.34 | **58.69** | 61.00 | **92.58** | **89.95** | **66.43** | **65.19** | 65.71 | **52.93** | **99.96** | **66.43** | **66.42** | **66.42** |
| MACRO-F1$_{\text{GREED}}$ | **100.00** | 20.53 | 55.17 | 31.35 | *36.27* | **99.12** | **60.06** | 65.64 | 57.17 | 59.99 | **92.58** | **89.95** | 66.36 | 64.94 | 65.55 | **52.93** | **99.96** | **66.43** | **66.42** | **66.42** |
| MACRO-F1$_{\text{BCA}}$ | **100.00** | 20.53 | *57.23* | *34.84* | **39.65** | **99.12** | **60.06** | *66.35* | *58.57* | **61.12** | **92.58** | **89.95** | **66.43** | *65.18* | **65.71** | **52.93** | **99.96** | **66.43** | **66.42** | **66.42** |
| | | | | | | | | | AMAZONCAT-13K | | | | | | | | | | | |
| WEIGHTED-TOP-K | **100.00** | 29.06 | 93.05 | *74.51* | 78.21 | **91.73** | **69.95** | **99.58** | 95.56 | 96.87 | **77.27** | **88.81** | **99.58** | 98.37 | 98.82 | **47.43** | **97.80** | **99.58** | *99.44* | 99.49 |
| MACRO-P$_{\text{GREED}}$ | **100.00** | 29.06 | 98.40 | 70.10 | 77.46 | **91.73** | **69.95** | 99.55 | 95.27 | 96.78 | **77.27** | **88.81** | **99.58** | 98.31 | 98.80 | **47.43** | **97.80** | **99.58** | 99.44 | 99.49 |
| MACRO-P$_{\text{BCA}}$ | **100.00** | 29.06 | **99.14** | 72.14 | *78.95* | **91.73** | **69.95** | **99.58** | 95.48 | *96.90* | **77.27** | **88.81** | **99.58** | 98.36 | *98.83* | **47.43** | **97.80** | **99.58** | 99.44 | 99.49 |
| MACRO-R$_{\text{PRIOR}}$ | **100.00** | 29.06 | 93.05 | *74.51* | 78.21 | **91.73** | **69.95** | **99.58** | *95.56* | 96.87 | **77.27** | **88.81** | **99.58** | **98.37** | 98.82 | **47.43** | **97.80** | **99.58** | *99.44* | 99.49 |
| MACRO-R$_{\text{BCA}}$ | **100.00** | 29.06 | 94.52 | **74.75** | 78.72 | **91.73** | **69.95** | **99.58** | **95.58** | 96.89 | **77.27** | **88.81** | **99.58** | **98.37** | 98.82 | **47.43** | **97.80** | **99.58** | 99.43 | 99.49 |
| MACRO-F1$_{\text{GREED}}$ | **100.00** | 29.06 | 98.29 | 72.53 | 78.68 | **91.73** | **69.95** | 99.56 | 95.38 | 96.82 | **77.27** | **88.81** | 99.58 | 98.32 | 98.80 | **47.43** | **97.80** | **99.58** | 99.43 | 99.49 |
| MACRO-F1$_{\text{BCA}}$ | **100.00** | 29.06 | 98.38 | 74.03 | **79.68** | **91.73** | **69.95** | **99.58** | 95.55 | **96.92** | **77.27** | **88.81** | **99.58** | 98.36 | **98.83** | **47.43** | **97.80** | **99.58** | 99.43 | **99.49** |
| | | | | | | | | | WIKI-31K | | | | | | | | | | | |
| WEIGHTED-TOP-K | **100.00** | 6.19 | 17.43 | 13.91 | 14.84 | **99.99** | **18.55** | 47.67 | 37.75 | 39.94 | **99.93** | **30.82** | 64.23 | 52.71 | 55.27 | **98.10** | **58.83** | 71.10 | 67.39 | 68.56 |
| MACRO-P$_{\text{GREED}}$ | **100.00** | 6.19 | 21.35 | 10.62 | 12.43 | **99.99** | **18.55** | 53.77 | 32.17 | 36.77 | **99.93** | **30.82** | 65.64 | 49.18 | 53.46 | **98.10** | **58.83** | **70.75** | 66.52 | 67.96 |
| MACRO-P$_{\text{BCA}}$ | **100.00** | 6.19 | **21.38** | 10.65 | 12.46 | **99.99** | **18.55** | **61.79** | 37.54 | 42.47 | **99.93** | **30.82** | **70.39** | 54.16 | 58.24 | **98.10** | **58.83** | *71.26* | 67.68 | 68.87 |
| MACRO-R$_{\text{PRIOR}}$ | **100.00** | 6.19 | 17.43 | 13.91 | 14.84 | **99.99** | **18.55** | 47.67 | *37.75* | 39.94 | **99.93** | **30.82** | 64.23 | *52.71* | 55.27 | **98.10** | **58.83** | 71.10 | 67.39 | 68.56 |
| MACRO-R$_{\text{BCA}}$ | **100.00** | 6.19 | 21.06 | **18.28** | *19.02* | **99.99** | **18.55** | 54.62 | **42.65** | *45.41* | **99.93** | **30.82** | 68.59 | **55.60** | *58.55* | **98.10** | **58.83** | 71.26 | **67.74** | 68.86 |
| MACRO-F1$_{\text{GREED}}$ | **100.00** | 6.19 | 21.20 | 14.60 | 16.09 | **99.99** | **18.55** | 54.49 | 37.19 | 40.99 | **99.93** | **30.82** | 66.47 | 51.53 | 55.19 | **98.10** | **58.83** | 70.80 | 66.73 | 68.09 |
| MACRO-F1$_{\text{BCA}}$ | **100.00** | 6.19 | *21.36* | *18.26* | **19.09** | **99.99** | **18.55** | *59.04* | *42.23* | **46.11** | **99.93** | **30.82** | *70.10* | *55.37* | **58.86** | **98.10** | **58.83** | **71.27** | *67.72* | **68.89** |
| | | | | | | | | | WIKIPEDIALARGE-500K | | | | | | | | | | | |
| WEIGHTED-TOP-K | **100.00** | 37.10 | 72.36 | 49.26 | 54.43 | **84.40** | **73.79** | 96.74 | 83.98 | 87.72 | **68.87** | **87.71** | 98.99 | 93.53 | 95.42 | **44.11** | **97.48** | 99.52 | 98.49 | 98.90 |
| MACRO-P$_{\text{GREED}}$ | **100.00** | 37.10 | 81.14 | 44.25 | 52.36 | **84.40** | **73.79** | 97.45 | 82.22 | 87.17 | **68.87** | **87.71** | 99.06 | 92.92 | 95.19 | **44.11** | **97.48** | **99.55** | 98.40 | 98.86 |
| MACRO-P$_{\text{BCA}}$ | **100.00** | 37.10 | **87.70** | 48.46 | 57.03 | **84.40** | **73.79** | **99.19** | 85.09 | *89.67* | **68.87** | **87.71** | **99.53** | 94.04 | *96.07* | **44.11** | **97.48** | **99.61** | 98.58 | 98.99 |
| MACRO-R$_{\text{PRIOR}}$ | **100.00** | 37.10 | 72.36 | 49.26 | 54.43 | **84.40** | **73.79** | 96.74 | 83.98 | 87.72 | **68.87** | **87.71** | 98.99 | 93.53 | 95.42 | **44.11** | **97.48** | 99.52 | 98.49 | 98.90 |
| MACRO-R$_{\text{BCA}}$ | **100.00** | 37.10 | 78.55 | **53.86** | *59.39* | **84.40** | **73.79** | 98.52 | **85.87** | 89.60 | **68.87** | **87.71** | 99.42 | **94.22** | 96.04 | **44.11** | **97.48** | 99.58 | **98.60** | 98.99 |
| MACRO-F1$_{\text{GREED}}$ | **100.00** | 37.10 | 80.53 | 48.26 | 55.46 | **84.40** | **73.79** | 97.63 | 83.29 | 87.80 | **68.87** | **87.71** | 99.08 | 93.19 | 95.33 | **44.11** | **97.48** | 99.54 | 98.43 | 98.88 |
| MACRO-F1$_{\text{BCA}}$ | **100.00** | 37.10 | *85.18* | *53.26* | **60.45** | **84.40** | **73.79** | *99.08* | *85.69* | **89.88** | **68.87** | **87.71** | *99.49* | *94.17* | **96.11** | **44.11** | **97.48** | *99.59* | *98.60* | **99.00** |
| | | | | | | | | | AMAZON-670K | | | | | | | | | | | |
| WEIGHTED-TOP-K | **100.00** | 25.99 | 17.51 | 14.64 | 15.41 | **93.39** | **60.62** | 40.14 | 35.43 | 36.71 | **87.65** | **87.65** | 49.91 | 47.71 | 48.41 | **51.70** | **100.00** | 51.81 | 51.81 | 51.81 |
| MACRO-P$_{\text{GREED}}$ | **100.00** | 25.99 | 21.37 | 12.72 | 14.61 | **93.39** | **60.62** | 43.83 | 34.21 | 37.10 | **87.65** | **87.65** | 50.45 | 47.56 | 48.63 | **51.70** | **100.00** | 51.81 | 51.81 | 51.81 |
| MACRO-P$_{\text{BCA}}$ | **100.00** | 25.99 | **21.84** | 12.96 | 14.88 | **93.39** | **60.62** | *46.14* | 35.91 | 38.92 | **87.65** | **87.65** | *51.13* | 48.27 | *49.33* | **51.70** | **100.00** | 51.81 | 51.81 | 51.81 |
| MACRO-R$_{\text{PRIOR}}$ | **100.00** | 25.99 | 17.51 | 14.64 | 15.41 | **93.39** | **60.62** | 40.14 | 35.43 | 36.71 | **87.65** | **87.65** | 49.91 | 47.71 | 48.41 | **51.70** | **100.00** | 51.81 | **51.81** | 51.81 |
| MACRO-R$_{\text{BCA}}$ | **100.00** | 25.99 | 19.58 | **16.63** | *17.43* | **93.39** | **60.62** | 43.45 | **37.54** | *39.18* | **87.65** | **87.65** | 50.88 | **48.44** | 49.23 | **51.70** | **100.00** | 51.81 | 51.81 | 51.81 |
| MACRO-F1$_{\text{GREED}}$ | **100.00** | 25.99 | 21.15 | 14.45 | 16.03 | **93.39** | **60.62** | 44.10 | 35.16 | *37.71* | **87.65** | **87.65** | 50.46 | 47.65 | 48.67 | **51.70** | **100.00** | 51.81 | 51.81 | 51.81 |
| MACRO-F1$_{\text{BCA}}$ | **100.00** | 25.99 | *21.41* | *16.50* | **17.73** | **93.39** | **60.62** | **46.17** | *37.24* | **39.70** | **87.65** | **87.65** | **51.14** | *48.38* | **49.36** | **51.70** | **100.00** | 51.81 | 51.81 | **51.81** |

Another factor that impacts the running time is stopping criterion $\epsilon$, which indicates minimal expected utility gain to continue BCA algorithm. Here we test and report results for 3 values of $\epsilon \in \{10^{-3}, 10^{-5}, 10^{-7}\}$ on AMAZONCAT-13K, WIKIPEDIALARGE-500K, and AMAZON-670K datasets that are the largest in terms of number of samples and labels.

The results of BCA methods with different values of $\epsilon$ and $k'$ are presented in Table 9. In this table, we additionally report the number of iterations (number of passes over datasets performed by the BCA algorithm) and time in seconds. The numbers are the mean results over 5 runs with different seeds. For all values of $\epsilon$ and $k'$, the results are very similar. In the case of WIKIPEDIALARGE-500K dataset, $k' = 1000$ achieves slightly better results than $k' = 100$ in a few cases. In the case of all datasets, the smallest $\epsilon = 10^{-7}$ is usually the best. However, even the largest tested $\epsilon = 10^{-3}$ that is greater than the inverse of a number of labels and samples is only slightly worse at the same time, requiring a much smaller number of iterations. For all values of $\epsilon$ and $k'$, the BCA algorithm terminates in just a few iterations, finishing always in less than 20 minutes. Please note that we implemented our algorithms in Python with some parts optimized using Numba [24] – LLVM-based just-in-time (JIT) compiler for Python. Because of that, we believe that the running time can be further reduced by using a more performant programming language.

### E.5 Results with mixed utilities

As we already discussed in Section 7, the optimization of macro-measures comes with the cost of a significant drop in performance on instance-wise measures, which in some cases may not be acceptable. To achieve the desired trade-off between tail and head label performance, one can optimize

Table 9: Impact of different values of $k'$ and $\epsilon$ on the results of @k measures calculated with $k \in \{3, 5\}$. Notation: P—precision, R—recall, F1—F1-measure, Cov—Coverage, I—number of iterations, T—time in seconds. The green color indicates cells in which the strategy matches the metric. The best results are in **bold** and the second best are in *italic*.

| Inference strategy | Instance @3 P | R | Macro @3 P | R | F1 | Cov | It./Time @3 I | T | Instance @5 P | R | Macro @5 P | R | F1 | Cov | It./Time @5 I | T |
|---|---|---|---|---|---|---|---|---|---|---|---|---|---|---|---|---|
| **AMAZONCAT-13K, $k'=100$** | | | | | | | | | | | | | | | | |
| Macro-P$_{BCA}$, $\epsilon=10^{-3}$ | 57.62 | 43.92 | 64.22 | 29.28 | 35.79 | 76.22 | 5.00 | 127.13 | 43.08 | 51.52 | 63.73 | 30.55 | 36.05 | 76.83 | 5.00 | 127.90 |
| Macro-P$_{BCA}$, $\epsilon=10^{-5}$ | 55.95 | 42.55 | *64.25* | 29.25 | 35.76 | 76.20 | 7.00 | 171.43 | 41.70 | 49.91 | *63.79* | 30.44 | 35.98 | 76.77 | 8.40 | 208.17 |
| Macro-P$_{BCA}$, $\epsilon=10^{-7}$ | 54.97 | 41.61 | **64.27** | 29.22 | 35.76 | 76.18 | 13.80 | 334.67 | 41.53 | 49.66 | **63.81** | 30.43 | 35.99 | 76.75 | 16.40 | 400.34 |
| Macro-F1$_{BCA}$, $\epsilon=10^{-3}$ | 70.60 | 53.84 | 51.90 | **48.23** | 47.91 | 77.82 | 3.00 | 91.22 | **60.70** | **71.93** | 50.78 | **52.40** | 49.44 | 79.72 | 3.00 | 91.50 |
| Macro-F1$_{BCA}$, $\epsilon=10^{-5}$ | *70.61* | **53.86** | 51.95 | 48.22 | *47.92* | 77.82 | 4.60 | 134.85 | *60.70* | *71.93* | 50.89 | *52.32* | **49.48** | 79.64 | 5.00 | 147.03 |
| Macro-F1$_{BCA}$, $\epsilon=10^{-7}$ | **70.61** | *53.86* | 51.95 | *48.22* | **47.93** | 77.83 | 6.80 | 201.24 | 60.70 | 71.93 | 50.89 | 52.32 | *49.48* | 79.63 | 8.20 | 248.82 |
| Cov$_{BCA}$, $\epsilon=10^{-3}$ | 4.55 | 2.30 | 34.23 | 35.20 | 15.87 | 82.65 | 3.00 | 21.35 | 3.21 | 2.64 | 28.61 | 39.11 | 14.18 | 84.39 | 3.00 | 21.75 |
| Cov$_{BCA}$, $\epsilon=10^{-5}$ | 4.53 | 2.29 | 34.83 | 35.16 | 15.90 | *82.67* | 5.20 | 34.64 | 3.20 | 2.63 | 29.30 | 39.05 | 14.23 | **84.39** | 5.00 | 33.54 |
| Cov$_{BCA}$, $\epsilon=10^{-7}$ | 4.53 | 2.29 | 34.93 | 35.16 | 15.91 | **82.67** | 8.60 | 57.96 | 3.20 | 2.63 | 29.40 | 39.05 | 14.23 | *84.39* | 8.20 | 54.86 |
| **WIKIPEDIALARGE-500K, $k'=100$** | | | | | | | | | | | | | | | | |
| Macro-P$_{BCA}$, $\epsilon=10^{-3}$ | 31.25 | 25.50 | 37.43 | 20.77 | 23.36 | 45.93 | 4.00 | 261.61 | 22.30 | 28.10 | 37.23 | 22.44 | 23.82 | 48.01 | 4.00 | 264.13 |
| Macro-P$_{BCA}$, $\epsilon=10^{-5}$ | 31.12 | 25.43 | *37.48* | 20.74 | 23.37 | 45.88 | 6.00 | 355.70 | 22.17 | 27.98 | *37.29* | 22.38 | 23.84 | 47.92 | 6.00 | 359.58 |
| Macro-P$_{BCA}$, $\epsilon=10^{-7}$ | 31.09 | 25.40 | **37.50** | 20.72 | 23.37 | 45.85 | 15.20 | 928.40 | 22.14 | 27.95 | **37.33** | 22.36 | 23.85 | 47.88 | 19.60 | 1198.52 |
| Macro-F1$_{BCA}$, $\epsilon=10^{-3}$ | **44.28** | **36.70** | 35.32 | 23.92 | 25.92 | 46.72 | 3.00 | 234.15 | **33.51** | **41.86** | 35.31 | 26.93 | 27.30 | 50.24 | 3.00 | 235.64 |
| Macro-F1$_{BCA}$, $\epsilon=10^{-5}$ | *44.28* | *36.69* | 35.40 | 23.89 | *25.95* | 46.68 | 5.00 | 358.02 | *33.50* | *41.84* | 35.42 | 26.87 | *27.33* | 50.14 | 5.00 | 361.94 |
| Macro-F1$_{BCA}$, $\epsilon=10^{-7}$ | 44.28 | 36.69 | 35.42 | 23.89 | **25.96** | 46.67 | 10.60 | 764.05 | 33.50 | 41.84 | 35.45 | 26.86 | **27.35** | 50.12 | 14.40 | 1040.98 |
| Cov$_{BCA}$, $\epsilon=10^{-3}$ | 27.56 | 24.71 | 25.95 | **26.86** | 21.68 | 50.16 | 3.00 | 54.30 | 19.66 | 28.26 | 23.17 | **32.29** | 21.19 | 55.45 | 3.00 | 55.61 |
| Cov$_{BCA}$, $\epsilon=10^{-5}$ | 27.48 | 24.65 | 25.98 | 26.85 | 21.67 | *50.18* | 6.00 | 95.63 | 19.60 | 28.19 | 23.19 | 32.28 | 21.17 | *55.46* | 5.00 | 79.93 |
| Cov$_{BCA}$, $\epsilon=10^{-7}$ | 27.48 | 24.65 | 25.98 | 26.85 | 21.66 | **50.19** | 9.00 | 143.27 | 19.60 | 28.19 | 23.20 | *32.28* | 21.17 | **55.47** | 9.00 | 142.02 |
| **WIKIPEDIALARGE-500K, $k'=1000$** | | | | | | | | | | | | | | | | |
| Macro-P$_{BCA}$, $\epsilon=10^{-3}$ | 25.26 | 21.87 | 37.65 | 20.21 | 23.43 | 45.12 | 4.00 | 447.43 | 16.33 | 22.62 | 37.86 | 21.16 | 24.09 | 46.29 | 4.00 | 448.26 |
| Macro-P$_{BCA}$, $\epsilon=10^{-5}$ | 25.19 | 21.81 | *37.69* | 20.18 | 23.44 | 45.08 | 6.00 | 597.66 | 16.26 | 22.54 | *37.87* | 21.13 | 24.09 | 46.24 | 5.20 | 550.41 |
| Macro-P$_{BCA}$, $\epsilon=10^{-7}$ | 25.19 | 21.81 | **37.69** | 20.18 | 23.44 | 45.08 | 9.60 | 971.00 | 16.25 | 22.54 | **37.88** | 21.12 | 24.10 | 46.23 | 10.00 | 1063.38 |
| Macro-F1$_{BCA}$, $\epsilon=10^{-3}$ | **43.83** | **36.41** | 35.30 | 23.73 | 25.97 | 46.41 | 3.00 | 386.38 | **32.97** | **41.23** | 35.67 | 26.27 | 27.61 | 49.32 | 3.00 | 388.71 |
| Macro-F1$_{BCA}$, $\epsilon=10^{-5}$ | *43.82* | *36.40* | 35.41 | 23.69 | *26.01* | 46.36 | 5.00 | 604.95 | 32.96 | 41.20 | 35.78 | 26.19 | *27.63* | 49.20 | 5.00 | 627.05 |
| Macro-F1$_{BCA}$, $\epsilon=10^{-7}$ | 43.82 | 36.40 | 35.42 | 23.69 | **26.01** | 46.36 | 8.80 | 1041.18 | *32.96* | *41.21* | 35.79 | 26.19 | **27.64** | 49.20 | 9.60 | 1168.04 |
| Cov$_{BCA}$, $\epsilon=10^{-3}$ | 27.40 | 24.61 | 25.89 | **26.76** | 21.63 | 50.13 | 3.00 | 102.62 | 19.53 | 28.13 | 23.11 | **32.14** | 21.11 | 55.38 | 3.00 | 103.42 |
| Cov$_{BCA}$, $\epsilon=10^{-5}$ | 27.31 | 24.55 | 25.92 | 26.76 | 21.60 | *50.16* | 6.00 | 186.13 | 19.46 | 28.06 | 23.13 | 32.13 | 21.08 | *55.39* | 5.00 | 160.21 |
| Cov$_{BCA}$, $\epsilon=10^{-7}$ | 27.31 | 24.55 | 25.92 | 26.76 | 21.60 | 50.16 | 9.00 | 268.57 | 19.46 | 28.06 | 23.14 | *32.13* | 21.08 | **55.40** | 8.60 | 261.12 |
| **AMAZON-670K, $k'=100$** | | | | | | | | | | | | | | | | |
| Macro-P$_{BCA}$, $\epsilon=10^{-3}$ | 33.13 | 19.38 | 17.45 | 10.50 | 12.11 | **17.76** | 4.00 | 53.67 | 26.79 | 25.44 | 21.21 | 14.22 | 15.84 | 21.88 | 4.00 | 53.32 |
| Macro-P$_{BCA}$, $\epsilon=10^{-5}$ | 33.12 | 19.38 | *17.46* | 10.49 | 12.11 | *17.76* | 6.00 | 70.77 | 26.78 | 25.44 | *21.21* | 14.22 | 15.85 | 21.88 | 5.00 | 59.85 |
| Macro-P$_{BCA}$, $\epsilon=10^{-7}$ | 33.12 | 19.38 | **17.46** | 10.49 | 12.11 | 17.76 | 7.60 | 89.95 | 26.77 | 25.44 | **21.22** | 14.22 | 15.85 | 21.88 | 8.40 | 100.14 |
| Macro-F1$_{BCA}$, $\epsilon=10^{-3}$ | **37.05** | **21.54** | 16.65 | 10.75 | 12.21 | 17.69 | 3.00 | 46.75 | **31.64** | **29.75** | 20.53 | 15.06 | 16.35 | 22.21 | 3.00 | 46.34 |
| Macro-F1$_{BCA}$, $\epsilon=10^{-5}$ | *37.04* | *21.53* | 16.67 | 10.75 | **12.22** | 17.69 | 5.00 | 71.70 | 31.63 | 29.73 | 20.55 | 15.06 | *16.36* | 22.21 | 5.00 | 72.32 |
| Macro-F1$_{BCA}$, $\epsilon=10^{-7}$ | 37.04 | 21.53 | 16.67 | 10.75 | *12.22* | 17.69 | 7.60 | 109.27 | *31.63* | *29.73* | 20.55 | 15.06 | **16.36** | 22.21 | 7.00 | 100.73 |
| Cov$_{BCA}$, $\epsilon=10^{-3}$ | 35.40 | 20.32 | 14.04 | 10.84 | 11.23 | 17.69 | 3.00 | 11.11 | 30.30 | 28.40 | 16.43 | 15.94 | 14.85 | **22.93** | 3.00 | 10.93 |
| Cov$_{BCA}$, $\epsilon=10^{-5}$ | 35.38 | 20.31 | 14.04 | *10.85* | 11.23 | 17.70 | 5.00 | 15.67 | 30.28 | 28.38 | 16.44 | **15.94** | 14.85 | *22.93* | 5.00 | 16.06 |
| Cov$_{BCA}$, $\epsilon=10^{-7}$ | 35.38 | 20.31 | 14.04 | **10.85** | 11.23 | 17.70 | 7.20 | 22.30 | 30.28 | 28.38 | 16.44 | *15.94* | 14.85 | *22.93* | 6.60 | 21.12 |
| **AMAZON-670K, $k'=1000$** | | | | | | | | | | | | | | | | |
| Macro-P$_{BCA}$, $\epsilon=10^{-3}$ | 33.08 | 19.36 | *17.46* | 10.48 | 12.11 | **17.75** | 4.00 | 101.39 | 26.64 | 25.31 | *21.25* | 14.16 | 15.85 | 21.82 | 4.00 | 102.97 |
| Macro-P$_{BCA}$, $\epsilon=10^{-5}$ | 33.07 | 19.35 | **17.46** | 10.48 | 12.11 | 17.75 | 5.40 | 117.07 | 26.63 | 25.30 | **21.25** | 14.16 | 15.85 | 21.81 | 5.00 | 111.32 |
| Macro-P$_{BCA}$, $\epsilon=10^{-7}$ | 33.79 | 19.75 | 17.27 | 10.53 | 12.12 | *17.75* | 7.40 | 164.02 | 27.52 | 26.09 | 21.09 | 14.38 | 15.96 | 21.96 | 7.40 | 166.23 |
| Macro-F1$_{BCA}$, $\epsilon=10^{-3}$ | *37.05* | *21.54* | 16.65 | 10.74 | *12.21* | 17.69 | 3.00 | 86.95 | *31.60* | *29.70* | 20.54 | 15.04 | 16.35 | 22.18 | 3.00 | 88.40 |
| Macro-F1$_{BCA}$, $\epsilon=10^{-5}$ | 37.03 | 21.53 | 16.67 | 10.74 | **12.21** | 17.69 | 5.00 | 131.28 | 31.58 | 29.68 | 20.57 | 15.03 | *16.36* | 22.17 | 5.00 | 133.04 |
| Macro-F1$_{BCA}$, $\epsilon=10^{-7}$ | **37.34** | **21.70** | 16.49 | 10.75 | 12.17 | 17.65 | 7.20 | 182.83 | **31.97** | **30.04** | 20.39 | 15.15 | **16.37** | 22.25 | 7.00 | 181.63 |
| Cov$_{BCA}$, $\epsilon=10^{-3}$ | 35.40 | 20.33 | 14.04 | 10.84 | 11.23 | 17.69 | 3.00 | 22.07 | 30.30 | 28.41 | 16.43 | 15.94 | 14.85 | **22.93** | 3.00 | 21.96 |
| Cov$_{BCA}$, $\epsilon=10^{-5}$ | 35.38 | 20.32 | 14.04 | *10.85* | 11.23 | 17.70 | 5.00 | 33.10 | 30.28 | 28.39 | 16.44 | *15.94* | 14.85 | **22.93** | 5.00 | 33.53 |
| Cov$_{BCA}$, $\epsilon=10^{-7}$ | 35.38 | 20.32 | 14.04 | **10.85** | 11.23 | 17.70 | 7.20 | 44.79 | 30.27 | 28.39 | 16.44 | **15.94** | 14.85 | *22.93* | 7.00 | 45.04 |

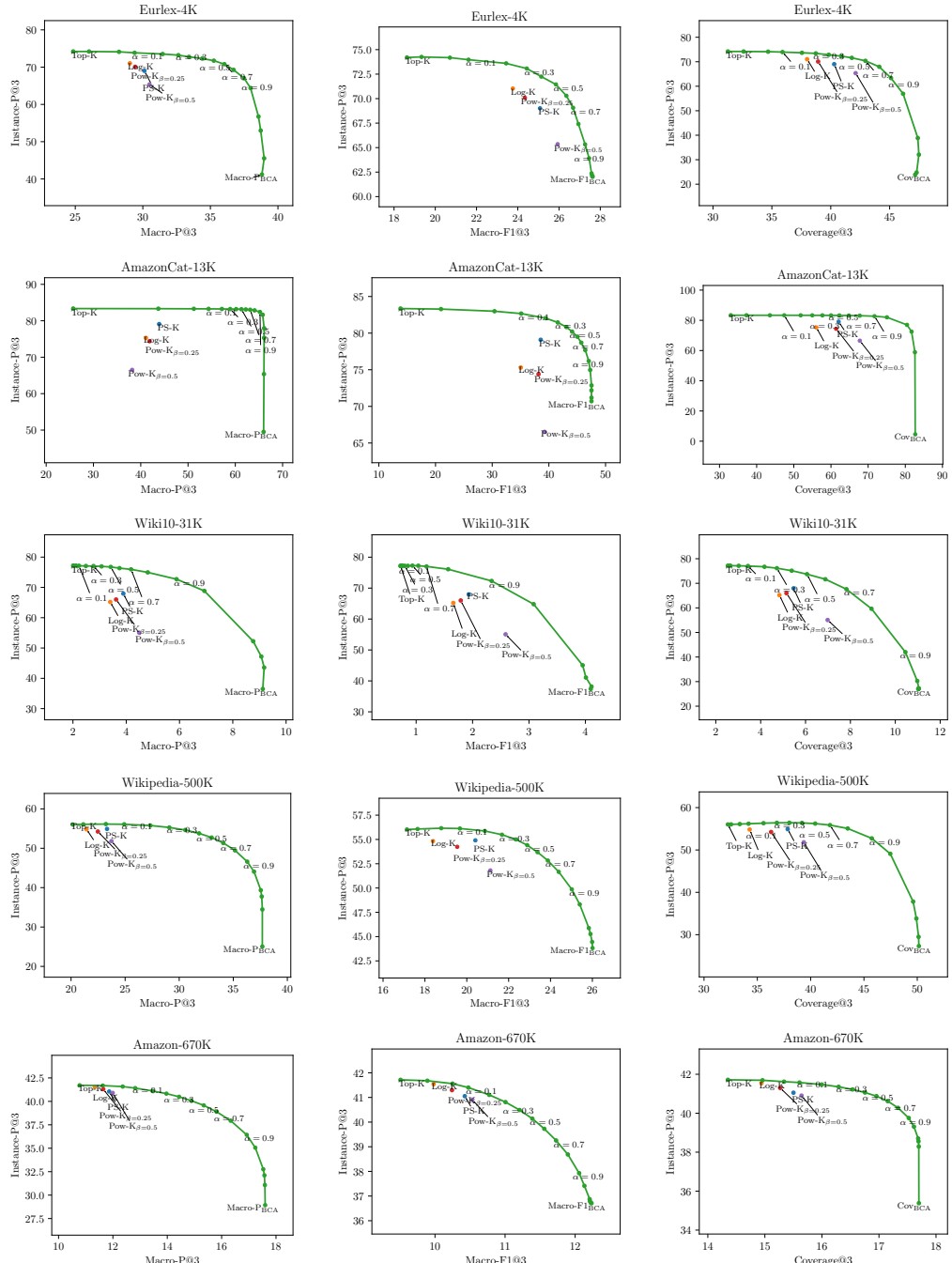

Figure 2: Comparison of the baseline algorithms with the BCA inference with mixed objectives with $k = 3$. The green line shows the results for different interpolations between two measures.

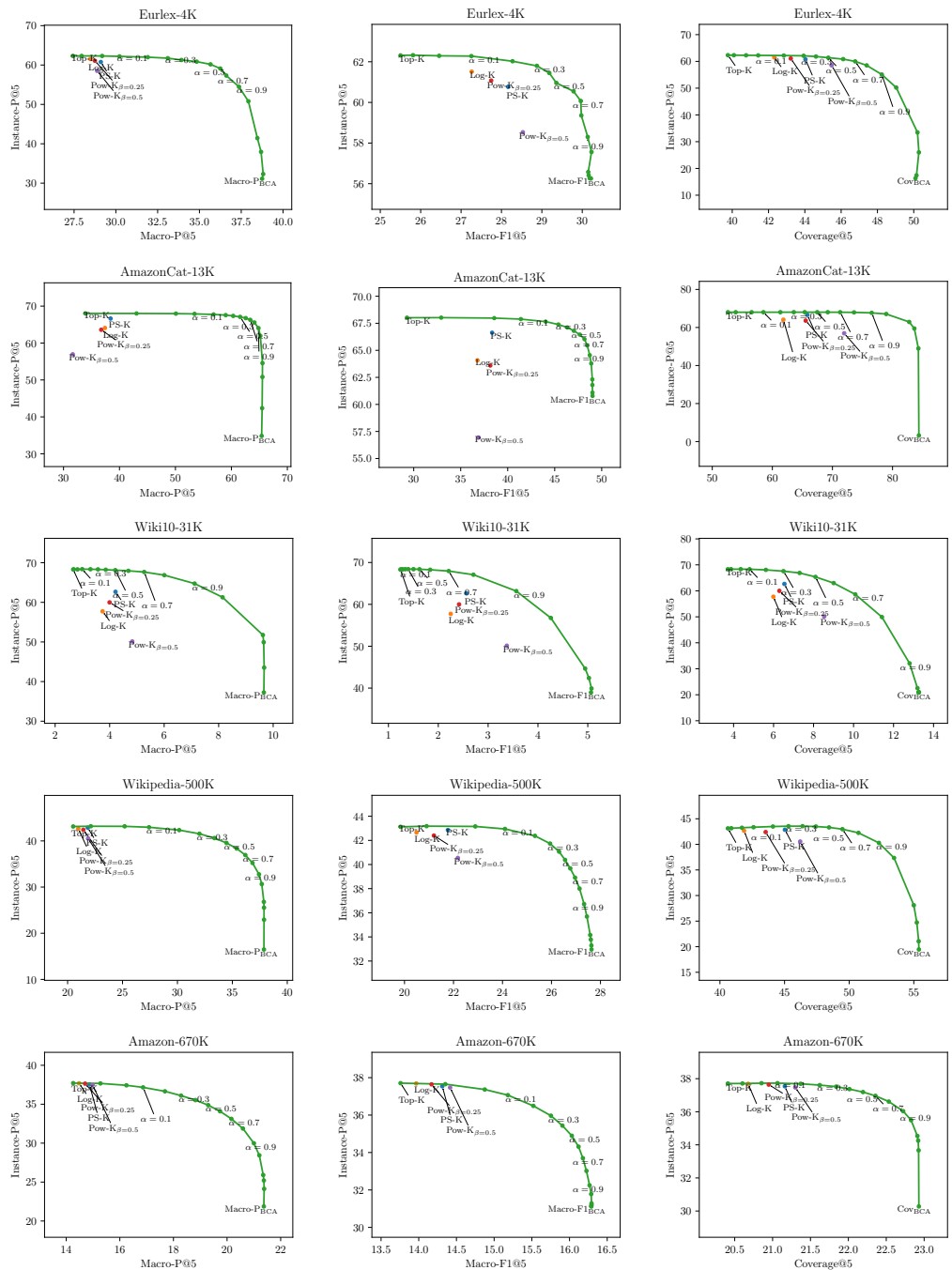

Figure 3: Comparison of the baseline algorithms with the BCA inference with mixed objectives with $k = 5$. The green line shows the results for different interpolations between two measures.

a mixed utility that is a linear combination of instance-wise measures and selected macro-measures. Here, we present the results for three such mixed utilities (combinations of instance-precision with macro-precision, macro-f1-measure, and coverage):

$$
\begin{aligned}
\Psi_1(\boldsymbol{Y}, \hat{\boldsymbol{Y}}) &:= (1-\alpha)\Psi_{\text{Instance-P}}(\boldsymbol{Y}, \hat{\boldsymbol{Y}}) + \alpha\Psi_{\text{Macro-P}}(\boldsymbol{Y}, \hat{\boldsymbol{Y}}) \\
&= \sum_{j=1}^{m}(1-\alpha)\psi_{\text{Instance-P}}(\boldsymbol{y}_{:j}, \hat{\boldsymbol{y}}_{:j}) + \alpha\psi_{\text{Macro-P}}(\boldsymbol{y}_{:j}, \hat{\boldsymbol{y}}_{:j}) \\
&= \sum_{j=1}^{m}(1-\alpha)\frac{t_j}{k} + \alpha\frac{t_j}{mq_j}\,, \\
\Psi_2(\boldsymbol{Y}, \hat{\boldsymbol{Y}}) &:= (1-\alpha)\Psi_{\text{Instance-P}}(\boldsymbol{Y}, \hat{\boldsymbol{Y}}) + \alpha\Psi_{\text{Macro-F1}}(\boldsymbol{Y}, \hat{\boldsymbol{Y}}) \\
&= \sum_{j=1}^{m}(1-\alpha)\psi_{\text{Instance-P}}(\boldsymbol{y}_{:j}, \hat{\boldsymbol{y}}_{:j}) + \alpha\psi_{\text{Macro-F1}}(\boldsymbol{y}_{:j}, \hat{\boldsymbol{y}}_{:j}) \\
&= \sum_{j=1}^{m}(1-\alpha)\frac{t_j}{k} + \alpha\frac{2t_j}{m(q_j + p_j)}\,, \\
\Psi_3(\boldsymbol{Y}, \hat{\boldsymbol{Y}}) &:= (1-\alpha)\Psi_{\text{Instance-P}}(\boldsymbol{Y}, \hat{\boldsymbol{Y}}) + \alpha\Psi_{\text{Cov}}(\boldsymbol{Y}, \hat{\boldsymbol{Y}}) \\
&= \sum_{j=1}^{m}(1-\alpha)\psi_{\text{Instance-P}}(\boldsymbol{y}_{:j}, \hat{\boldsymbol{y}}_{:j}) + \alpha\psi_{\text{Cov}}(\boldsymbol{y}_{:j}, \hat{\boldsymbol{y}}_{:j}) \\
&= \sum_{j=1}^{m}(1-\alpha)\frac{t_j}{k} + \alpha\frac{\mathbb{1}[t_j > 0]}{m}\,,
\end{aligned}
\tag{75}
$$

In Figure 2 and Figure 3, we present the plots with results on two combined measures for different values of $\alpha \in \{0.01, 0.05, 0.1, 0.2, 0.3, 0.4, 0.5, 0.6, 0.7, 0.8, 0.9, 0.95, 0.99, 0.995, 0.999\}$. Once again, the presented results are the mean values over 5 runs with different seeds. The plots show that the instance-vs-macro curve has a nice concave shape that dominates simple baselines. In particular, we can initially improve macro-measures significantly with only a minor drop in instance-measures, and only if we want to optimize even more strongly for macro-measures, we get larger drops in instance-wise measures. A particularly notable feature of the plug-in approach is that the curves in the figure are cheap to produce since there is no requirement for expensive re-training of the entire architecture, so one can easily select an optimal interpolation constant according to some criteria, such as a maximum decrease of instance-wise performance.

### E.6 Hardware

The LIGHTXML model was trained on a workstation with a single Nvidia Tesla V100 GPU with 32 GB of memory and 64 GB of RAM. All the inference strategies were then run on the workstation with 64 GB of RAM. However, to run them, one requires only 16 GB of memory.

# F Efficient inference with Probabilistic Label Trees

In this section, we describe probabilistic labels trees (PLTs) [16], which allow for efficient retrieval of only the top-$k$ labels with the highest conditional probabilities. Then, we introduce a weighted version and show how to modify it to achieve efficient and exact optimal prediction for any $g_j(\eta_j(\boldsymbol{x}))$ that is monotonous in $\eta_j(\boldsymbol{x})$. Thus, if the LPE $\boldsymbol{\eta}$ is modeled by a PLT, this provides an alternative to the sparse inference introduced in Appendix D that also scales to XMLC problems.

## F.1 Probabilistic labels trees

We denote a tree by $\mathcal{T}$, the set of all its nodes by $\mathcal{V}_\mathcal{T}$, the root node by $\mathsf{r}_\mathcal{T}$, and the set of its leaves by $\mathcal{L}_\mathcal{T}$. The leaf $\mathsf{l}_j \in \mathcal{L}_\mathcal{T}$ corresponds to the label $j \in [m]$. The parent node of $\mathsf{v}$ is denoted by $\mathrm{pa}(\mathsf{v})$, and the set of child nodes by $\mathrm{ch}(\mathsf{v})$. The set of leaves of a (sub)tree rooted in node $\mathsf{v}$ is denoted by $\mathcal{L}_\mathsf{v}$, and path from node $\mathsf{v}$ to the root by $\mathrm{path}(\mathsf{v})$.

A PLT uses a tree $\mathcal{T}$ to factorize conditional probabilities of labels, $\eta_j(\boldsymbol{x}) = \mathbb{P}[y_j = 1 \mid \boldsymbol{x}]$, $j \in [m]$, by using the chain rule. Let us define an event that $\mathcal{L}_\mathsf{v}$ contains at least one relevant label in $\boldsymbol{y}$, denoting $z_\mathsf{v} := \mathbb{1}[\exists j : \mathsf{l}_j \in \mathcal{L}_\mathsf{v} \wedge y_j = 1]$. Now for every node $\mathsf{v} \in \mathcal{V}_\mathcal{T}$, the conditional probability of containing at least one relevant label is given by:

$$\eta_\mathsf{v}(\boldsymbol{x}) := \mathbb{P}[z_\mathsf{v} = 1 \mid \boldsymbol{x}] = \prod_{\mathsf{v}' \in \mathrm{path}(\mathsf{v})} \eta(\boldsymbol{x}, \mathsf{v}'), \tag{76}$$

where $\eta(\boldsymbol{x}, \mathsf{v}) := \mathbb{P}[z_\mathsf{v} = 1 \mid z_{\mathrm{pa}(\mathsf{v})} = 1, \boldsymbol{x}]$ for non-root nodes, and $\eta(\boldsymbol{x}, \mathsf{v}) := \mathbb{P}[z_\mathsf{v} = 1 \mid \boldsymbol{x}]$ for the root. Notice that (76) can also be stated recursively as

$$\eta_\mathsf{v}(\boldsymbol{x}) = \eta(\boldsymbol{x}, \mathsf{v}) \cdot \eta_{\mathrm{pa}(\mathsf{v})}(\boldsymbol{x}), \tag{77}$$

and that for leaf nodes we get the conditional probabilities of labels

$$\eta_{\mathsf{l}_j}(\boldsymbol{x}) = \eta_j(\boldsymbol{x}), \quad \text{for } \mathsf{l}_j \in \mathcal{L}_\mathcal{T}. \tag{78}$$

To obtain a PLT, it suffices, for a given $\mathcal{T}$, to train probabilistic classifiers estimating $\eta(\boldsymbol{x}, \mathsf{v})$ for all $\mathsf{v} \in \mathcal{V}_\mathcal{T}$. Analogously to the main paper, we denote estimates of $\eta$ by $\hat{\eta}$.

## F.2 Weighted PLTs

[46] introduced an $A^*$-search based algorithm for efficiently finding the $k$ leaves with the highest gain $g(\mathsf{l}_j, \boldsymbol{x}) = w_j \eta_j(\boldsymbol{x})$, where $w_j \in [0, \infty)$ is the weight given to label $j$. The outline of the search method presented below is an adapted description from [46].

For the gain function $g(\mathsf{l}_j, \boldsymbol{x}) = w_j \hat{\eta}_j(\boldsymbol{x})$, the procedure uses a cost function $c(\mathsf{l}_j, \boldsymbol{x})$ for each path from the root to a leaf. Notice that the following holds:

$$g(\mathsf{l}_j, \boldsymbol{x}) = w_j \hat{\eta}_j(\boldsymbol{x}) = \exp\left(-\left(-\log w_j - \sum_{\mathsf{v} \in \mathrm{path}(\mathsf{l}_j)} \log \hat{\eta}(\boldsymbol{x}, \mathsf{v})\right)\right). \tag{79}$$

Because $\exp$ is monotonous, we can define the following cost function for a selected label $j$, which the algorithm will aim to minimize:

$$c(\mathsf{l}_j, \boldsymbol{x}) := -\log w_j - \sum_{\mathsf{v} \in \mathrm{path}(\mathsf{l}_j)} \log \hat{\eta}(\boldsymbol{x}, \mathsf{v}). \tag{80}$$

We can then guide the $A^*$-search with the function $\bar{c}(\mathsf{v}, \boldsymbol{x}) = p(\mathsf{v}, \boldsymbol{x}) + h(\mathsf{v}, \boldsymbol{x})$, estimating the value of the optimal path in the subtree of node $\mathsf{v}$, where

$$p(\mathsf{v}, \boldsymbol{x}) = -\sum_{\mathsf{v}' \in \mathrm{path}(\mathsf{v})} \log \hat{\eta}(\boldsymbol{x}, \mathsf{v}') \tag{81}$$

is the cost of reaching tree node $\mathsf{v}$ from the root, and

$$h(\mathsf{v}, \boldsymbol{x}) = -\log \max_{j \in \mathcal{L}_\mathsf{v}} w_j \tag{82}$$

is a heuristic function estimating the cost of reaching the best leaf from node $\mathsf{v}$. The $A^*$-search in our procedure evaluates nodes in ascending order of their estimated cost values $\bar{c}(\mathsf{l}_j, \boldsymbol{x})$.

This approach has been proven by [46] to guarantee that $A^*$-search finds the optimal solution—top-$k$ labels with the highest $c(\mathsf{l}_j, \boldsymbol{x})$ and thereby top-$k$ labels with the highest $w_j \eta_j(\boldsymbol{x})$—in the optimally efficient way, i.e., there is no other algorithm used with this heuristic that expands fewer nodes [38].

**Algorithm 9** Select top-$k$ labels with highest gain using PLTs $(\mathcal{T}, \hat{\eta}, \boldsymbol{x}, k, g_j(\cdot))$

1: $\hat{\boldsymbol{y}} \leftarrow \boldsymbol{0}^m$        ▷ Initialize prediction $\hat{\boldsymbol{y}}$ vector to all zeros
2: $\mathcal{Q} \leftarrow \emptyset$        ▷ Initialize priority queue $\mathcal{Q}$, **ordered descending by** $\overline{g}(\mathsf{v}, \boldsymbol{x})$
3: $p(\mathsf{r}_{\mathcal{T}}, \boldsymbol{x}) \leftarrow \hat{\eta}(\boldsymbol{x}, \mathsf{r}_{\mathcal{T}})$        ▷ Calculate estimated conditional probability $p(\mathsf{r}_{\mathcal{T}}, \boldsymbol{x})$ for the tree root
4: $\overline{g}(\mathsf{r}_{\mathcal{T}}, \boldsymbol{x}) \leftarrow \max_{j \in \mathcal{L}_{\mathsf{r}_{\mathcal{T}}}} (g_j(p(\mathsf{r}_{\mathcal{T}}, \boldsymbol{x})))$        ▷ Calculate estimated gain $\overline{g}(\mathsf{r}_{\mathcal{T}}, \boldsymbol{x})$ for the tree root
5: $\mathcal{Q}.\mathrm{add}((\mathsf{r}_{\mathcal{T}}, p(\mathsf{r}_{\mathcal{T}}, \boldsymbol{x}), \overline{g}(\mathsf{r}_{\mathcal{T}}, \boldsymbol{x})))$        ▷ Add the tree root with estimates $p(\mathsf{r}_{\mathcal{T}}, \boldsymbol{x})$ and $\overline{g}(\mathsf{r}_{\mathcal{T}}, \boldsymbol{x})$ to the queue
6: **while** $\|\hat{\boldsymbol{y}}\|_1 < k$ **do**        ▷ While the number of predicted labels is less than $k$
7:     $(\mathsf{v}, p(\mathsf{v}, \boldsymbol{x}), \_) \leftarrow \mathcal{Q}.\mathrm{pop}()$        ▷ Pop the element with the lowest cost from the queue
8:     **if** $\mathsf{v}$ is a leaf **then** $\hat{y}_{\mathsf{v}} \leftarrow 1$        ▷ If the node is a leaf, set the corresponding label in the prediction vector
9:     **else for** $\mathsf{v}' \in \mathrm{ch}(\mathsf{v})$ **do**        ▷ If the node is an internal node, for all child nodes
10:        $p(\mathsf{v}', \boldsymbol{x}) \leftarrow p(\mathsf{v}, \boldsymbol{x}) \hat{\eta}(\boldsymbol{x}, \mathsf{v}')$        ▷ Calculate $p(\mathsf{v}', \boldsymbol{x})$
11:        $\overline{g}(\mathsf{v}', \boldsymbol{x}) \leftarrow \max_{j \in \mathcal{L}_{\mathsf{v}}} (g_j(p(\mathsf{v}', \boldsymbol{x})))$        ▷ Calculate estimate $\overline{g}(\mathsf{v}', \boldsymbol{x})$
12:        $\mathcal{Q}.\mathrm{add}((\mathsf{v}', p(\mathsf{v}', \boldsymbol{x}), \overline{g}(\mathsf{v}', \boldsymbol{x})))$        ▷ Add $\mathsf{v}'$, and its estimates $p(\mathsf{r}_{\mathcal{T}}, \boldsymbol{x})$ and $\overline{g}(\mathsf{r}_{\mathcal{T}}, \boldsymbol{x})$ to the queue
13: **return** $\hat{\boldsymbol{y}}$        ▷ Return the prediction vector

### F.3   PLTs for monotonous gain functions

Notice that in Weighted PLTs instead of minimizing cost, one can directly optimize gain by using a tree-search procedure that evaluates nodes in descending order of their estimated gain $\overline{g}(\mathsf{v}, \boldsymbol{x}) = p(\mathsf{v}, \boldsymbol{x}) \cdot h(\mathsf{v}, \boldsymbol{x})$, where $p(\mathsf{v}, \boldsymbol{x}) = \sum_{\mathsf{v}' \in \mathrm{path}(\mathsf{v})} \hat{\eta}(\boldsymbol{x}, \mathsf{v}')$ and $h(\mathsf{v}, \boldsymbol{x}) = \max_{j \in \mathcal{L}_{\mathsf{v}}} w_j$. To generalize this formulation and apply it to the proposed algorithms, we need to be able to find the top-$k$ with any gain function $g_j(\eta_j(\boldsymbol{x}))$ that is monotonous with $\eta_j(\boldsymbol{x})$.

We can guide the tree search using the estimated gain function $\overline{g}(\mathsf{v}, \boldsymbol{x}) = h(\mathsf{v}, p(\mathsf{v}, \boldsymbol{x}))$, where

$$p(\mathsf{v}, \boldsymbol{x}) = \prod_{\mathsf{v}' \in \mathrm{path}(\mathsf{v})} \hat{\eta}(\boldsymbol{x}, \mathsf{v}') \,, \tag{83}$$

the same as in the case of Weighted PLT, it simply corresponds to the conditional probability of finding at least one positive label in the subtree of node $\mathsf{v}$, and the heuristic part is:

$$h(\mathsf{v}, \boldsymbol{x}) = \max_{j \in \mathcal{L}_{\mathsf{v}}} (g_j(p(\mathsf{v}, \boldsymbol{x}))) \,. \tag{84}$$

As in the case of Weighted PLT, we would like to guarantee that this search procedure finds the optimal solution—the top-$k$ labels with the highest $g(\mathsf{l}_j, \boldsymbol{x})$. To do that, we need to ensure that $\overline{g}(\mathsf{v}, \boldsymbol{x})$ is admissible, i.e., in case of maximization, it never underestimates the gain from reaching a leaf node [38]. We also would like $\overline{g}(\mathsf{v}, \boldsymbol{x})$ to be consistent, making the proposed tree search procedure optimally efficient, i.e., there is no other algorithm used with the same $\overline{g}(\mathsf{v}, \boldsymbol{x})$ that expands fewer nodes [38]. Algorithm 9 outlines this procedure for finding the top-$k$ labels with highest values of $g_j(\eta_j(\boldsymbol{x}))$ that is monotonous with $\eta_j(\boldsymbol{x})$, and that can replace the select top-k$(\boldsymbol{g})$ operation in the proposed BCA and Greedy algorithms. Unfortunately, not all gain functions $g_j(\eta_j(\boldsymbol{x}))$ can be calculated efficiently. To calculate the semi-empirical quantity $\tilde{p}$ exactly, one needs to know *all* the $\eta_j(\boldsymbol{x})$. To obtain them, the whole tree need to be evaluated, which prevents this method from being efficient. Because of that, we use this technique to get exact predictions for the measures that are defined solely on $\tilde{t}$ and $\tilde{q}$, like macro-precision and coverage.

### F.4   Experimental comparison of inference with PLTs

In this experiment, we evaluate the inference using probabilistic label trees that can efficiently perform the exact select top-k$(\boldsymbol{g})$ operation. We compare inference times of these variants of the methods that require only one pass over the dataset: TOP-K, PS-K, POW-K, LOG-K, MACRO-P$_{\mathrm{GREED}}$, MACRO-R$_{\mathrm{PRIOR}}$, and COV$_{\mathrm{GREED}}$, for $k = \{3, 5\}$. As a baseline, we use the same PLT to obtain the $k' = 100$ and $k' = 1000$ highest $\boldsymbol{\eta}(\boldsymbol{x}_i)$, and then run the sparse algorithms on top of these predictions. We use a modified implementation of PS-PLT [46, 17], trained with default settings.

We present the results in Table 10. As in the case of the main experiment, we again observe that the specialized inference strategies are indeed the best on the measure they aim to optimize. Compared to the cost of obtaining marginals for $k' = 100$ and $k' = 1000$ labels, which can be later used with one of the proposed inference algorithms (the cost of running this step is not included in the table), the greedy methods combined with PLT-based search consistently achieve speed-up when compared to

Table 10: Results of different inference strategies on @$k$ measures with $k = \{3, 5\}$ using PLT model. Notation: P—precision, R—recall, F1—F1-measure, Cov—Coverage, T—time in seconds, $k'$=100/1000—speed-up relative to the time required to obtain top-100/1000 $\boldsymbol{\eta}(\boldsymbol{x})$ for all the instances in $\boldsymbol{X}$. The green color indicates cells in which the strategy matches the metric. The best results are in **bold** and the second best are in *italic*.

| Inference strategy | Instance @3 P | Instance @3 R | Macro @3 P | Macro @3 R | Macro @3 F1 | Macro @3 Cov | Time/Speed-up @3 T | $k'$=100 | $k'$=1000 | Instance @5 P | Instance @5 R | Macro @5 P | Macro @5 R | Macro @5 F1 | Macro @5 Cov | Time/Speed-up @5 T | $k'$=100 | $k'$=1000 |
|---|---|---|---|---|---|---|---|---|---|---|---|---|---|---|---|---|---|---|
| | | | | | | | EURLEX-4K | | | | | | | | | | | |
| PLT-TOP-1000 | - | - | - | - | - | - | 27.80 | 0.58 | 1.00 | - | - | - | - | - | - | 27.80 | 0.58 | 1.00 |
| PLT-TOP-100 | - | - | - | - | - | - | 16.19 | 1.00 | 1.72 | - | - | - | - | - | - | 16.19 | 1.00 | 1.72 |
| PLT-TOP-K | *67.28* | *39.78* | 20.65 | 12.59 | 14.71 | 39.66 | 2.12 | 7.62 | 13.09 | *56.28* | *54.53* | 23.51 | 20.02 | 20.41 | 50.73 | 2.95 | 5.50 | 9.44 |
| PLT-PS-K | **67.45** | **39.90** | 21.27 | 13.36 | 15.43 | 41.19 | 5.82 | 2.78 | 4.77 | **56.53** | **54.85** | 25.77 | 25.16 | *24.09* | 59.71 | 5.82 | 2.78 | 4.77 |
| PLT-POW-K $_{\beta=0.5}$ | 67.03 | 39.61 | 21.79 | 14.57 | 16.45 | 43.43 | 8.00 | 2.03 | 3.48 | 54.63 | 53.03 | 25.62 | *27.39* | **24.93** | 63.19 | 8.00 | 2.03 | 3.48 |
| PLT-LOG-K | 67.42 | 39.86 | 20.87 | 12.87 | 14.98 | 40.18 | 4.15 | 3.90 | 6.70 | 56.70 | 54.92 | 24.53 | 23.01 | 22.50 | 55.48 | 4.15 | 3.90 | 6.70 |
| PLT-MACRO-P$_{\text{GREED}}$ | 19.68 | 11.39 | **25.49** | 16.05 | 14.94 | *57.95* | 14.02 | 1.16 | 1.98 | 13.86 | 13.33 | **28.92** | 18.09 | 18.15 | 61.50 | 16.74 | 0.97 | 1.66 |
| PLT-MACRO-R$_{\text{PRIOR}}$ | 54.53 | 32.09 | *24.61* | **21.72** | **20.96** | 56.38 | 14.21 | 1.14 | 1.96 | 38.04 | 36.97 | 24.76 | **28.69** | 23.40 | *67.90* | 14.21 | 1.14 | 1.96 |
| PLT-COV$_{\text{GREED}}$ | 34.19 | 19.84 | 23.80 | *20.84* | *18.65* | **63.11** | 6.58 | 2.46 | 4.22 | 24.79 | 23.88 | 21.10 | 25.91 | 18.76 | **68.84** | 8.79 | 1.84 | 3.16 |
| | | | | | | | AMAZONCAT-13K | | | | | | | | | | | |
| PLT-TOP-1000 | - | - | - | - | - | - | 2952.01 | 0.26 | 1.00 | - | - | - | - | - | - | 2952.01 | 0.26 | 1.00 |
| PLT-TOP-100 | - | - | - | - | - | - | 768.51 | 1.00 | 3.84 | - | - | - | - | - | - | 768.51 | 1.00 | 3.84 |
| PLT-TOP-K | **78.44** | **59.42** | 33.16 | 11.09 | 14.79 | 40.97 | 66.31 | 11.59 | 44.52 | **63.71** | **74.66** | 44.40 | 30.69 | 33.00 | 61.09 | 87.49 | 8.78 | 33.74 |
| PLT-PS-K | *78.27* | *59.32* | 37.09 | 13.08 | 17.39 | 46.23 | 229.04 | 3.36 | 12.89 | *63.56* | *74.55* | *50.10* | 48.12 | **45.94** | 78.46 | 229.04 | 3.36 | 12.89 |
| PLT-POW-K $_{\beta=0.5}$ | 72.20 | 54.56 | *41.86* | 22.84 | *27.46* | 60.19 | 481.53 | 1.60 | 6.13 | 56.06 | 66.33 | 40.20 | *60.10* | *45.28* | 84.52 | 481.53 | 1.60 | 6.13 |
| PLT-LOG-K | 76.15 | 57.63 | 37.90 | 14.01 | 18.59 | 47.89 | 142.18 | 5.41 | 20.76 | 61.44 | 72.13 | 48.26 | 42.91 | 42.68 | 72.71 | 142.18 | 5.41 | 20.76 |
| PLT-MACRO-P$_{\text{GREED}}$ | 3.98 | 2.07 | **42.12** | 32.69 | 20.55 | *85.07* | 1135.02 | 0.68 | 2.60 | 2.67 | 2.30 | **54.90** | 34.99 | 32.58 | 86.40 | 1127.85 | 0.68 | 2.62 |
| PLT-MACRO-R$_{\text{PRIOR}}$ | 46.83 | 31.15 | 34.79 | **49.92** | **37.54** | 79.72 | 1248.41 | 0.62 | 2.36 | 31.02 | 33.37 | 27.86 | **68.51** | 35.42 | *90.11* | 1248.41 | 0.62 | 2.36 |
| PLT-COV$_{\text{GREED}}$ | 6.20 | 3.26 | 29.86 | *45.04* | 20.32 | **89.03** | 965.45 | 0.80 | 3.06 | 3.96 | 3.42 | 24.46 | 48.22 | 17.08 | **91.33** | 1082.03 | 0.71 | 2.73 |
| | | | | | | | WIKI-31K | | | | | | | | | | | |
| PLT-TOP-1000 | - | - | - | - | - | - | 638.46 | 0.38 | 1.00 | - | - | - | - | - | - | 638.46 | 0.38 | 1.00 |
| PLT-TOP-100 | - | - | - | - | - | - | 239.52 | 1.00 | 2.67 | - | - | - | - | - | - | 239.52 | 1.00 | 2.67 |
| PLT-TOP-K | **72.30** | **12.55** | 1.97 | 0.31 | 0.48 | 3.06 | 16.45 | 14.56 | 38.82 | *63.30* | *18.05* | 2.98 | 0.69 | 1.01 | 4.91 | 25.57 | 9.37 | 24.97 |
| PLT-PS-K | *72.00* | *12.53* | 3.47 | 0.88 | 1.24 | 5.47 | 100.75 | 2.38 | 6.34 | **63.83** | **18.36** | 6.79 | 2.97 | 3.70 | 11.44 | 100.75 | 2.38 | 6.34 |
| PLT-POW-K $_{\beta=0.5}$ | 64.46 | 11.25 | 5.94 | 2.35 | 3.00 | 10.17 | 227.97 | 1.05 | 2.80 | 53.91 | 15.51 | 9.58 | 5.78 | 6.52 | 17.42 | 227.97 | 1.05 | 2.80 |
| PLT-LOG-K | 65.28 | 11.26 | 3.22 | 0.82 | 1.17 | 5.39 | 73.84 | 3.24 | 8.65 | 57.32 | 16.28 | 5.68 | 2.33 | 2.99 | 10.17 | 73.84 | 3.24 | 8.65 |
| PLT-MACRO-P$_{\text{GREED}}$ | 24.66 | 4.15 | **10.19** | *4.81* | *5.71* | 17.64 | 268.05 | 0.89 | 2.38 | 18.83 | 5.24 | *11.63* | 6.86 | 7.57 | 21.83 | 337.87 | 0.71 | 1.89 |
| PLT-MACRO-R$_{\text{PRIOR}}$ | 30.96 | 5.33 | *9.79* | **6.01** | **6.38** | 17.37 | 422.81 | 0.57 | 1.51 | 22.97 | 6.52 | **11.88** | **9.57** | 8.77 | *23.24* | 422.81 | 0.57 | 1.51 |
| PLT-COV$_{\text{GREED}}$ | 35.13 | 5.99 | 7.79 | 3.76 | 4.52 | **18.01** | 152.66 | 1.57 | 4.18 | 27.92 | 7.90 | 8.98 | 5.82 | 6.30 | **24.38** | 193.64 | 1.24 | 3.30 |

top $k' = 1000$ inference. Unfortunately, they are also often slower than $k' = 100$ inference. As we showed in the previous section, $k' = 100$ is usually enough to get good predictive performance on macro measures using BCA and Greedy algorithms. However, here, the PLT-based methods obtain the exact solution for finding the prediction with the highest expected gain. While it is worth noting that tree structure and the type of node estimator have an impact on the performance of PLT-based algorithms and we conducted this experiment with just a single setup, we believe that this experiment shows that PLT-based inference, in some cases, can be an alternative to inference with sparse marginal estimates.

