# OpenReview forum: "Generalized test utilities for long-tail performance in extreme multi-label classification"
_NeurIPS.cc/2023/Conference — NeurIPS 2023 poster_

### Official Review · Reviewer_m6Nv · 2023-07-04

**Soundness:** 3 good
**Presentation:** 2 fair
**Contribution:** 3 good
**Rating:** 5
**Confidence:** 5

**Summary:**

This paper focuses on the long-tail problem in extreme multi-label classification. To address this problem,  the authors propose a new method to optimize performance metrics for extreme multi-label tasks via the expected test utility (ETU) framework. Experimental and theoretical results are also provided.

**Strengths:**

1. The long-tail problem in the extreme multi-label classification task is important. It is good to explore how to fairly evaluate the performance of multi-label learning algorithms on tail labels.
2. The proposal is technically sound. Directly optimizing performance measures for tail labels is make sense.

**Weaknesses:**

The main concern about the paper is the experiments. There are also some other novel algorithms that are proposed for the long-tail problem in multi-label classification tasks and can achieve good performance, but the authors did not compare with these studies in the experiments.

Moreover, there are some minor issues, such as the related works should be discussed.

**Questions:**

1. There are also some other works that focus on the long-tailed multi-label classification tasks, instead of directly optimizing the performance metrics, how about the performance of these works?

---

> ### Author Rebuttal · Authors · 2023-08-09
>
> We sincerely thank the reviewer for the review and questions. We appreciate the hard work. Below we address the main question.
>
> In contrast to many works in extreme classification, which are mainly algorithmic and aimed to obtain "better" performance on tail-labels with existing metrics, our work mainly focuses on the performance metric itself. Through Table 1 of the paper, we argue that the prevalent instance-based measures, such as P@k and PSP@k, used by the community, are largely insensitive to the removal of a large fraction of tail-labels. We speculate that the novel algorithms mentioned by the reviewer would not help much in this regard, as the existing metrics are mainly focusing on head labels.
>
> Furthermore, we would like to highlight the comments by reviewer 4FGQ : "It is well known that the existing metrics for XMC problems cannot measure the performance on tail labels because of the data distribution. I agree with the authors that the widely used propensity scores are heuristics that are not to be used as metrics, especially it requires hyper-parameters which would introduce much consistency issues when used for comparison. The authors express the metrics in the ETU framework, thus unifying the existing and potential metrics." In other words, a direct algorithm-to-algorithm comparison on existing metrics would be misleading. The comparison on the budgeted@k macro-measure could also be challenging as the mentioned algorithms would need to be tailored for predicting top-$k$ labels.
>
> In our work we are (i) investigating some shortcomings of the existing metrics, (ii) proposing alternatives based on macro-measures for budgeted@k predictions, and (iii) introducing algorithms to optimize them based on a plug-in approach assuming access to an estimator for conditional label probabilities $\eta$. The algorithms mentioned by the reviewer might be useful in improving the estimates of $\eta$s. This is an interesting question, but beyond the scope of this paper. Nevertheless, we would appreciate if the reviewer would explicitly say which algorithms she/he has in mind.

---

### Official Review · Reviewer_UXPM · 2023-07-05

**Soundness:** 3 good
**Presentation:** 3 good
**Contribution:** 2 fair
**Rating:** 5
**Confidence:** 2

**Summary:**

This paper analyzes generalized metrics budgeted “at k” by formulating it in the expected test utility (ETU) framework. They derive optimal prediction rules and construct their computationally efficient approximations with provable regret guarantees and being robust against model misspecification.

**Strengths:**

This paper is sound and clearly written.

**Weaknesses:**

Limited contribution:

The paper investigates optimal solutions for the class of utility functions that can be linearly decomposed over labels into binary utilities. Is it possible to extend this method to a more broad class of utility functions.

The algorithm block coordinate ascent looks not new to me.




**Questions:**

Table 1 shows that the macro measures are suitable for long tails experimentally. Would you please explain the reason for it in theory?

**Limitations:**

Yes.

---

> ### Author Rebuttal · Authors · 2023-08-09
>
> We sincerely thank the reviewer for the review and questions. We appreciate the hard work. Below we address the main concerns.
>
> **Regarding the extensibility of the proposed approach**
>
> We first want to stress that the class of linearly decomposable into labels functions studied in this paper is already quite general, including standard instance-wise measures (at k) like precision, Hamming loss, weighted instance-wise losses such as PSP, as well as macro measures with arbitrary binary loss function as basis (ref. Table). Further, linear combinations of these utilities are permitted (see our answer to Reviewer ELgR).
>
> The first challenge in extensibility is that a crucial feature of the investigated loss functions is that they can be optimized using only marginal label probabilities $\eta_i(x) = P[Y_i=1 | X]$. This is important for the plug-in approach, as estimating the full joint distribution of labels is intractable. This still leaves some design space for generalizations, foremost probably replacing the arithmetic mean in the macro-average with a more general aggregation function. We haven't investigated these options, but if the reviewer has a concrete example of a loss function in mind, we would be happy to discuss it.
>
> ---
>
> > *Table 1 shows that the macro measures are suitable for long tails experimentally. Would you please explain the reason for it in theory?*
>
> Please note that we do not claim that *all* macro-averages are long-tail metrics, e.g., macro-average Hamming loss is equivalent to instance-wise Hamming loss, and thus not more tail-adapted. The main claim of Table 1 is that typical instance-wise loss functions are almost invariant under the removal of a large portion of tail labels, whereas there exist macro metrics that are sensitive to this operation. There is no "first principles" theoretical explanation for why macro-averages are good for measuring tail-performance. We can give a partial explanation as follows:
>
> Assume that the binary utility over which we take the macro-average has range [0, 1], and it is zero if there is no true positive in the prediction. If we have m labels in total, of which m' are tail labels, then removing the entire tail means that the binary utility on all these m' labels is zero, and the overall macro-average is upper-bounded by 1 - m'/m. This is not true for popular instance-wise measures such as Precision@k.

---

> > ### Comment · Reviewer_UXPM · 2023-08-18
> >
> > Thanks for your clarifications. I keep my score unchanged

---

### Official Review · Reviewer_4FGQ · 2023-07-06

**Soundness:** 3 good
**Presentation:** 3 good
**Contribution:** 3 good
**Rating:** 7
**Confidence:** 3

**Summary:**

In this paper, the authors studies the evaluation metrics for the long tailed extreme multilabel classification problems. Compared with existing heuristics, such as PSP, they formulate the metrics in the expected test utility framework. Inference rules are derived to obtain optimal metrics. Approximations are construct for the metrics that are difficult to compute.

**Strengths:**

This paper picks an interesting but not well studied direction and gives thorough study with reliable derivations and proofs. It is well known that the existing metrics for XMC problems cannot measure the performance on tail labels because of the data distribution. I agree with the authors that the widely used proensity scores are heuristics that are not to be used as metrics, especially it requires hyper-parameters which would introduce much consistency issues when used for comparison. The authors express the metrics in the ETU framework thus unifying the existing and potential metrics.

The paper gives detailed derivations for the optimal prediction rules for each of the derived metrics and provides cheap and easily applicable approximations for metrics that are hard to compute. Empirical results on 4 public XMC benchmark datasets indicates that the optimal rules can actually obtain the best score for the corresponding metrics.

**Weaknesses:**

The paper is overall well presented but some of the terms are used without introducing/explaining, such as "macro-average".

Minor:
* Line 82: $C_{11}$ to true positives

**Questions:**

Inference with a global budget k is easy to implement but not always optimal in terms of utility-budget trade-off. If there's no top-k constraint, how can we design the metrics/inference rules to determine the number of predictions for each instance?

**Limitations:**

It would be great to see the complexity analysis of the optimal prediction rules to understand the inference overhead.

---

> ### Author Rebuttal · Authors · 2023-08-09
>
> We sincerely thank the reviewer for the review and questions. We appreciate the hard work. Below we address the main questions.
>
> > *Inference with a global budget k is easy to implement but not always optimal in terms of utility-budget trade-off. If there's no top-k constraint, how can we design the metrics/inference rules to determine the number of predictions for each instance?*
>
> We are not entirely sure what the reviewer is asking for. We would appreciate additional clarification if possible.
>
> We agree that the global budget of $k \times n$ labels, with $n$ being the number of test examples, could be an alternative definition of the problem. However, it would be suited to different applications than our formulation. In such a case, the optimization procedure would determine the number of predicted labels per instance.  Let us also notice that by using unbudgeted metrics like Hamming loss or the standard macro-averaged F-measure one naturally obtains a varying number of predictions per instance.
>
> ---
>
> > *It would be great to see the complexity analysis of the optimal prediction rules to understand the inference overhead.*
>
> We discuss the complexity in Section 6 of our paper. One iteration of the proposed block coordinate algorithm, assuming we have a non-sparse matrix of estimates of $\eta$, has time complexity of $O(n(m \log k))$ and space complexity of $O(nm).
>
> This can be improved by short-listing using a sparse matrix of $\eta$s with only $k’$ non-zero elements per row, resulting in $O(n(k ′ \log k))$ time and $O(n(k ′ + k))$ space complexity. The important question is then how many iterations the algorithm runs before terminating. We also answer this question in response to Reviewer 4ZEa. For metrics taking values in a bounded range and $\epsilon > 0$, the algorithm can only make up to $O(1/\epsilon)$ steps in the `while’ loop, as each step must decrease the objective by at least $\epsilon$. However, even when $\epsilon=0$, the algorithm terminates after a finite number of steps.
>
> In practice, despite small $\epsilon$, the algorithm usually terminates after a few iterations (in the worst observed cases, it was a few tens of iterations). In Table 1 in the PDF attached to the global response, we also present the number of iterations and CPU time that was used for our inference procedures.

---

> > ### Comment · Reviewer_4FGQ · 2023-08-21
> >
> > Thanks for the author's response. I will keep my score unchanged.

---

### Official Review · Reviewer_ELgR · 2023-07-07

**Soundness:** 3 good
**Presentation:** 2 fair
**Contribution:** 2 fair
**Rating:** 5
**Confidence:** 3

**Summary:**

This paper critiques that the existing extreme classification evaluation metrics don't give the complete picture with respect to all labels (more specifically performance on head labels overpowers performance on tail labels), hence it recommends using macro-averaged metrics which are more favorable to tail labels. The authors use expected test utility framework to come up with BCA based methods to get modified prediction rules given a prediction matrix from a trained XMC model. Experiments on benchmark datasets indicate that just taking the top-k predictions is not ideal for marco averaged metrics and the proposed approach performs significantly better.

**Strengths:**

- The proposed approach works as a plugin method that can be used with any existing XMC method without any need for retraining
- Proposed approach is well reasoned
- Approximate versions scalable to large datasets without the need for specialized hardware

**Weaknesses:**

- Not sure how important are macro measures for evaluating performance on XMC datasets as the data imbalance is inherent to XMC applications and in most scenarios we do want to mimic this imbalance in model predictions as well (for e.g. in ad recommendation problem with a billion ads in corpus, certain products are popular, so giving equal weightage to each product during evaluation might not be very useful). I agree that we do want to improve quality of predictions on tail labels as well but not at the cost of making worse overall predictions. Results indicate that trying to do well on macro-averaged metrics heavily impacts the performance on standard metrics which is not desirable.
- Section 4 is a bit hard to follow

Minor typo:
line 82, "c11 to true negatives." -> "c11 to true positives."

**Questions:**

is it possible to optimize for a weighted combination of standard precision and macro precision, so that we can smoothly interpolate between these two metrics?

**Limitations:**

Yes

---

> ### Author Rebuttal · Authors · 2023-08-09
>
> We sincerely thank the reviewer for the review and questions. We appreciate the hard work. Below we address the main questions:
>
> **Regarding the importance of macro measures for evaluating performance on XMC datasets as the data imbalance and a trade-off between performance on standard metrics and macro-averaged metrics.**
>
> We think it is a serious problem if a model only manages to get *at least one* true positive on about 25% of all labels (as suggested by the coverage results in Table 2). Ideally, we would like to improve the performance on tail labels without sacrificing too much performance on head labels. This naturally leads to the question the reviewer has posed whether it is possible to interpolate between the instance and macro metrics to find a better trade-off. The answer is yes, and there are actually two ways this can be achieved.
>
> The first is a straight-forward interpolation, as instance-wise precision-at-k is covered by our framework, and our class of utility functions is closed under linear combinations. Such an objective can be optimized by the proposed block-coordinate algorithm without any modification. In the PDF attached to the general response, we present plots with a result of the optimization of two following objectives with different values of $\alpha$:
>
> $\Psi(\mathbf{Y}, \mathbf{\hat Y}) = (1 - \alpha) \times \text{Instance-P}@k(\mathbf{Y}, \mathbf{\hat Y}) + \alpha \times \text{Macro-P}@k(\mathbf{Y}, \mathbf{\hat Y})$
>
> and
>
> $\Psi(\mathbf{Y}, \mathbf{\hat Y}) = (1 - \alpha) \times \text{Instance-P}@k(\mathbf{Y}, \mathbf{\hat Y}) + \alpha \times \text{Macro-F1}@k(\mathbf{Y}, \mathbf{\hat Y})$.
>
> The plots show that the instance-vs-macro curve has a nice concave shape that dominates simple baselines. In particular, we can initially improve macro-measures significantly with only a minor drop in instance-measures, and only if we want to optimize even more strongly for macro-measures we do get larger drops in instance-wise measures. A particularly notable feature of the plug-in approach is that the curves in the figure are cheap to produce since there is no requirement for expensive re-training of the entire architecture, so one can easily select an optimal interpolation constant according to some criteria, such as a maximum decrease of instance-wise performance.
>
> Second, we note that our framework admits using different binary measures for each label. The most simple way to exploit this is to use a weighted macro-average, giving more weight to head labels. We have actually employed this in the paper, in the sense that instance-wise precision-at-k is just macro-precision-at-k with a 1/prior weighting for the different labels. Parametrizing the exponent $(1/\text{prior}^\beta)$ allows for a different way of interpolation to standard macro-averages, which are realized by beta=0. The results for beta=1/2 are presented in Table 2.
>
> ---
>
> > *Section 4 is a bit hard to follow*
>
> We thank for the suggestion. We will definitely try to improve the clarity of this section in the next version using additional space (if the reviewer has some more specific suggestions, we will be glad to hear them).

---

> > ### Comment · Reviewer_ELgR · 2023-08-21
> >
> > I thank the authors for their response. Looking at Figure 1 in the additional pdf, it seems that improvements over baselines offered on the bigger Amazon-670K dataset are minor compared to the smaller ones. I am still not entirely convinced about the utility of macro measures in isolation for XMC scenarios but I do think it can have applications in some specific cases and the proposed approach and its modifications (which interpolate between macro and instance-wise precision) is a reasonable way of solving the problem.

---

> > > ### Author Response · Authors · 2023-08-21
> > > **Re: Official Comment by Reviewer ELgR**
> > >
> > > We thank the reviewer for reading our responses and for the comment.
> > >
> > > The improvements on Amazon-670K are indeed generally small (in most cases, both on the plots as well as in the tables presented in the main paper). Nevertheless, we would like to remark that the results for the Wikipedia-500K dataset, given in the Appendix, indicate more substantial improvements.
> > >
> > > We appreciate that the reviewer admits that our approach can be an interesting option for some applications.

---

### Official Review · Reviewer_4ZEa · 2023-07-23

**Soundness:** 3 good
**Presentation:** 3 good
**Contribution:** 3 good
**Rating:** 6
**Confidence:** 3

**Summary:**

This paper discusses evaluation metrics to measure the long-tail labels' performance of the extreme multi-label classification (XMLC) problems. The author proposed that macro-average based metrics (e.g., macro-avg Precision/Recall/F1 at k) are more suitable to measure the performance of long-tail labels compared to conventional instance-wised precision/recall at k. The author further discuss how to optimize these metrics on a given test set, with theoretical justification on the objective functions and deviation bounds. The empirical results verified the effectiveness of their proposed optimization methods for the given target metric.

**Strengths:**

- Leverage the ETU framework to approximately optimize Macro-average metrics seems novel
- Theoretical justifications sounds reasonable and technical derivations are solid

**Weaknesses:**

- Selecting thresholds to optimize Marco-average metrics are not new in multi-label community. The author seems not aware of some  classical/heuristic approaches. For example, the sCut and SCutFBR methods studied in [1,2,3]. The author should also compare these methods in the experiment sections.

**Reference**
- [1] Y. Yang. A Study on Thresholding Strategies for Text Categorization. SIGIR 2001
- [2] Lewis et al. RCV1: A New Benchmark Collection for Text Categorization Research. JMLR 2004
- [3] Fan et al. A Study on Threshold Selection for Multi-label Classification. Technical Report 2007

**Questions:**

**Major Questions**
- Q1: How does the proposed method relate and connect to other conventional threshold tuning approaches (e.g., [1,2,3])?
- Q2: How does the proposed method compare empirically to other conventional threshold tuning approaches (e.g., [1,2,3])?
- Q3: For the block coordinate ascent algorithm, how sensitive is it to the initial point? If it is a non convex function, I suppose the initialization problem matters? Any better heuristic than "predicting k random labels" that described at line 185?
- Q4: For Line 195-198, monotonicity alone is not enough to ensure convergence [4]. Any more thorough/technical statement for the convergence analysis of the proposed block coordinate ascent algorithm?

**Minor issues**
- In Eq(5), the first summation has index $i$ in the subscript, but not index $i$ seems not being used in the function of $u_w()$.
- At Line 154: "the proof is given in Appendix F". However, it seems to me that the proof appeared in Appendix G (Theorem G.4)?

**Reference**
- [1] Y. Yang. A Study on Thresholding Strategies for Text Categorization. SIGIR 2001
- [2] Lewis et al. RCV1: A New Benchmark Collection for Text Categorization Research. JMLR 2004
- [3] Fan et al. A Study on Threshold Selection for Multi-label Classification. Technical Report 2007
- [4] https://web.eecs.umich.edu/~fessler/course/600/l/lmono.pdf

**Limitations:**

To my knowledge, there's no potential negative societal impact for this submission.

---

> ### Author Rebuttal · Authors · 2023-08-09
>
> We sincerely thank the reviewer for the review and questions. We appreciate the hard work. Below we address the main questions:
>
> > *Selecting thresholds to optimize Marco-average metrics are not new in multi-label community. The author seems not aware of some classical/heuristic approaches. For example, the sCut and SCutFBR methods studied in [1,2,3]. The author should also compare these methods in the experiment sections.*
>
> > *Q1: How does the proposed method relate and connect to other conventional threshold tuning approaches (e.g., [1,2,3])?*
>
> We fully agree with the reviewer that unbudgeted macro-averaged measures are popular in multi-label problems. It is easy to notice that optimization of those metrics boils down to solving $m$ independent binary classification problems. Each such binary problem can be solved using one of two frameworks for optimizing complex performance measures. The methods from the cited papers can be seen as implementation of the optimal strategy for the Population Utility (PU) framework. For the Expected Test Utility (ETU) framework, the exact optimization on a test set for those measures is of quadratic complexity. An approximate solution can be obtained in linear time. For more detailed discussion on PU and ETU frameworks and optimization of macro- and micro-averages in multi-label classification we refer the reviewer to these papers: [4, 5, 6].
>
> In this paper, we optimize macro-averaged measures with budgeted@k predictions in the ETU framework. This additional constraint is critical in many practical applications of extreme multi-label classification such as recommender systems and search engines. Unlike in the case of unbudgeted metrics, this constraint does not allow us to solve the problem for each label independently, and hence requires more sophisticated treatment. Therefore, our considerations are novel and not trivial.
>
> ---
>
> > *Q2: How does the proposed method compare empirically to other conventional threshold tuning approaches (e.g., [1,2,3])?*
>
> The conventional thresholding methods do not have the restriction on the number of predicted labels. Therefore, a direct comparison of these approaches is not straight-forward. Moreover, the conventional thresholding methods are designed for the PU framework. For ETU without the budgeted predictions, one would rather use different algorithms to find the threshold. For a detailed discussion please refer to [4].
>
> **References for Q1 and Q2:**
> 1. Y. Yang. A Study on Thresholding Strategies for Text Categorization. SIGIR 2001
> 2. Lewis et al. RCV1: A New Benchmark Collection for Text Categorization Research. JMLR 2004
> 3. Fan et al. A Study on Threshold Selection for Multi-label Classification. Technical Report 2007
> 4. Dembczynski et al.: Consistency analysis for binary classification revisited. ICML 2017.
> 5. Koyejo et al.: Consistent Multilabel Classification. NeurIPS 2015.
> 6. Kotłowski, Dembczyński: Surrogate regret bounds for generalized classification performance metrics. Machine Learning 2017.
>
> ---
>
> > *Q3: For the block coordinate ascent algorithm, how sensitive is it to the initial point? If it is a non convex function, I suppose the initialization problem matters? Any better heuristic than "predicting k random labels" that described at line 185?*
>
> In addition to predicting $k$ random labels as the initial point, we also tested top-$k$ labels with the highest marginals $\eta$. Both initialization methods gave very similar final results in all cases, so we omitted this experiment in the submitted paper. Also, please note that all the experiments were repeated 5 times with different random initial points. We reported the standard deviation in Tables 6 and 7. In most cases, the deviation is no more than 0.1%.
>
> ---
>
> > *Q4: For Line 195-198, monotonicity alone is not enough to ensure convergence. Any more thorough/technical statement for the convergence analysis of the proposed block coordinate ascent algorithm?*
>
> We thank for this interesting comment. For metrics taking values in a bounded range and $\epsilon > 0$, the algorithm can only make up to $O(1/\epsilon)$ steps in the `while’ loop, as each step must decrease the objective by at least $\epsilon$. However, even when $\epsilon=0$, the algorithm terminates after a finite number of steps. This is because each step must strictly decrease the objective (otherwise, the algorithm terminates), which is only possible if the algorithm changes at least one prediction (from 0 to 1 or the other way around) among all instances and labels in that step. Since the number of possible predictions is finite (precisely, $\binom{m}{k}^n$), this guarantees that the algorithm will stop after a finite number of steps.
>
> In practice, despite small $\epsilon$, the algorithm usually terminates after a few iterations (in the worst observed cases, it was a few tens of iterations). In Tables 6 and 7 in the Appendix, one can observe that the Greedy variant (one-pass algorithm) obtains good results in most cases, and based on that, larger improvements than $\epsilon$ are expected.
>
> In Table 1 in the PDF attached to the global response, we also present the number of iterations and CPU time that was used for our inference procedures.
>
> ---
>
> **Minor issues**
>
> We thank for pointing out the typographical issues. We will correct them in the final version of the paper.

---

> > ### Comment · Reviewer_4ZEa · 2023-08-20
> >
> > I appreciate the author's response. I will keep my score unchanged.

---

### Author Rebuttal · Authors · 2023-08-09

We thank the reviewers for their thorough comments and questions. We answer them in specific responses to the reviewers. We use the global comment to attach a PDF with additional plots containing results for “meta-measures” that combine instance and macro-averaged measures. These plots have been prepared as a part of the response to Reviewer ELgR. A detailed response has been given in the specific response to this Reviewer. In the PDF, we also include a table with the running time and the number of iterations of the proposed inference method in addition to the response to Reviewer 4ZEa and 4FGQ.

---

### Decision · Program_Chairs · 2023-09-21

**Decision:**

Accept (poster)

**Comment:**

Reviewers were unanimously supportive of this paper, which provides a set of new top-k metrics for better capturing performance of multi-label classifiers on tail labels. This is supplemented with rigorous theory and solid experimental results. In all, this work should be of broad interest to the community.